# Allosteric Regulation of G-Protein-Coupled Receptors: From Diversity of Molecular Mechanisms to Multiple Allosteric Sites and Their Ligands

**DOI:** 10.3390/ijms24076187

**Published:** 2023-03-24

**Authors:** Alexander O. Shpakov

**Affiliations:** Sechenov Institute of Evolutionary Physiology and Biochemistry, Russian Academy of Sciences, 194223 St. Petersburg, Russia; alex_shpakov@list.ru; Tel.: +7-905-275-38-83

**Keywords:** G protein-coupled receptor, allosteric site, allosteric modulator, pepducin, heterotrimeric G protein, autoantibody, thyroid-stimulating hormone receptor, luteinizing hormone receptor, proteinase-activated receptor, chemokine receptor

## Abstract

Allosteric regulation is critical for the functioning of G protein-coupled receptors (GPCRs) and their signaling pathways. Endogenous allosteric regulators of GPCRs are simple ions, various biomolecules, and protein components of GPCR signaling (G proteins and β-arrestins). The stability and functional activity of GPCR complexes is also due to multicenter allosteric interactions between protomers. The complexity of allosteric effects caused by numerous regulators differing in structure, availability, and mechanisms of action predetermines the multiplicity and different topology of allosteric sites in GPCRs. These sites can be localized in extracellular loops; inside the transmembrane tunnel and in its upper and lower vestibules; in cytoplasmic loops; and on the outer, membrane-contacting surface of the transmembrane domain. They are involved in the regulation of basal and orthosteric agonist-stimulated receptor activity, biased agonism, GPCR-complex formation, and endocytosis. They are targets for a large number of synthetic allosteric regulators and modulators, including those constructed using molecular docking. The review is devoted to the principles and mechanisms of GPCRs allosteric regulation, the multiplicity of allosteric sites and their topology, and the endogenous and synthetic allosteric regulators, including autoantibodies and pepducins. The allosteric regulation of chemokine receptors, proteinase-activated receptors, thyroid-stimulating and luteinizing hormone receptors, and beta-adrenergic receptors are described in more detail.

## 1. Introduction

G protein-coupled receptors (GPCRs), located in the plasma membrane, are the largest superfamily of receptor (sensory) proteins in multicellular eukaryotes. GPCRs have been found in fungi [1,2,3], plants [4], and in all studied invertebrates and vertebrates [5,6,7,8,9], including trypanosomes [10] and ciliates [11]. At the same time, the yeast *Saccharomyces cerevisiae* has only 3 genes encoding GPCRs [12], the slime mold *Dictyostelium discoideum* has 55 such genes [13], while in the human genome there are more than 800 genes for GPCRs [14]. Prototypes of the structural domains of both GPCRs and the adapter and regulatory proteins that interact with them appeared at the earliest stages of evolution, already at the level of prokaryotes and unicellular eukaryotes [2,9,15]. During the early evolution of GPCRs, different structural models of these receptors existed, including hybrid constructs that consisted of an N-terminal GPCR-like molecule and a C-terminal catalytic phosphatidylinositol phosphate kinase, which were identified in some representatives of lower eukaryotes [11,16,17].

Through GPCRs, various extracellular signals, including photons, protons, hormones, neurotransmitters, growth factors, nutrients, metabolites, and odorants, exert their regulatory effects on target cells. The result of the interaction of the GPCR with them is its transition to an active conformation and triggering of intracellular signaling cascades, which, through genomic and non-genomic mechanisms, regulate fundamental cellular processes, such as growth, metabolism, differentiation, apoptosis, and autophagy. The fact that the therapeutic effect of about a third of pharmacological drugs used in medicine is due to their influence on GPCRs and their signaling pathways [18,19] is the basis for the great importance of studying the molecular mechanisms of GPCR regulation.

Over the past two decades, many paradigms regarding the functioning of GPCRs and the transduction of hormonal signals through them have undergone revision. In the 1990s, it was generally accepted that signal transduction from the hormone-activated GPCR occurs almost exclusively through the heterotrimeric G proteins. However, later evidence was obtained that various adapter and regulatory proteins, primarily β-arrestins, which are able to interact specifically with the hormone-activated receptor, are also involved in signal transduction, thereby regulating and modulating GPCR-mediated intracellular signaling [20,21,22,23,24]. Arrestins are an evolutionarily ancient family of structurally and functionally related scaffolding proteins, which in vertebrates includes two retinal arrestins and two non-visual arrestins, β-arrestin1 (arrestin2) and β-arrestin2 (arrestin3), the latter being widely distributed in various tissues and are of great importance for GPCR-mediated signaling transduction [25,26]. In early studies of the role of β-arrestins in the GPCR-signaling, it was recognized that these proteins are responsible for the desensitization of GPCRs and also mediate the endocytosis of ligand-receptor complexes. The study of the further processing of these complexes showed that they are first internalized and transported to early endosomes. Then, the GPCRs are either recycled back to the plasma membrane in an active, ligand-free state, or they are sorted within the endocytic pathway and packaged into intraluminal vesicles, forming multivesicular bodies that fuse with lysosomes, and this leads to complete degradation of the receptors [27,28]. At the turn of 1990–2000, the participation of β-arrestins in GPCR-mediated regulation of the mitogen-activated protein kinases (MAPKs) and several other effector proteins and transcription factors were demonstrated [21,24,29,30,31,32,33,34,35]. This has changed the paradigm of the exclusive role of the heterotrimeric G proteins as signal transducers in GPCR-signaling. At the same time, the question of whether β-arrestins are able to carry out signal transduction independently of G proteins remains open to date. In recent years, evidence has been obtained that in the absence of G proteins, β-arrestins are unable to activate MAPKs, which indicates the need for coordinated participation of G proteins and β-arrestins in signal transduction and does not support the concept of G protein-independent β-arrestin signaling [36,37,38,39]. It is also shown that β-arrestins are able to modulate the interaction of GPCRs with different types of G proteins, as is observed in the case of the type 1 parathyroid hormone receptor (PTH1R) coupled to both G_s_ and G_q/11_ proteins [40].

The signaling functions of β-arrestins are determined by the pattern of GRK-mediated phosphorylation of intracellular loops (ICLs) and the cytoplasmic C-terminal domain of the receptor, which predetermines a large number of variants for β-arrestin-dependent effects [41,42,43]. Phosphorylation of the intracellular regions by GRK2 and GRK3 usually leads to the β-arrestin1-dependent desensitization of GPCRs and their trafficking into the cell, while GRK5- and GRK6-mediated phosphorylation induces β-arrestin2-dependent signaling, although this is not a general rule [44]. The signaling functions of β-arrestins also depend on the spatial positioning of phosphorylation sites, the so-called “barcode”. In one variant of the location of phosphorylation sites in the V2-vasopressin receptor and AT1a-angiotensin II receptor, β-arrestin1 stimulates the MAPK cascade, while in another location in the B2-bradykinin receptor, β-arrestin1 inhibits ERK1/2 activity, and a change in the “barcode” in the mutant receptor leads to a change in β-arrestin1-mediated stimulating effect on the MAPK cascade [45]. The factors that influence the recruitment of β-arrestins, the dynamics of the formation of their complex with the GPCR, and the further dissociation of β-arrestins from the internalized receptor complex are currently being intensively studied. It is assumed that for some types of GPCRs, the lipid composition of the plasma membrane, including the content of phosphoinositides, is important for the implementation of these processes. For these GPCRs, phosphoinositides, including phosphatidylinositol-4,5-bisphosphate (PI(4,5)P_2_), are required to form the functionally active GPCR–β-arrestin complexes [46]. A decrease in their content in the membrane during endocytosis leads to the dissociation of the GPCR–β-arrestin complexes and recycling of the free receptor into the plasma membrane. On the other hand, after GRK phosphorylation, some types of GPCRs do not require phosphoinositides for the formation of such complexes, which remain stable during endocytosis and continue to perform signaling functions in endosomes, where phosphoinositide’s content is significantly reduced [46]. It is important that the stability of the GPCR–β-arrestin complexes in endosomes is the basis for the implementation of “intracellular” signaling through β-arrestins and different types of G proteins in the endosomes and the Golgi apparatus [47,48]. Thus, the paradigm of GPCR-mediated signal transduction and generation of second messengers only in the plasma membrane has been revised. Moreover, there is now evidence that for a number of GPCRs, it is “intracellular” signaling that makes the main contribution to the production of second messengers and the regulation of intracellular effector systems and GPCR-dependent gene transcription [48,49].

At an early stage of GPCR studies, it was believed that after high-affinity binding to an orthosteric agonist, the receptor is transformed into a single active conformation, in which it becomes capable of stimulating a certain type of G protein and its coupled enzyme, which generate second messengers. As a result, GPCRs have been subdivided into families of G_s_-, G_q/11_-, G_i/o_- and G_12/13_-coupled receptors based on the type of G protein that is preferentially activated. As is known, the activation of the G_s_ protein leads to the stimulation of adenylate cyclase (AC) and an increase in the intracellular cAMP level, and the activation of G_q/11_ proteins causes the stimulation of phosphoinositide-specific phospholipase Cβ (PLCβ) and the generation of second messengers, such as intracellular calcium and diacylglycerol. The activation of G_i/o_ proteins, the main donors of Gβγ-subunits [50], affects the activity of the Gβγ-dimer-sensitive isoforms of AC and G protein-regulated ion channels, modulates the activity of PLCβ, and inhibits (through the Gα_i_-subunit) hormone-stimulated AC activity. However, over the past 20 years, there has been extensive evidence that an agonist-coupled GPCR is able to simultaneously, albeit with varying efficiency, activate several types of G proteins, as well as at least two types of β-arrestins, triggering several intracellular signaling cascades (for details see [8,34]). This is due to the fact that GPCRs are able to exist in several active conformations that are in dynamic equilibrium and are quite close in energy characteristics but each of which mediates the activation of a certain type of G protein or β-arrestin [51,52]. The time during which an agonist-bound receptor can be in each such conformation largely determines its ability to activate a certain transducer protein. The first data on the possibility of GPCRs being in several active conformations, which depends, among other things, on the nature of the agonist, were obtained in the late 1990s [53,54,55] and confirmed the concept of selective signaling agonism, also referred to as “agonist-specific trafficking of receptor signaling”, formulated by Terry Kenakin back in 1995 [56].

The multiplicity of active conformations of the agonist-bound GPCR, as well as the various scenarios of its signaling and traffic that depend on this, suggest the existence of mechanisms that affect the stability, dynamics, and ratio of these conformations, and thus determine the formation of a functionally active complex with a certain transducer and (or) regulatory protein. Along with the nature of an orthosteric agonist, the physicochemical properties and lipid composition of the plasma membrane, ionic strength, composition and acidity (pH) of the extra- and intracellular medium, the redox potential, the concentration and ratio of amino acids, lipids, oligo- and polypeptides, and nutrients are of great importance. These physicochemical factors and chemical substances, including the simplest ions, can affect the stability of the ligand-bound GPCR and its biased activity towards intracellular effectors, thereby predetermining the intensity and complexity of the cell response to an external stimulus. All this formed the concept of allosteric regulation of GPCRs and their decisive role in the biased regulation of intracellular signaling pathways and changed the paradigm, according to which ligands of the orthosteric site were considered the key, if not the only, regulators of GPCRs (for more details, see [57]).

As is known, the site of the receptor to which its endogenous ligand specifically binds is designated as orthosteric. Depending on the class of receptors, it can be located within the transmembrane domain (TMD) near the extracellular entrance to the transmembrane tunnel, in the extracellular loops (ECLs), or in the large extracellular domain of the GPCR. Binding of an orthosteric agonist to a receptor typically results in a significant stimulating effect, inducing receptor activation. In most cases, the affinity of orthosteric ligands for the GPCR is higher than that of allosteric ligands, although this is not a general rule. Compared with orthosteric agonists, interaction of the receptor with allosteric ligands generally results in more moderate and selective effects on basal GPCR activity and also modulates receptor activity stimulated by orthosteric agonists. The GPCR contains not one but several allosteric sites that differ in localization, configuration, and functional activity. These sites also differ in their influence on the conformation and accessibility of the orthosteric site, the efficiency of the interaction with transducer and regulatory proteins, and, for some receptor types, on the ability of GPCR to form homo- and heterooligomeric receptor complexes [57,58,59,60,61,62].

Allosteric sites can be topologically isolated or partially overlap with each other, as well as with an orthosteric site, which suggests the existence of both relatively simple and more complex reciprocal relationships between them. In most cases, the orthosteric site is highly conserved among GPCRs that are activated by the same or structurally related ligands, while allosteric sites are characterized by significant structural variability, although this is not a general rule for them [63,64]. As a result, the action of allosteric regulators specific for a certain type of GPCR is highly selective and often biased towards intracellular targets. There are few examples of how allosteric regulators specific to one GPCR are able to bind to allosteric sites of another receptor. This has been shown for allosteric sites located at the interface between the TMD and ICLs in structurally related types of GPCRs, such as chemokine receptors [65,66], β_1_- and β_2_-adrenergic receptors (β_1_/β_2_-ARs) [67,68], and the glucagon and glucagon-like peptide-1 receptors [69].

The currently developed paradigm of the decisive role of allosterism in the functioning of GPCRs is in good agreement with the rethinking of the principles of organization of receptor complexes. In the recent years, there has been much evidence that some GPCRs function as di- and oligomeric complexes, and these complexes can be both homo- and heteromeric, including different types and subtypes of receptors, often implementing different functions. The formation of complexes not only significantly changes the affinity of GPCRs for orthosteric agonists but also determines their specificity and effectiveness for various types of G proteins and β-arrestins and also affects the endocytosis of ligand–receptor complexes and their further fate in the cell [70,71]. Each GPCR protomer in the receptor complex functions as an allosteric regulator with respect to other protomers of this complex. The properties of allosteric regulators are also inherent in heterotrimeric G proteins, β-arrestins, and receptor-activity-modifying proteins (RAMPs) [72,73,74,75]. Their effect on the receptor affinity for an orthosteric agonist largely depends on the functional state and set of these proteins, and in the case of G proteins, the subunit composition and stability of the heterotrimeric complex are of great importance [34]. According to modern concepts, GPCRs and signaling proteins interacting with them form multicomponent complexes that are stabilized by anchoring on the scaffolding proteins, multifunctional regulators of intracellular signaling, which can also include β-arrestins [8,32,76]. Such complexes also exhibit a wide range of allosteric effects, which can affect not only the GPCR affinity for orthosteric and allosteric ligands but also the signaling bias and features of intracellular GPCR traffic.

This review is devoted to the problem of allosteric regulation of various classes of GPCRs and also focuses on the multiplicity of allosteric sites, their location and functional role, as well as on various regulators and modulators that specifically interact with these sites. Considerable attention is paid to the diversity of endogenous and synthetic allosteric regulators that act on the receptor both from outside and inside the cell, as well as by changing the contact between the membrane lipids and the receptor TMD.

## 2. Classification of Allosteric GPCR Regulators

According to the ability to influence the basal and orthosteric/allosteric agonist-stimulated activity, the ligands of GPCR allosteric sites can be divided into allosteric modulators that have no intrinsic activity and allosteric regulators that affect GPCR activity in the absence of orthosteric agonists [58]. In the case of allosteric modulators, the ligand, by binding to the allosteric site, changes or retains the affinity of the orthosteric agonist to GPCR and/or its ability to activate the receptor, which is assessed by its maximum stimulating effect but has no intrinsic activity (Table 1). Allosteric ligands that have their intrinsic activity can function as full agonists, inverse agonists, and neutral antagonists, and their action is independent of orthosteric site occupancy (Table 1). Such independence of the action of allosteric ligands can be realized only when the orthosteric and allosteric sites do not overlap and do not interact through ligand-induced conformational rearrangements [58]. When an allosteric ligand acts as a full agonist and affects the affinity and/or potency of an orthosteric agonist, it is classified as ago-PAM or ago-NAM (Table 1). In the case when allosteric ligand reduces the effectiveness of an orthosteric agonist but increases its affinity to GPCR, it is classified as a PAM-antagonist [77].

Regulatory influences caused by the specific binding of ligands to the orthosteric and allosteric sites of the GPCRs are reciprocal. Just as binding of a ligand to an allosteric site can change the binding characteristics and activation pattern of an orthosteric site, binding of a ligand to an orthosteric site can change the accessibility and regulatory properties of one or more allosteric sites. Given the multiplicity of allosteric sites, the mechanisms of such relationships between receptor sites can be quite complex. Moreover, depending on the nature and binding characteristics of orthosteric ligands, the same allosteric regulator can influence the affinity and efficiency of these ligands in different ways, functioning as a PAM, NAM, or SAM [58,78,79]. Thus, the assignment of a ligand to a certain group of allosteric regulators or modulators is relative. In each specific case, the pharmacological profile of the allosteric ligand largely depends on the nature of the receptor-bound orthosteric agonist, the structural and functional characteristics of the receptor (phosphorylation, N-glycosylation, and other post-translational modifications), the formation of homo- or hetero-oligomeric complexes, the microenvironment (multicomponent complexes with transducer, adapter, and regulatory proteins), as well as on the physicochemical properties of the plasma membrane, ionic composition, and ionic strength and acidity of the extracellular and intracellular environment.

Given the variety of regulatory influences of allosteric ligands on the interaction of orthosteric agonists with the GPCR, in the recent years a model has been widely used that describes a ternary complex that includes GPCR and receptor-bound allosteric and orthosteric ligands. The formation of a ternary complex is described by equations A + R + B ⟷ AR + B (K_A_) ⟷ ARB (K_B_/α) in the case when, at the first stage, the GPCR forms a complex with the orthosteric agonist A, and by equations A + R + B ⟷ A + RB (K_B_) ⟷ ARB (K_A_/α), when, at the first stage, the receptor forms a complex with the allosteric ligand B (K_A_ and K_B_, the equilibrium dissociation constants for the GPCR-orthosteric agonist and GPCR-allosteric ligand complexes, respectively; α, the factor of binding cooperativity between the orthosteric agonist A and allosteric ligand B) [80]. Therefore, the effect of an allosteric ligand on the affinity of an orthosteric agonist is described by the factor α. In turn, its effect on the efficacy (maximum regulatory effect) of an orthosteric agonist is described by the factor β. When the allosteric ligand increases the affinity and efficacy of orthosteric agonist, the factors α and β are above 1 (PAM) (Table 1). If the influence of the allosteric ligand is the opposite, then these factors have values below 1 (NAM). In the absence of a significant effect, the α and β are equal to 1 (SAM). To assess the intrinsic activity of allosteric ligands, the factor τ is used, which for full agonists has values above 1 (full agonist, ago-PAM, and ago-NAM), and for an inverse agonist or neutral antagonist it has values below 1. For “pure” allosteric modulators (PAM, NAM, SAM, and PAM-antagonist), the factor τ is equal to 1 (Table 1). However, for each specific case and for each specific “allosteric ligand–orthosteric ligand” pair, the values of α, β, and τ may vary.

A separate group consists of compounds that are able to simultaneously interact with both orthosteric and allosteric sites, which are classified as bitopic GPCR ligands [79,81,82,83]. They have two pharmacophores: one which binds with the orthosteric site, and the other binds with the allosteric site. If these sites are spatially separated in the receptor, then the pharmacophores in the bitopic ligand must be connected with a flexible linker, the length of which exactly corresponds to the distance between the orthosteric and allosteric sites. At the same time, it is important that the linker does not significantly affect the conformational rearrangements in the receptor induced by its activation by orthosteric and (or) allosteric agonists [79,82].

## 3. Localization and Number of Allosteric Sites in GPCRs

Allosteric sites can be localized in all structural domains of the GPCRs (Table 2). Estimation of the number and localization of allosteric sites in each receptor is a very difficult task, although significant progress in this direction has been made due to the development of new approaches for the identification of GPCR allosteric sites [61,62,84,85].

The FTMap (http://ftmap.bu.edu/; 1 January 2023) and FTSite (http://ftsite.bu.edu/; 1 January 2023) programs are widely used to search for allosteric sites in GPCRs. The FTMap has been successfully tested to search for allosteric sites in β_2_-AR, A2A-adenosine, and M2-muscarinic receptors [86,87,88]. In 2019, using the FTMap and its modified version FTSite, a large-scale study was carried out to identify allosteric sites in 17 GPCRs belonging to their various families. It is important that for all these receptors there was information on their allosteric regulation and X-ray diffraction data, as a result of which the data obtained were compared with the available experimental evidence on the localization of such sites in the studied receptors [84]. For allosteric sites located within the TMD, both near the entrance to the transmembrane tunnel and in the internal cavity of the TMD, the predictive ability of the FTMap and FTSite was 80 and 88%, respectively. For sites located in hydrophilic loops and on the outer-lateral surface of the seven-helix transmembrane (TM7) bundle, the predictive ability was significantly lower. The overall predictive ability of the FTMap and FTSite programs for all allosteric sites was 69 and 76%, respectively [84]. In the case of hydrophilic loops, the limitations of the approaches used are due to the high mobility of these loops and, in most cases, the lack of reliable X-ray diffraction data, while for sites located on the outer-lateral surfaces of the TMD, such limitations are due to the features of FTMap and FTSite adapted mainly to hydrophilic and globular proteins without taking into account their interaction with the hydrophobic phase.

Molecular dynamics methods can also be used to search for allosteric sites, and the most interesting here is the identification and study of allosteric sites located on the outer-lateral surface of the TM7 bundle and, therefore, available for interaction with membrane lipids, as was recently shown for β_2_-AR, glucagon receptor, M2-muscarinic receptor, P2Y1 purinergic receptor, D_2_-dopamine receptor, type 2 proteinase-activated receptor (PAR2), and C5a anaphylatoxin chemotactic 1 receptor [85,89,90]. Since these sites are involved in the formation and stabilization of GPCR complexes (homo- and hetero-oligomeric), they mediate the allosteric effects of receptor di- and oligomerization on their functional activity. Along with this, these sites are targets for plasma membrane lipids, which, therefore, can function as allosteric modulators as demonstrated for cholesterol and phospholipids [91,92,93].

Recently, a comprehensive analysis of the topology of allosteric sites in GPCRs, which also makes it possible to predict “orphan” allosteric sites, was performed using the in-silico docking of small molecules probes, during which 557 GPCR structures for 113 receptors were studied [61]. As a result, a “pocketome” of allosteric sites was constructed for different classes of GPCRs (A, B1, B2, C, D1, and F). In total, for receptors of the largest class A, in the cavities between the hydrophobic helical regions forming the 7TM bundle up to 11 allosteric sites were identified, the existence of which has experimental evidence, and up to 8 sites that can be considered as potential allosteric sites. Along with this, Interhelical Binding Site 1 (IBS1) and adjacent secondary binding sites IBS2 and IBS3 were identified in the central part of the TMD [61]. In GPCRs of classes A and B, IBS1 functions as an orthosteric site, and in GPCRs of class B, which are activated by peptide hormones, in addition to IBS1, the segments of the hydrophilic ECL and the outer vestibule of the transmembrane tunnel also take part in the formation of the orthosteric site. In GPCRs of class C, which have a significant ectodomain containing an orthosteric site, the IBS1 site does not interact with the orthosteric ligand, usually a large glycoprotein hormone, and functions as an allosteric site. Of considerable interest are the results of the structural and functional analysis of two “orphan” allosteric sites OS5 and OS9, which are located at the lower portion of the 7TM bundle. The OS5 site is located in the cavity between the fifth and sixth transmembrane regions (TM5 and TM6), while the OS9 is in the cavity formed by the intracellularly oriented ends of TM1 and TM7 and the H8 helix. The cavity that forms the OS5 is highly conserved among GPCRs of classes A and B1, while the OS9 cavity is structurally similar among all known GPCRs, except class F. Mutations of amino acid residues that form the OS5 and OS9 sites in β_2_-AR and M3-muscarinic receptors had a strong effect on both the activation of G proteins and the recruitment and activation of β-arrestins, and these effects were significantly different, which may indicate a bias of structural changes in the OS5 and OS9 sites in relation to the intracellular signaling pathways [61]. Mutations in the locus corresponding to the OS9 in the AT1-angiotensin receptor led to changes in the activity of the G_q/11_-protein- and β-arrestin-dependent cascades [94], which confirms the functional importance of this site in allosteric regulation of class A GPCRs. Thus, new tools have been obtained for the search and validation of allosteric sites in GPCRs, and this may soon allow mapping each of the classes of these receptors by the location and functional activity of allosteric sites, which is important for an allosteric ligand-based drug discovery and design [61,62].

However, it should be noted that the identification of allosteric sites in hydrophilic loops, and in particular, in large extracellular and cytoplasmic domains, is a very complex and far from being solved, especially since these sites, as a rule, are targets for a large number of proteins involved in the GPCR-signaling, as well as for autoantibodies to the extracellular regions of GPCRs.

## 4. Diversity of Endogenous Allosteric Regulators of GPCRs

The multiplicity of GPCR allosteric sites, which differ both in topology and environment, provides for the presence of allosteric regulators of different chemical nature, both extracellular and intracellular. At present, such regulators have been found in almost all classes of compounds from simple ions to large protein complexes, including those involved in GPCR-dependent signaling pathways (Figure 1). Most of them function as allosteric modulators and have no intrinsic activity [74,75,93,95,96,97,98,99,100]. It should be noted that the molecular mechanisms and their targets in GPCR molecules differ significantly, but their regulatory effects are based on allosteric effects on the basal and orthosteric agonist-stimulated activity of different GPCR classes.

### 4.1. G Proteins and β-Arrestins

The most important and characteristic group of allosteric GPCR regulators are signal proteins, such as heterotrimeric G proteins, β-arrestins, and RAMPs, which are capable of forming a functionally active complex with the receptor and are responsible for the transduction of the hormonal signal into the cell. The G proteins and β-arrestins interact with different intracellular regions of GPCRs and with the cytoplasmic part of their TMD, and the most important site for such interaction is the cavity formed by TM6 and the hydrophobic helix H8. Both the α5 helix of the α-subunit of the G protein, which is responsible for the formation of the complex between the ligand-activated GPCR and the G protein, and the “finger” region of β-arrestin, which is responsible for the formation of the complex between the GRK-phosphorylated receptor and β-arrestin, interact with this cavity [101]. Displacement of the G protein α-subunit by β-arrestin from this cavity is the main mechanism mediating the termination of G protein-mediated signaling after GPCR phosphorylation. At the same time, RAMPs interact with the TMD of GPCRs and, in the case of class B GPCRs, with the extracellular domain [102,103], indirectly affecting the interaction of receptors with transducer proteins by changing the conformation of TM6 and ICL2 [103].

As is known, as a result of the interaction with a ligand-activated receptor, a heterotrimeric G protein dissociates into a GTP-bound Gα-subunit and a Gβγ-dimer, which become available for binding to effector proteins (AC, PLCβ, phosphatidylinositol-3-kinase, G protein-activated ion channels, and others). In turn, GTP-free G protein, which forms a stable αβγ-heterotrimeric complex, increases the affinity of the receptor for an orthosteric agonist, and this is due to the stabilization of the “closed” conformation of the GPCR, which is characterized by restricted access to the hormone-binding site [74]. Due to the difficulty in dissociating the agonist from the “closed” conformation of the receptor, the stability of the GPCR–orthosteric ligand complex is increased, resulting in an increase in the affinity of the receptor for agonists. It is important to note that the interaction of the constitutively activated GPCR with the G protein also results in the stabilization of the “closed” conformation, which prevents the binding of orthosteric agonists with the receptor and makes GPCR signaling independent of them [74].

Along with GTP-free G proteins, antibodies that mimic them also stabilize the “closed” conformation of the GPCR, as was shown for nanobody Nb80 produced in response to agonist-activated β_2_-AR [95]. Antibodies have been developed that act in the opposite way, stabilizing the inactive GPCR conformation, which is characterized by low affinity for the ligand as demonstrated for nanobody Nb60. This proves the existence of both positive and negative allosteric mechanisms of the influence of G proteins and antibodies mimicking them on binding and efficiency of the orthosteric agonist [75]. The receptor allosteric site responsible for interaction with the G protein is located at the interface between the cytoplasmic ends of the TMs and the proximal regions of the ICL2 and ICL3, and it interacts specifically with the Gα-subunit C-terminal segment and other GPCR-interacting regions of the G protein [104]. As in the case of the agonist–GPCR–G protein ternary complex, allosteric interactions were demonstrated for the agonist–GPCR–β-arrestin ternary complex, the existence of which was postulated back in 1997 by Gurevich and colleagues [105]. Binding of β-arrestins to GPCRs increases the affinity of the receptor for orthosteric agonists [72,75], especially those that are biased towards β-arrestin signaling [37].

The preference for the formation of a certain ternary complex for GPCRs capable of interacting with several transducer proteins depends on a number of factors, but their role and influence on the stability of these complexes is not well understood. There is no doubt that the allosteric mechanisms that regulate the formation of a functionally active ternary complex play a decisive role in biased agonism, leading to selective activation of a certain type of G protein or β-arrestin by an orthosteric agonist [57]. At the same time, it is not completely clear what is the trigger for the formation of a biased ternary complex. Such triggers can be a biased orthosteric agonist, which recruits a certain type of transducer protein into the complex, or the formation of a pre-activation complex between the agonist-free GPCR and an inactive G protein or β-arrestin, which creates suitable conditions for effective binding of a biased agonist and triggering signal transduction [106,107,108].

In favor of the first model, the so-called “ligand” mechanism, evidence suggests that agonist binding to the receptor is able to recruit inactive, GDP-bound, G proteins with the formation of a ternary complex and, thereby, triggers signal transduction [109]. At the same time, the limited availability of a certain G protein and the competition between the different types of G proteins for binding to the ligand-activated GPCR are not consistent with the data on the high rate and high efficiency of the formation of the ternary complex and the transduction of the hormonal signal into the cell. The “ligand” mechanism is unable to explain the fact that, in the absence of an orthosteric ligand, constitutively activated GPCRs recruit and activate G proteins [110,111]. Moreover, such receptors selectively activate certain types of G proteins or β-arrestins [112,113]. The alternative mechanism involves a different sequence of events. At the first stage, a pre-activation complex is formed that includes a ligand-free GPCR and an inactive form of a certain G protein, and this pre-determines not only the binding of a certain orthosteric agonist but also biased agonism [108]. Currently, the formation of such pre-activation complexes has been postulated for a number of GPCRs [114,115,116,117,118,119]. In the pre-activation complex, the interaction between TM3 and TM6 is weakened, and their conformational mobility increases. Although this is not enough for full activation of the receptor, the weakening of the interaction between TM3 and TM6 facilitates the agonist-induced shift of TM6 outward, which changes the location of the α5-helix of the Gα-subunit in the intracellular vestibule of the GPCR transmembrane tunnel, induces a decrease in the affinity of the Gα-subunit for GDP, and provides an effective interaction of activated G protein with receptor ICLs [108]. Instead of G proteins, the pre-activation complex may contain β-arrestins, which stabilize the conformation of the orthosteric site, which has a high affinity for β-arrestin-biased agonists [106,120].

### 4.2. The Other Endogenous Allosteric Regulators, including Ions and Lipids

Along with highly specialized components of GPCR signaling, endogenous allosteric regulators of GPCR include some classes of antibodies produced against extracellular regions of receptors, small proteins, oligopeptides, steroids, fatty acids, amino acids, and even simple ions [93,96,97,98,99,100]. Some allosteric regulators specifically affect a particular receptor or closely related GPCRs, while others are non-specific and affect the activity of a large number of GPCRs belonging to different families. Such receptor-nonspecific activity has been demonstrated for sodium ions, which are NAMs for a large number of GPCRs, stabilizing their inactive state and reducing their stimulation by full agonists [98,121,122]. Sodium ions are able to bind to the allosteric site of these receptors located in the internal cavity of their TMD. This site, a target for sodium ions, includes several highly conserved amino acid residues, the most important of which is the negatively charged aspartic acid located in TM2. This residue is critical for the translocation of sodium ions to the allosteric site and for their retention by ionic bonds, and the replacement of aspartic acid with alanine blocks the NAM activity of these ions [121,123]. Sodium ions make a significant contribution to the selectivity of activation of intracellular cascades by agonists, as shown for opioid receptors [124,125]. Using site-directed mutagenesis and molecular docking, a sodium ion-binding site has been identified in all classes of GPCRs. It is located in the middle of the transmembrane tunnel and includes amino acid residues from TM1, TM2, TM3, TM6, and TM7 [61]. Zinc and magnesium ions, being allosteric modulators for a large number of GPCRs, can function both as a PAM and NAM [97,98,126]. It should be noted that a certain contribution to the effects of magnesium ions can be made by their ability to stimulate the GDP/GTP exchange and GTPase activity of G proteins [127].

Among lipids, cholesterol is the most important allosteric regulator of GPCRs, and its effects on receptor activity are highly dependent on the content of cholesterol in the membrane, the type of GPCR, and the nature of the orthosteric ligand [93,128,129]. A significant number of GPCRs contain consensus motifs for cholesterol binding (Cholesterol Recognition/Interaction Amino Acid Consensus motif, CRAC), which were first identified in β_2_-AR [130] and then found in other GPCRs, although often in a modified form [93,131,132]. Along with cholesterol, other lipids, such as phosphatidylserines [91] and phosphoinositides, including PI(4,5)P_2_ [92], are also involved in the allosteric regulation of GPCRs. As noted above, some evolutionarily ancient forms of GPCRs were hybrids of GPCR and phosphatidylinositol phosphate kinase catalyzing the synthesis of PI(4,5)P_2_ [11,16,17]. As a result, it can be assumed that this phosphoinositide could already then function as an allosteric modulator of GPCRs. Different lipids act on receptors through binding to different allosteric sites and, accordingly, affect their activity in different ways. For the 5-HT_1A_ (serotonin) receptor, it has been shown that cholesterol directly affects the binding characteristics of the orthosteric site, while phospholipid phosphatidylinositol 4-phosphate interacts with the intracellular site and improves the functional coupling between the agonist-activated receptor and the G_i/o_-protein [133]. More complex allosteric effects of lipids on GPCR activity are possible due to different mechanisms of their action on receptors and their complexes with the other signal proteins. In addition to direct interaction with GPCR allosteric sites, lipids, affecting the physicochemical properties of the plasma membrane (fluidity, tension, and mechanical stability), are able to change the conformational characteristics of GPCRs and the kinetics of its activated states, as shown for photosensitive rhodopsin [134]. Lipids can affect subcellular compartmentalization and stability of complexes between GPCRs and different regulatory and adapter proteins, as well as the formation of caveolae and lipid rafts, which make a significant contribution to the functional activity of GPCRs [135,136]. This is especially relevant in connection with the concept of “spatial bias” in signaling from the plasma membrane to intracellular compartments, such as endosomes, the Golgi apparatus, and nuclear membranes [47,137,138]. All of these mechanisms can work together, and thus affect GPCR activity in an unpredictable way making it difficult to assess the contribution of each specific lipid and each specific mechanism to receptor-mediated signaling.

Unexpected was the discovery of the fact that the activity of receptors can be voltage-dependent, and this is due to allosteric mechanisms. As a result, in the recent years, the membrane potential has been considered as a “physicochemical” allosteric regulator [139]. Depolarization can affect both the activity of GPCRs and the efficiency of their stimulation by various ligands in different ways [140]. Substitutions of tryptophan residues (Trp^99^ in TM3 and Trp^422^ in TM7) for alanine in the transmembrane allosteric site of the M2-muscarinic receptor abolished the voltage dependence of agonist affinity, although they did not affect the conformational changes induced by depolarization [141]. All this indicates complex relationships between clusters of amino acid residues in the receptor TMD, which function as voltage sensors, and the activity of allosteric and orthosteric sites [139]. It is possible that depolarization can affect the binding of mono- and divalent ions, in particular sodium ions, to the receptor [142], although experimental data do not yet confirm this [143,144].

## 5. GPCR-Complexes and Allosteric Regulation

Significant allosteric effects on GPCR activity are exerted by their formation of homo- and heteromeric complexes since the affinity of orthosteric and allosteric ligands for receptors, their efficiency, and bias towards intracellular cascades largely depend on the degree of oligomerization and the subunit composition of such complexes [71]. This has been clearly demonstrated for a significant number of class C GPCRs, such as metabotropic glutamate and γ-aminobutyric acid (GABA_B_) receptors, taste receptors, and calcium-sensing receptors. For these receptors, the effect of di(oligo)merization on their functional activity was shown, and the involvement of allosteric mechanisms in it was demonstrated [60,145,146,147,148,149,150,151,152,153,154,155,156] (Figure 1).

In vitro experiments with allosteric modulators of glutamate receptors showed that Ro 64-5229 (mGlu2-specific NAM), biphenylindanone A (mGlu2-specific PAM), and VU0361737 and PHCCC (mGlu4-specific PAMs) did not affect affinity and efficacy of orthosteric agonists for heterodimeric mGlu2/mGlu4-glutamate receptors. At the same time, these modulators were effective in the case of homodimeric glutamate receptors, such as mGlu2/mGlu2 (Ro 64-5229 and biphenylindanone A) and mGlu4/mGlu4 (VU0361737 and PHCCC) [145]. Surprisingly, the simultaneous addition of mGlu2- and mGlu4-specific PAMs did not affect the binding characteristics of the heterodimer receptors, indicating a more complex nature of the PAM’s effects on receptor activity. Under in vivo conditions, the regulation of mGlu2- and mGlu4-glutamate receptors by these modulators differed from that in the in vitro conditions, which was due to the existence of various populations of homo- and heterooligomeric glutamate receptors in the target tissues [146].

Another model of heterodimer-specific allosterism is due to the effect of transactivation or transinhibition of one subunit of the heterodimeric GPCR complex upon binding of the allosteric modulator to its other subunit. In the complex formed by subtypes 2 and 3 of the type 1 taste receptor (T1R2 and T1R3), the orthosteric agonist binds only to the T1R2 subunit, while the allosteric modulators cyclamate (PAM) and lactisol (NAM) regulate the activity of the heterodimeric T1R2/T1R3 by specific binding to the T1R3 subunit [147]. The functional activity of the metabotropic γ-aminobutyric acid receptor (GABA_B_) is due to the formation of a heterodimeric complex formed by the GABA_B_1 and GABA_B_2 subunits, in which the GABA_B_1 binds to the orthosteric agonist, while the GABA_B_2 increases the affinity of GABA_B_1 to the orthosteric agonist and provides signal transduction to the G protein [148]. The compound CGP7930 with PAM activity binds to the TMD of GABA_B_2 subunit, which leads to increased stimulation of the GABA_B_1 subunit by a specific agonist [149]. At the same time, the compound COR758, a recently developed NAM for GABA_B_-receptors, binds to an allosteric site localized in the GABA_B_1 subunit [150] indicating the coexistence of regulatory mechanisms, both dependent and independent of heterodimerization [151].

The reciprocal relationships between complex formation and the activity of allosteric regulators are determined by the localization of allosteric sites and their involvement in the formation of contacts between receptor subunits in the di- and oligomeric complex. This allows allosteric regulators to stabilize such complexes, thereby controlling the binding characteristics of the receptor and its ability to activate transducer proteins or, conversely, prevent the formation of complexes. Binding of the mGlu5-glutamate receptor to PAM (Nb43 antibody and low-molecular-weight compound CDPPB) leads to the formation of a more compact homodimeric complex due to closer contact between the TM6 of both protomers, and this increases the functional response of the receptor to the orthosteric agonist [152]. The compound MPEP with NAM activity for the mGlu5-glutamate receptor has the opposite effect, destabilizing the complex [152].

The convergence of protomers through increased interaction between their TM6 has also been demonstrated for active forms of other receptors, including the GABA_B_ receptor and the calcium-sensing receptor [153]. At the same time, dimeric complexes with close TM3 and TM4 in the mGlu2-glutamate receptor [154] and close TM3 and TM5 in the GABA_B_ receptor [155] correspond to the inactive state of the receptors. There is every reason to believe that the leading role in this also belongs to allosteric sites located in the receptor regions involved in di- and oligomerization. For example, in TM6, according to molecular docking data, several allosteric sites are located: KS7 in the central part of the TM7 bundle, KS8 in its lower portion, and KS9 at the border of the TMD and ICLs [61]. All of them are targets for PAMs [152,156]. At the same time, in the TM3/TM4 and TM3/TM5 contacts, there are a significantly smaller number of such sites: in the first case, KS2 site located in the upper portion of the TM7 bundle, and in the second case, KS5 site located in its lower portion [61].

It cannot be ruled out that allosteric sites that are located outside the molecular determinants responsible for GPCR di- and oligomerization are involved in the formation of receptor complexes. In turn, there are reasons to believe that the formation of complexes significantly affects not only the conformation and binding characteristics of the orthosteric site, but also the corresponding characteristics of allosteric sites, including those that are topologically distant from the protomer contact surface in the GPCR complex. Perhaps this explains the fine targeted regulation of biased agonism through the formation of di- and olgomeric complexes since, in this case, the conformation of allosteric sites that are located in the lower part of the TM7 bundle and in the intracellular regions (ICLs and cytoplasmic C-terminal domain) and interact with G proteins and β-arrestins can change significantly and specifically.

Despite the fact that the data obtained mainly relate to class C GPCRs, di- and oligomerization affects the functional activity of other classes of GPCRs, including the most representative class A [157], although the data in this case are not always so unambiguous. There are a number of recent studies showing that heterodimerization of class A receptors affects their binding characteristics and the efficiency of agonist activation [158,159,160,161] and intracellular signaling bias [159] and also modulates physiological responses under the conditions of agonist-induced stimulation of GPCRs [162,163]. Homodimerization, as shown for the type 1 angiotensin II receptor, can also affect the functional activity of the class A GPCRs [164]. Like class C, in most cases the effect of dimerization on the activity of class A GPCRs is due to allosteric mechanisms.

Along with this, there are alternative data that signal that transduction through class A GPCRs does not require the formation of a di(oligo)heteromeric complex between their protomers [165]. In this case, the functional interaction between uncomplexed protomers belonging to different GPCR types is due to the influence of one protomer on the internalization, traffic, and GRK-mediated phosphorylation of another protomer [166,167,168]. In addition, competition between protomers for binding to β-arrestins can make a certain contribution [169]. All this must be taken into account when evaluating the contribution of the formation of receptor complexes to the allosteric regulation of the GPCR-signaling. In any case, it is necessary to differentiate the mechanisms of allosteric regulation that take place in GPCR complexes in monomeric forms of GPCRs, as well as in the case of dynamic monomeric–dimeric equilibrium among GPCR protomers [165,170].

## 6. Allosteric Sites in Different Families of GPCRs

All of the above indicates a variety of allosteric effects on GPCR signaling, which are realized with the involvement of a large number of allosteric sites located in different domains and subdomains of the receptors. The following sections will discuss the localization of allosteric sites, the mechanisms of allosteric regulation, and some endogenous and synthetic allosteric regulators for several groups of receptors, including various types of chemokine receptors, PARs, the receptors of thyroid-stimulating (TSHR) and luteinizing hormones (LHR), and β-AR. The choice of these receptors is due to rather extensive information about their allosteric sites and the diversity of their endogenous and synthetic allosteric regulators, including autoantibodies and synthetic pepducins. At the same time, for a number of other GPCRs, there is also a lot of data on allosteric regulation and their analysis is presented in a number of recent comprehensive reviews and analytical articles: for muscarinic acetylcholine receptors [171,172,173,174], for metabotropic glutamate receptors [171,175,176], for 5-hydroxytryptamine (serotonin) receptors [177,178], for dopamine receptors [177,179], for opioid receptors [178,180,181], for cannabinoid receptors [176,182,183,184], for adenosine receptors [185], for neuropeptide Y receptors [186], for melanocortin receptors [186], for angiotensin receptors [187], for glucagon-like peptide-1 receptors [176,188], and for free fatty acid receptors [58].

## 7. Allosteric Regulators of Chemokine Receptors

A feature of chemokine receptors is a large number of specific endogenous ligands and many signaling pathways regulated through them. Thus, the CCR1 chemokine receptor is a target for at least 11 chemokines, some of which activate predominantly G proteins, while others activate β-arrestins [189]. Accordingly, allosteric modulators and regulators are very important for biased agonism in the case of chemokine-dependent signaling cascades [190].

In terms of the development of allosteric regulators, the chemokine receptors CXCR1, CXCR2, CXCR3, CXCR4, CCR1, CCR2, CCR5, CCR7, and CCR9 are the most studied [190,191]. Structurally related CXCR1 and CXCR2 receptors are activated by chemokines of the CXCR1/2 family, which causes increased proliferation of endothelial cells, stimulates their migration, and makes a significant contribution to survival, proliferation, and angiogenesis of vascular endothelial cells [192]. CXCR4 and its endogenous ligand, stromal cell-derived factor-1 (CXCL12/SDF-1), are involved in the regulation of leukocyte transport, B-cell lymphopoiesis, bone marrow myelopoiesis, and are involved in the regulation of survival and proliferation of hematopoietic stem cells [193]. CXCR4 is highly expressed in many tumors and also stimulate tumor angiogenesis and metastasis [194]. CXCR3, which is expressed on the surface of activated T cells, B cells, and natural killer cells, and its ligands (CXCL9, CXCL10, and CXCL11) have been implicated in the development of infectious, autoimmune, and neoplastic diseases [195]. CCR2, CCR7, and CCR9 are involved in the immune response and pathogenesis of numerous inflammatory diseases [196,197,198], as well as in tumor metastasis [194,199]. CCR5 functions as a co-receptor upon HIV entry into the cell making it a valuable target for HIV therapy [200] and is also involved in acute graft-versus-host disease after allogeneic hematopoietic cell transplantation [201].

Some of the currently developed allosteric regulators of chemokine receptors bind to sites located inside the transmembrane tunnel, but most of them, both of low-molecular-weight and peptide nature (pepducins), interact with allosteric sites located in ICLs and their interfaces with TMs (Figure 2). The pharmacological profile of allosteric agonists varies greatly in terms of receptor activation by different endogenous agonists, as well as in the signaling cascades they regulate, indicating their inherent biased allosterism [202]. As noted above, Na^+^ and ions of divalent metals binding to the receptor transmembrane allosteric site function as their allosteric modulators. As a result, the compounds that form a complex with them can significantly affect GPCR activity. Chelators, such as phenanthroline and bipyridine complexed with zinc or copper cations, when penetrated within the transmembrane allosteric site of the CCR1 receptor enhanced specific binding of chemokine CCL3 to the receptor acting as a PAM. At the same time, they inhibited the binding of another chemokine, CCL5, to the receptor acting as a NAM, which indicates their biased allosterism [203].

One of the first allosteric regulators of chemokine receptors used in medicine was maraviroc developed in 2007. It has demonstrated NAM activity for CCR5 and is being used to treat HIV patients [204]. The anti-HIV activity of maraviroc is due to its ability to prevent the entry of the virus into the cell by inhibiting CCR5 [205]. Maraviroc penetrates the transmembrane tunnel of CCR5, where it binds to an allosteric site located just below the outer vestibule, which is formed by amino acid residues from all TMs, except TM4 [205,206].

Of considerable interest among the allosteric regulators of chemokine receptors are reparixin, ladarixin (DF 2156A), and a number of their functional analogs that interact with the transmembrane allosteric site of the CXCR1 and CXCR2 (Figure 2). This site is formed by TM1, TM3, TM6, and TM7. Currently, reparixin and its analogues are considered as drugs that can reduce or prevent inflammatory and autoimmune processes, as well as suppress the development and metastasis of tumors [202,207]. Ladarixin, a potent allosteric inhibitor of CXCR1 and CXCR2, is highly effective (IC_50_ of about 0.1 nM) in preventing damage to pancreatic islets caused by proinflammatory factors and autoimmune reactions indicating its potential antidiabetic activity [208,209]. Ladarixin binds to an allosteric site located in the upper portion of the transmembrane tunnel, and the key residues interacting with it are Lys^126^ TM3, Asp^293^ TM7, and Arg^289^ in ECL2 [210]. Reparixin, a non-competitive allosteric inhibitor of CXCR1 and CXCR2, suppresses the stimulatory effects of interleukin-8 (IL-8), to the greatest extent in the case of CXCR1, but does not affect their affinity for this agonist [211,212]. Reparixin has now been shown to be clinically successful as an anti-inflammatory drug, including in the treatment of patients with severe pneumonia caused by COVID-19, and reparixin treatment has not been associated with an increased risk of secondary infections [207].

Studies of the CXCR3 receptor made it possible to identify two partially overlapping allosteric sites, one of which was located in the outer vestibule of the transmembrane tunnel and the other was located deep in this tunnel in its middle [213]. For the first allosteric site the biased allosteric agonist FAUC1036 and for the second site the nonselective NAM RAMX3 were developed [214]. Later based on the RAMX3 structure, the biased NAMs, 8-azaquinazolinone derivative 1b [215], and the compounds BD064 and BD103 were created [216]. It is assumed that the recently developed compound DF2755A with the activity of a noncompetitive allosteric antagonist of CXCR1 and CXCR2 interacts with the same site [217]. DF2755A suppressed the stimulatory effects of orthosteric agonists of these receptors on neutrophil chemotaxis and under the in vivo conditions reduced inflammatory and post-operative pain [217,218].

The allosteric inhibitor plerixafor (AMD3100), which interacts with an allosteric site located in the outer vestibule of the transmembrane tunnel of CXCR4, suppresses the stimulatory effects of the CXCL12 chemokine on the activity of this receptor but does not affect its activation by the CXCL12 chemokine fragment, which also has the activity of a full CXCR4 agonist but differs from the full-length chemokine in the receptor-binding site [219]. Plerixafor is not selective for intracellular signaling cascades, inhibiting both G protein-regulated signaling and β-arrestin-mediated endocytosis, and the consequence of this is an increase in the expression of CXCR4 and the development of tolerance to this drug during its long-term use [220], which limits its clinical use for mobilization of stem cells from the bone marrow during transplantation. As an alternative, a peptide allosteric antagonist X4-2-6 was developed, structurally corresponding to TM2 and ECL1 of CXCR4, which formed a ternary complex with the receptor and the chemokine CXCL12 [221]. Being in this complex, CXCL12 remained capable of stimulating G protein-dependent pathways but did not affect the recruitment of β-arrestins, as a result of which there was no compensatory accumulation of CXCR4 on the surface of target cells and no tolerance was developed to the peptide antagonist. Simultaneously with the peptide antagonist X4-2-6, the biased small allosteric antagonists T140 and its more stable and less toxic analog TN14003 were developed, which, in the case of inhibition of β-arrestin-mediated pathways, had IC_50_ values 100 times higher than G_i/o_-protein-dependent pathways [222]. Compound TN14003 showed high activity in preventing degeneration of articular cartilage in osteoarthritis in guinea pigs [223]. It is important that in CXCR4 both small allosteric antagonists and the X4-2-6 peptide bind to similarly located allosteric sites [221,222].

Based on a comparative analysis of the structures of the chemoattractant receptor C5aR and the chemokine receptors CXCR1 and CXCR2, as well as taking into account the data of site-directed mutagenesis, an allosteric site was identified in the region of the external entry into the TMD of C5aR, which did not coincide with its orthosteric site. This allowed the development of DF2593A, which reduced agonist-stimulated C5aR activity and suppressed pain syndromes caused by acute and chronic inflammation [224].

Currently, a number of small allosteric GPCR antagonists have been developed, which, like pepducins, interact with intracellular allosteric sites and prevent the effective interaction of receptors with G proteins and (or) β-arrestins (Figure 2). Among the chemokine receptors, such antagonists have been found for three of their types, such as CCR2, CCR7, and CCR9. Among the allosteric CCR9 antagonists, the most interesting are vercirnon, which has a very high selectivity for binding to CCR9 [65], and its analogs (including 4-aminopyrimidine derivatives), which may be useful in the treatment of Crohn’s disease [225]. They interact with the intracellular allosteric site of CCR9, which is formed by the cytoplasmic ends of TM1, TM2, TM3, and TM6, and also includes both charged and hydrophobic residues localized in TM7, H8, and ICL1 [225]. A similar environment has intracellular allosteric sites in the CCR2 and CCR7 receptors, which are similar in their architecture to CCR9. This is supported by data on the 3D structure of complexes of CCR2 and CCR7 with small allosteric antagonists CCR2-RA-[R] [66] and Cmp2105 [226], respectively. In all three types of receptors, the main molecular determinants responsible for the binding of allosteric inhibitors are a tyrosine residue located at the cytoplasmic end of TM7 and a relatively variable Gly-Glu/Val-Lys/Arg-Phe-XX-Asp/Tyr sequence located in the H8 helix. The hydroxypyrrolinone group of CCR2-RA-[R] (CCR2), the thiadiazole-dioxide group of Cmp2105 (CCR7), and the sulfonamide group of vercirnon (CCR9) interact with this sequence [226]. Navarixin, which is similar in pharmacological profile and efficacy to Cmp2105 and is an antagonist of CXCR1, CXCR2, and CCR7 [227], also interacts with the allosteric site formed by TM7 and H8. However, in this case, the cyclobutene-dione group of navarixin is less suitable, which reduces the effectiveness of the interaction with CCR7, although it does not prevent the high efficiency of navarixin as an inhibitor of CXCR1 and CXCR2, which have a structurally different intracellular allosteric site [226]. The occupancy of intracellular allosteric sites of chemokine receptors by the antagonists prevents their interaction with the C-terminal segment of the Gα_i/o_-subunit and β-arrestin, since these sites, although to varying degrees, overlap with G protein and β-arrestin-binding surfaces of the receptors [228,229]. The consequence of this is the stabilization of the inactive state of the receptor and its low affinity for the agonist, since, as noted above, the formation of a complex between the GPCR and the inactive forms of G protein or β-arrestin is a necessary condition for the transition of the receptor to a high-affinity state with its further activation by the agonist. 

### 7.1. Pepducins as Regulators of Chemokine Receptors

Of considerable interest are pepducins, derivatives of ICLs of chemokine receptors, which, interacting with cytoplasmically oriented allosteric sites, modulate the activity of receptors and their coupling with G proteins and β-arrestins [230,231,232,233,234] (Figure 2). Given their ability for multicenter binding, their regulatory effects are much broader and more diverse than those of small ligands of intracellular allosteric sites.

*N*-palmitoylated peptide Palm-RTLFKAHMGQKHRAMR (X1/2pal-i3), corresponding to the ICL3 of CXCR1 and CXCR2, demonstrated the properties of NAM for CXCR1/CXCR2, inhibiting endothelial cell proliferation induced by endogenous CXCR1/CXCR2 ligands, IL-8 and the chemokine CXCL1/GRO, and also reduced the proliferation of human umbilical vein endothelial cells induced by metalloproteinase MMP1, a stimulator of chemokine secretion with angiogenic activity. Pepducin X1/2pal-i3 and peptide YSRVGRSVTD (X1/2LCA-i1) modified with lithocholic acid at the N-terminus and structurally corresponding to the ICL1 of CXCR1 and CXCR2 inhibited IL-8- and CXCL1/GRO-induced vascular tube formation, and their efficacy was comparable to the anticancer drug bevacizumab. Three-week treatment of mice with X1/2pal-i3 decreased the intensity of angiogenesis by five times and significantly reduced tumor growth in ovarian cancer xenografts [192]. A truncated analog of pepducin X1/2pal-i3, Palm-RTLFKAHMGQKHR, reduced the spontaneous formation of intestinal adenomas in Apc^−/+^ mice [235].

*N*-palmitoylated peptides PZ-218 (Palm-MGYQKKLRSMTD) and PZ-210 (Palm-SKLSHSKGHQKRKALK), corresponding to the ICL1 (63–74) and ICL3 (224–239) of CXCR4 inhibited CXCL12/SDF-1-induced responses in human neutrophils and mice and also blocked CXCL12/SDF-1-mediated migration of lymphoma and lymphocytic leukemia cells while palmitate-free analogs were inactive [231,236]. Both CXCR4 pepducins, including those in combination with the anticancer drug rituximab, suppressed the survival and metastasis of disseminated lymphoma xenografts. The effect of rituximab was blocked by treatment with the chemokine CXCL12/SDF-1, while this did not occur with the combined use of pepducins and rituximab indicating synergism of their inhibitory effect on CXCR4 and the different localization of binding sites [237].

Along with pepducins with NAM activity for CXCR4, CXCR4-derived pepducins with the activity of an allosteric agonist and/or PAM have been developed, including pepducin ATI-2341 corresponding to the ICL1 of CXCR4 [230]. Pepducin ATI-2341 regulates chemotaxis in various cell cultures and is involved in the mobilization and activation of polymorphonuclear neutrophils and hematopoietic stem cells [230,238]. Unlike CXCL12/SDF-1 used in the clinic, which promotes the mobilization of not only polymorphonuclear neutrophils and hematopoietic stem cells but also lymphocytes, ATI-2341 does not have a noticeable effect on the mobilization of lymphocytes, which indicates its selectivity. This is due to the fact that the targets of ATI-2341 are various isoforms of G_i/o_-proteins, while CXCL12/SDF-1, along with G_i/o_-proteins, stimulate β-arrestins and G_12/13_-proteins. ATI-2341 does not induce the GRK-mediated phosphorylation of CXCR4, which prevents its desensitization and maintains cell sensitivity to CXCR4-agonists [233]. In addition, ATI2341 protects brain endothelial cells from radiation damage, which indicates the prospect of its use for the repair of tissues damaged during chemotherapy or radiation therapy through increasing the survival of vascular endothelial cells [239].

These data suggest that the ICL3 and ICL1 in the CXCR1, CXCR2 and CXCR4 receptors are either themselves involved in the formation of cytoplasmically oriented allosteric sites or interact with complementary receptor regions that form such sites [231,237]. Moreover, as in the case of small allosteric antagonists [65,66,235,236], the pepducin-induced decrease in the accessibility of the vestibule leading to a transmembrane tunnel of chemokine receptors on the cytoplasmic side, or a change in its conformation, can prevent the formation of the GPCR–G protein/β-arrestin complex and thereby inhibit signal transduction.

### 7.2. Autoantibodies to Chemokine Receptors CXCR3 and CXCR4

It is generally accepted that GPCR autoantibodies are produced against the extracellular regions of the receptor. Depending on the epitope to which they specifically bind and on their type (monovalent or bivalent), autoantibodies are able to either stimulate or suppress the basal or agonist-stimulated activity of a recognized receptor and thereby be involved in the pathogenesis of various diseases [100,240,241]. In this regard, it is interesting that in the case of CXCR3 and CXCR4 patterns of autoantibodies have been identified that are specific not only to extracellular regions but also to their ICLs. It has been shown that in the blood of patients with systemic sclerosis, a severe autoimmune disease, autoantibodies to ICL2 and ICL3 and the membrane-proximal region of the cytoplasmic C-terminal domain of CXCR3 predominate, while in healthy subjects the proportion of autoantibodies to extracellular regions of this receptor was significantly higher [242]. Importantly, autoantibodies against the extracellular N-terminal region of CXCR3 were associated with a better prognosis for systemic sclerosis [243]. This indicates a significant therapeutic potential of autoantibodies to extracellular regions of CXCR3 and CXCR4 but requires further study of the allosteric regulation of these receptors.

## 8. Allosteric Regulators of Proteinase-Activated Receptors

A significant progress has now been made in the development of allosteric regulators of the PAR family of GPCRs. This family includes four types of PARs (PAR1, PAR2, PAR3, and PAR4), which are critical for the control of hemostasis and platelet function and are involved in the regulation of embryonic development, inflammation, wound healing, as well as tumorigenesis and metastasis [244,245,246,247]. Thrombin-induced platelet activation is enhanced in the case of erosion and rupture of atherosclerotic plaques and in percutaneous coronary intervention causing arterial thrombosis, the main cause of myocardial infarction and ischemic stroke. Thrombin exerts its effects on platelets through PAR1 and PAR4, coupled to different types of G proteins (G_i/o_, G_q/11_, and G_12/13_), as well as through β-arrestin pathways, and the activating effect of thrombin on platelet aggregation is realized mainly through G_q/11_-proteins [245,247,248]. After the G_q/11_-mediated activation of PLCβ in platelets, the level of intracellular calcium increases and diacylglycerol-sensitive isoforms of protein kinase C are activated, which leads to activation of αIIb/β3-integrins, their translocation to the plasma membrane, and increased platelet aggregation. Along with this, the release of ADP, an agonist of P2Y12-purinergic receptors, from platelets is enhanced, which stimulates the synthesis of thromboxane A2, a mediator of platelet aggregation and degranulation. It has been established that PAR1 expression is increased in tumor cells, and this is accompanied by an increase in the activity of matrix metalloproteinase-1 (MMP1), which functions as a PAR1 agonist [249,250]. Accordingly, PAR1 agonists enhance oncogenesis, invasion, and metastasis, while PAR1 antagonists, on the contrary, suppress them, preventing the tumor progression and improving the cancer prognosis [249,251,252]. Agonist-induced activation of PAR1 has a strong anti-inflammatory effect and protects endothelial cells from destruction during sepsis [253]. However, prolonged PAR1 activation may have the opposite effect, impairing endothelial cell function and exacerbating sepsis [254,255].

Regulatory effects mediated through PAR1 and PAR4 are determined by the ability of these receptors to form heterodimeric complexes with PAR2 and PAR3 [245,256]. In the case of PAR1, the formation of a heterodimeric complex with PAR2 provides PAR1-induced activation of PAR2 and stimulation of thrombin-mediated β-arrestin signaling, which significantly contributes to PAR-dependent effects on blood coagulation and inflammation [257]. As a result, allosteric regulators of the PARs with antagonistic or NAM activity can be used to prevent thrombosis, to attenuate endothelial dysfunctions in sepsis, and also as anticancer drugs. In turn, allosteric agonists and PAMs may be useful for increasing cell survival and wound healing [247,258,259].

### 8.1. Small Allosteric Ligands of PARs

A relatively large group of PAR1 allosteric regulators are parmodulins, which are small molecules that act by binding to a cytoplasmically oriented allosteric site [260,261,262,263] (Figure 3). Parmodulins have a biased activity because they act on some types of PAR1-coupled G proteins but have little effect on the activity of other types of G proteins. They have been shown to inhibit G_q/11_-mediated granule secretion caused by the mobilization of calcium ions from intracellular depots but do not affect G_12/13_-mediated signaling pathways, thereby normalizing platelet activity but not affecting platelet shape. Unlike the orthosteric PAR1 agonist, parmodulins inhibit PAR1 activity moderately and reversibly, which is consistent with their pharmacological profile as NAMs [260,262,263]. All this indicates the prospects for the development of parmodulin-based antithrombotic agents [260,262]. In animal models, the therapeutic effects of parmodulins are due to the attenuation of inflammatory reactions and their protective effect on endothelial cells [262,264]. Parmodulin ML161 (PM2) significantly reduced the size of myocardial infarction in mice with myocardial ischemia–reperfusion injury [265].

The study of the binding of parmodulins to PAR1 made it possible to identify the segments involved in the formation of the intracellular allosteric site that interacts with these regulators. It includes the cytoplasmic portion of the TM7 bundle, ICL1, and the H8 helix since, as was shown for parmodulin JF5, palmitoylation of the C-terminal domain of the receptor, which stabilizes the H8 helix, is critical for JF5-induced activity [260]. Accordingly, localization of the allosteric site in the area of contact between PAR1 and transducer proteins (G proteins and β arrestins) can mediate selective effects of parmodulins in relation to certain signaling cascades.

Along with allosteric modulators of PAR1, the compound GB83 with PAM activity was recently developed, which potentiated the stimulatory effects of thrombin and peptidic PAR1-agonists on PAR1 activity but did not affect the basal activity of PAR1 [266]. Using molecular docking, GB83 was shown to bind to a site located near or overlapping with a transmembrane orthosteric site to which the PAR1-antagonist vorapaxar specifically binds (Figure 3). GB83 significantly enhanced PAR1-mediated cell survival and migration and accelerated skin wound healing in the in vivo experiments [266].

Among the small allosteric modulators of PAR2, the compound AZ3451 has been the most studied [267,268,269]. It binds to the allosteric site of PAR2 formed by TM2, TM3, and TM4 and at nanomolar concentrations suppresses the stimulation of G_q/11_-proteins, ERK1/2, and β-arrestin2-mediated signaling induced by PAR2-agonists, including the small orthosteric agonist AZ2429 [269]. Since the allosteric site, the target for AZ3451, overlaps with the orthosteric site with which the tethered ligand interacts, the conformation of this site should change significantly after activation of the receptor, which makes a significant contribution to PAR-mediated signal transduction. In the absence of PAR2 activation by the agonist, AZ3451 had no effect on the basal activity of the receptor, which allows it to be considered as a NAM [269]. A similar pattern of activity is demonstrated by heterobivalent PAR1/PAR2 ligands, which are constructed based on PAR1- (RWJ-58259) and PAR2-antagonists (imidazopyridazine derivatives) cross-linked with polyethylene glycol [270]. Such constructs should affect the conformation of the PAR1 and PAR2 allosteric sites, which leads to the inhibition of G_q/11_-mediated calcium mobilization in endothelial and cancer cells induced by PAR1- and PAR2-agonists. A possible range of therapeutic applications for AZ3451 include the treatment of postoperative osteoarthritis as indicated by its ability to prevent postoperative cartilage degradation in rats [268].

Recently, a non-competitive allosteric inhibitor of I-191, an imidazopyridazine derivative, has been developed that suppresses PAR2 activation by different agonists (trypsin, peptide, and non-peptidic PAR2-agonists) [271]. Compound I-191 reduced agonist-stimulated PAR2-effects, including G_q/11_-mediated Ca^2+^ release, G_i_-mediated inhibition of forskolin-stimulated AC, stimulation of ERK1/2 activity, and activation of small G proteins of the Ras family (RhoA) [271]. Inhibiting the effects of PAR2-agonists at nanomolar concentrations, I-191 showed no activity in their absence, which allows us to classify it as a PAR2-specific NAM. Currently, several weaker PAR2-antagonists have been developed, including those selective for certain intracellular cascades, which are active at micromolar concentrations, which is typical for allosteric regulators and are potentially capable of binding to the transmembrane allosteric site of PAR2. However, no convincing evidence has yet been obtained to attribute them to NAMs [267,272,273].

### 8.2. Pepducins as Allosteric Regulators of PARs

The greatest success in the creation of allosteric regulators of PAR1, PAR2, and PAR4 is due to the development of pepducins, which are lipidated derivatives of their ICLs [246,274,275,276,277,278,279,280] (Figure 3).

Pepducin Palm-ATGAPRLPST (P4pal-i1), corresponding to the ICL1 of PAR4, inhibited the signal transduction generated by thrombin through its cognate receptor [275]. P4pal-i1 blocked PAR4-mediated chemotaxis and prevented platelet aggregation induced by the PAR4 agonist AYPGKF-amide without affecting PAR1-mediated effects. The combined use of P4pal-i1 and the PAR1 antagonist RWJ-56110 led to the complete disappearance of the effects of thrombin indicating the effectiveness of simultaneous blockade of PAR1 and PAR4 [275]. The combined use of P4pal-i1 and the thrombolytic bivalirudin, which specifically binds to thrombin, completely prevented platelet aggregation induced by a submaximal concentration of α-thrombin [275,281]. P4pal-i1 also prevented the development of arterial thrombosis in guinea pigs with carotid artery occlusion, and the combined use of P4pal-i1 and bivalirudin was more effective than monotherapy [275]. Palm-SGRRYGHALR (P4pal-10), corresponding to PAR4 ICL3, was less specific than P4pal-i1 and had both PAR4-antagonist and inverse PAR1-agonist activity [274,276]. It reduced platelet aggregation induced by the PAR4-agonist AYPGKF-amide by 45–70% but had little effect on platelet aggregation induced by the PAR1 agonist SFLLRN-amide [274].

At micromolar concentrations, pepducin Palm-RCLSSSAVANRS (P1pal-12) and its truncated analog P1pal-7, corresponding to PAR1 ICL3, inhibited PAR1 agonist-induced PLCβ stimulation [274,282,283]. Pepducin P1pal-12 also inhibited rat aortic vasorelaxation induced by the PAR1-agonist TFLLR-amide without affecting the corresponding effect of the PAR2-agonist [284]. PZ-128, which was developed from the structure of P1pal-7, inhibited PAR1-mediated platelet aggregation and arterial thrombosis in guinea pigs and monkeys, and its effect was enhanced in the presence of the antiplatelet drug clopidogrel [278]. Further studies have shown that PZ-128 has no side effects both in experimental animals and in patients with ischemic heart disease or with multiple risk factors for coronary artery disease [246,278,285,286]. PZ-128 had a dose-dependent inhibitory effect on platelet aggregation stimulated by the PAR1-agonist SFLLRN-amide but did not affect the corresponding effects of PAR4 agonists. PZ-128 did not affect bleeding, coagulation, biochemical and metabolic parameters, and electrocardiogram parameters and the drug was well tolerated when administered intravenously at a dose of 0.5 mg/kg. Currently, PZ-128 has successfully completed clinical trials (phase 2) in patients with cardiovascular disease demonstrating safety and efficacy in the suppression of myonecrosis and arterial thrombosis [280].

The longer pepducin Pal-RCLSSSAVANRSKKSRALF (P1pal-19), in contrast to P1pal-12 and P1pal-7, demonstrated full PAR1 agonist activity and stimulated G_q/11_- and G_i/o_-proteins similar to selective PAR1 agonists [274,284]. This indicates that the modification of the structure of PAR-pepducins not only affects the efficiency and selectivity of their action but also changes their pharmacological profile.

Significant progress has been made in the development of PAR1-pepducins with antitumor activity [246,279]. The first of these was P1pal-7 (PZ-128), which inhibited tumor growth, suppressed metastasis, and reduced cell survival in breast, ovarian, and lung cancers [246,249,250,278,285,287]. P1pal-7 and its extended analog P1pal-12 suppressed the PAR1-dependent migration of OVCAR-4, IGROV-1, and SKOV-3 ovarian carcinoma cells into ascitic fluid obtained from patients with ovarian cancer [249]. P1pal-7 enhanced the inhibitory effect of docetaxel, an anticancer drug, on tumor angiogenesis in ovarian cancer [249]. The antitumor effect of P1pal-7 was based on the inhibition of Akt kinase, a key antiapoptotic protein [287]. P1pal-7 was also effective in the treatment of lung cancer, reducing lung tumor growth by 75%, and in this it was comparable to bevacizumab (Avastin), a potent inhibitor of tumor angiogenesis [288]. In this case, the antitumor effect of P1pal-7 was due to the inhibition of PAR1-mediated activation of ERK1/2, an effector component of the MAPK cascade. P1pal-7 also suppressed the growth and invasive activity of Lewis lung carcinoma cells [250].

Pepducin Palm-RSLSSSAVANRS (P1pal-12S), with allosteric PAR1 antagonist activity, had an anti-inflammatory effect in a mouse model of bacterial peritonitis, significantly increasing animal survival. However, its injection 4 h after the induction of peritonitis did not affect the mortality of mice. Palm-AVANRSKKSRALF (P1pal-13), which is also a derivative of PAR1 ICL3, with the activity of an allosteric PAR1-agonist, was not effective when administered immediately after inflammation induction but increased the survival rate of mice when administered 4 h later under the conditions of developed sepsis. In *Par1* knockout mice, none of the studied pepducins affected sepsis [253]. These data indicate that at the early and late stages of sepsis, the optimization of the use of PAR1-pepducins with agonistic and antagonistic activity can lead to a pronounced therapeutic effect. P1pal-13 also had a proangiogenic effect in mice with carotid ligation [289]. Its effect required the presence of both PAR1 and PAR2, which indicates that the target of pepducin is the PAR1–PAR2 complex. This complex is also a target for Palm-RSSAMDENSEKKRKSAIK^270−287^ (P2pal-18S), an ICL3 derivative of PAR2 that inhibits the migration of neutrophils and cholangiocarcinoma cells stimulated by both trypsin (PAR1-agonist) and selective PAR2-agonists [289,290].

Pepducin PZ-235, corresponding to PAR2 ICL3, with allosteric antagonist activity, showed a potent anti-inflammatory effect in the chronic inflammation and liver fibrosis model [291] and the atopic dermatitis model [292]. Treatment of mice with CCl_4_-induced liver fibrosis with PZ-235 suppressed the manifestations of fibrosis, reduced collagen deposits, reduced levels of pro-inflammatory cytokines and other biochemical markers of non-alcoholic fatty liver disease, attenuated the severity of steatosis, reduced triglyceride levels and transaminase activity, and inhibited stellate cell proliferation by 50–100% [291]. PZ-235 had a protective effect in the case of hepatocellular necrosis by attenuating the PAR2-mediated production of reactive oxygen species in hepatocytes. When interacting with the PAR2 allosteric site, PZ-235 forms a well-structured amphipathic α-helix, structurally similar to the corresponding ICL3 region and the neighboring TM6 segment of this receptor [291]. In the in vitro experiments, PZ-235 significantly reduced PAR2-mediated expression of inflammatory factors, including nuclear factor-κB (NF-κB), thymic stromal lymphopoietin (TSLP), tumor necrosis factor-α (TNF-α) and differentiation marker K10 in human keratinocytes, as well as led to a decrease in the expression of IL-4 and IL-13 in human mast cells [292]. In mice with oxazolone- and DNFB-induced dermatitis, PZ-235 reduced skin thickening (by 43–100%) and leukocyte crusting (by 57%) and also inhibited ex vivo leukocyte chemotaxis to PAR2-agonists [292]. PZ-235 treatment of filaggrin-deficient mice exposed to dust mite allergens for 8 weeks reduced total leukocyte and T-cell infiltration, epidermal thickness, and incidence of skin thickening, scaling, excoriation, and total lesion severity score [292].

In some cases, PAR-derived pepducins are able to influence the activity of receptors that are not homologous to PARs, as was shown for pepducin P4Pal_10_ corresponding to the ICL3 of PAR4 [293]. This pepducin not only functions as an antagonist and NAM for PAR4 but is also able to modulate the functional activity of a number of other receptors that do not belong to the PAR family [293,294]. P4Pal_10_ reduces the stimulation of G_i/o_-proteins via the type 2 formyl peptide receptor (FPR2) and is a weak agonist for the type 2 short chain free fatty acid receptor (FFAR2) [293]. Therefore, structural similarity with the ICLs of the target receptor is not a necessary condition for the effect of pepducin on it. Possible explanations for the “non-specificity” of the PAR4-derived pepducin P4Pal_10_ on other GPCRs may be the existence of a common configuration of allosteric sites even in unrelated receptors. At the same time, which is more likely, pepducin can interact with heterooligomeric complexes, which, in addition to PAR4, also include non-homologous receptors (FPR2 and FFAR2).

### 8.3. Autoantibodies to PAR1

Endogenous allosteric regulators of PAR1 are autoantibodies produced against its extracellular domains [295,296,297]. A significant proportion of patients with systemic sclerosis have autoantibodies to extracellular regions of PAR1, which are able to activate the receptor, such as thrombin, thus acting as allosteric agonists [296]. Incubation of human dermal microvascular endothelial cells (HMECs) with IgG isolated from the blood of patients with systemic sclerosis resulted in an increased release of IL-6 and phosphorylation of a number of protein kinases (Akt, p70S6K, and ERK1/2), the targets of PAR1-agonists, and these effects of autoantibodies were specifically blocked by PAR1 antagonists [296]. Interestingly, patients with severe forms of COVID-19 also have high titers of anti-PAR1 autoantibodies that stimulate PAR1 in endothelial cells and platelets, and thus contribute to the pathogenesis of microthrombosis. A positive correlation was shown between the level of autoantibodies to PAR1 and the content of D-dimer, a marker of thrombosis [297].

## 9. Allosteric Regulators of Pituitary Glycoprotein Hormone Receptors

The family of pituitary glycoprotein hormone receptors includes the receptors of thyroid-stimulating hormone (TSH), luteinizing hormone (LH), and follicle-stimulating hormone (FSH) [298,299]. A feature of this family is a significant extracellular domain (ectodomain), where the orthosteric site is located to which αβ-heterodimeric hormones containing the same α-subunit and variable β-subunits bind with high affinity.

### 9.1. Thyroid-Stimulating Hormone Receptor

The TSH receptor (TSHR) is localized predominantly on the surface of thyrocytes but is sometimes found on the surface of adipocytes, fibroblasts, and other cells [300]. In the active state, the TSHR forms a dimeric or oligomeric complex [301,302,303]. The binding of TSHR to TSH stimulates the synthesis and release of thyroid hormones by thyrocytes and also regulates the growth and differentiation of these cells. In TSHR, the ectodomain contains a subdomain with 11 leucine-rich repeats (LRRs) and a cysteine-rich hinge region that functions as a spacer and is located between the LRR subdomain and TMD [304]. Endogenous regulators of TSHR are TSH, specific autoantibodies to TSHR, and thyrostimulin, which bind to the orthosteric site of TSHR, as well as to allosteric sites formed by the LRR subdomain, the hinge region, and the ECLs [305,306].

The main function of the hinge region is its inhibitory effect on the activity of TSHR, as well as on the specificity of its interaction with TSH and other endogenous regulators [307,308,309,310,311]. A relatively short region of Asp^403^–Asn^406^, located at the C-terminus of the hinge region, is considered as an “intrinsic” allosteric TSHR agonist [309,312,313]. Upon binding to the hormone, the conformation of TSHR changes, which allows this region to interact with the receptor’s ECLs and TMDs and activate G proteins. The peptide containing this region has specific TSHR agonist activity, which is due to its interaction with TSHR ECL1 [308,310,312]. The existence of an internal allosteric agonist is unique for GPCRs and indicates the complex nature of the regulation of the interaction between the ectodomain and the TMD in TSHR [311]. It should be noted that the extracellular N-terminal regions of a number of other GPCRs also function as “intrinsic” allosteric regulators and are involved in biased allosterism, as shown for GPR35 [314].

After the hormone binds to the TSHR ectodomain, the cavity within the TMD remains free, although it is involved in the stabilization of the TSHR–G protein pre-activation complex and in the transmission of conformational changes from the ligand-bound ectodomain to the ICLs [315,316]. Allosteric sites are located in this cavity, and there is every reason to believe that their activity can be regulated by simple ions and (or) lipophilic molecules interacting with TSHR hydrophobic helices. At the same time, these sites are targets for synthetic low-molecular-weight compounds, which, through allosteric mechanisms, can regulate the intrinsic and hormone-stimulated activity of TSHR (Figure 4). At present, significant progress has been made both in the study of the configuration of TSHR transmembrane allosteric sites and in the development of small compounds that can bind to them and demonstrate the activity of allosteric regulators [317,318,319,320,321,322,323,324,325,326].

The first allosteric regulators of TSHR were developed back in the 2000s based on the thieno[2,3-d]-pyrimidine structure with the activity of allosteric agonists of the LH receptor (LHR), including the most active LHR-agonists Org41841 and Org43553 [327,328]. By analyzing their binding to TSHR, the 3D structure of the transmembrane allosteric site of TSHR was proposed [329,330,331], which was similar to that of LHR, and this subsequently allowed the development of a large number of its small ligands with different pharmacological profiles [318,319,320,321,332,333,334,335]. It was found that this site is formed by amino acid residues located in TM3, TM4, TM5, TM6, and TM7 and is covered from above by ECL2 segments. Unlike LHR, the entrance to the allosteric site of TSHR is narrower and is formed mainly by hydrophobic and large amino acid residues. These residues are located at the interfaces between ECL2 and neighboring TM4 and TM5 segments, as well as at the interface formed by ECL3 and TM6 [331].

Among the large number of full allosteric TSHR agonists, the compound NCGC00168126-01 and its analogue NCGC00165237-01 were characterized by high specific activity [318,336]. Under the in vitro conditions, they stimulated the AC activity in cells with expressed TSHR and increased the expression of thyroglobulin and other TSH-dependent genes in human thyrocytes, and their effects were comparable to those of TSH [318,336]. When administered to mice, NCGC00165237–01 increased the thyroxine level and increased the uptake of radioactive iodine by the thyroid gland, which was due to an increase in the expression of thyroglobulin, a precursor of thyroid hormones, and thyroperoxidase, which catalyzes the binding of iodine to tyrosine residues of thyroglobulin [336]. Based on the structure of thieno[2,3-d]-pyrimidine, we developed the compound TPY3m, which stimulated the production of thyroxine when it was administered intraperitoneally to rats and also stimulated the expression of thyroid-specific genes in a cell culture of thyrocytes FRTL-5 and in the thyroid of rats [337]. Quite unexpectedly, TPY3m did not cause a decrease in the expression of the *Tshr* gene, and when combined with TSH (cultured FRTL-5 cells) or thyroliberin (rat thyroid), it partially restored it. This may be one of the mechanisms for maintaining the stimulating effects of TSH (in vitro, FRTL-5) and thyroliberin (in vivo, rats) on the production of thyroid hormones under conditions of their combined use with TPY3m [337].

Small agonists are able to activate TSHR mutants that are insensitive to TSH [338]. In a culture of HEK-EM293 cells with expressed mutant TSHR (Cys^41^Ser or Leu^252^Pro substitutions in the ectodomain), the low-molecular-weight agonist C2 increased the intracellular cAMP level while TSH was inactive. At the same time, the Leu^467^Pro substitution in the allosteric site, which prevented TSHR binding to C2, blocked its effect on AC activity [338]. Thus, allosteric TSHR agonists may be effective in subclinical hypothyroidism associated with thyroid resistance to TSH due to mutations in the TSHR ectodomain.

Since TSHR is functionally coupled to several types of G proteins, including G_i/o_-proteins, the main donors of the Gβγ-dimer, an important task was to create allosteric TSHR agonists biased against certain types of G proteins. In 2015, Rauf Latif and colleagues made the first attempt to create biased agonists and developed compounds MS437 and MS438, which at nanomolar concentrations were able to activate G_s_-, G_q/11_- and G_12_-proteins but did not affect G_i_-proteins [339]. Moreover, MS438 showed high efficiency in relation to the activation of the G_s_-proteins and AC. Both compounds specifically bound to an allosteric site located in the TMD, but despite a similar regulatory pattern, there were some differences in the interaction with the amino acid residues that form this site. When interacting with TM3, compound MS437 formed close contacts with Thr^501^, while MS438 formed close contacts with Ser^505^ and Glu^506^ [339]. Further studies led to the creation of a biased allosteric MSq1 agonist, which, in CHO cells with expressed TSHR at nanomolar concentrations, stimulated the G_q/11_-protein and increased the activity of calcium-dependent isoforms of protein kinase C with almost no effect on G_s_- and G_12/13_-proteins and the MAPK cascade [340]. Using molecular docking, it was found that MSq1 binds to a site that is formed by TM1, TM2, TM3, and TM7 and is covered from above by segments of the ECL1 and ECL2. Comparison of the binding of the G_q/11_-selective agonist MSq1 with the compound MS438, which preferentially activates the G_s_-protein, revealed significant differences in the pattern of interaction with the amino acid residues that form the allosteric site, which is the “key” to the further development of biased allosteric TSHR agonists [340], including those focused on the biased regulation of β-arrestin pathways [341].

The need to develop PAMs for the regulation of TSHR is due to the prospects for their use to prevent osteoporosis, since TSH plays an important role in the formation of bone tissue preventing its loss and stimulating its formation [342]. The stimulating effect of TSH on osteoblasts and bone tissue formation is realized mainly through β-arrestin1-regulated cascades [343] and, therefore, the search for PAMs for TSHR was carried out among ligands biased for β-arrestins. As a result, the compound NCGC00379308 with PAM activity was developed, which potentiated the TSH-induced translocation of β-arrestin1 to TSHR but did not affect the G_s_- and G_q/11_-dependent cascades demonstrating specificity for β-arrestin signaling [320]. NCGC00379308, when treated with human U2OS cells, did not affect the expression of the TSH-dependent genes *OPN* and *ALPL*, encoding osteopontin and alkaline phosphatase and the secretion of osteopontin but significantly potentiated their stimulation by TSH. Thus, NCGC00379308 can increase the efficiency of endogenous TSH on the growth and differentiation of osteoblast progenitors through β-arrestin-dependent mechanisms [320].

The development of allosteric inverse agonists and neutral antagonists for TSHR is necessary for the treatment of autoimmune hyperthyroidism (Graves’ disease) and associated ophthalmopathy, as well as for the treatment of thyroid cancer caused by activating mutations in TSHR [318,323,326]. In 2008, the compound NIDDK/CEB-52 was developed with the activity of a neutral TSHR antagonist in the phenyl ring of which a methoxypropylene group was introduced to facilitate penetration into the transmembrane allosteric site of the receptor [344]. Incubation of HEK-EM 293 cells with NIDDK/CEB-52 significantly reduced AC stimulation by both TSH and TSHR-stimulating antibodies, and incubation of human thyrocytes with NIDDK/CEB-52 caused a three-fold decrease in TSH-stimulated thyroperoxidase gene expression [344]. Since NIDDK/CEB-52 suppressed LHR activity to a small extent, more selective neutral TSHR antagonists, such as NCGC00242595 and NCGC00242364, were subsequently developed, which in vitro reduced the AC stimulation by TSH and TSHR-stimulating antibodies without affecting the basal AC activity [319,345]. Treatment of mice with NCGC00242364 reduced the thyroliberin-stimulated thyroxine levels by 44% and the expression of genes encoding the Na^+^/I^−^ symporter and thyroperoxidase by 75 and 83%. NCGC00242364 also reduced thyroxine levels and thyroid gene expression in mice treated with TSHR-stimulating antibodies. These data indicate the high efficacy of NCGC00242364 and its analogues in the treatment of Graves’ disease. Since these compounds do not affect the basal activity of TSHR, this avoids the development of hypothyroidism, one of the undesirable effects when using allosteric inverse agonists of TSHR [318,319].

Among inverse TSHR agonists, compounds NCGC00161856 [346] and NCGC00229600 [347] had significant activity. They inhibited not only stimulated TSH or antibodies but also basal TSHR activity. NCGC00229600 also prevented TSH- and antibody-dependent AC stimulation in a primary culture of fibroblasts obtained from the retroorbital region of patients with Graves’ disease and was characterized by increased TSHR expression [348]. The ability of NCGC00229600 to suppress TSHR activity in retroorbital fibroblasts is important in terms of preventing ophthalmopathy, the most severe symptom of Graves’ disease [349].

We have developed two thieno[2,3-d]-pyrimidine derivatives, TPY1 and TP48, with antagonistic activity against TSHR [350,351]. When administered to rats, TPY1 reduced thyroliberin-stimulated production of thyroid hormones and the expression of thyroglobulin, thyroperoxidase, and Na^+^/I^−^ symporter genes in the thyroid but did not affect basal levels of thyroid hormones indicating its activity as a neutral allosteric TSHR antagonist [351]. In turn, TP48 not only suppressed thyroid hormones levels and expression of thyroid-specific genes stimulated by thyroliberin, but 3.5 h after administration to rats, it reduced the basal levels of thyroid hormones and the expression of the *Nis* gene encoding the Na^+^/I^−^ symporter [350,351]. This indicates that TP48 functions as a mild inverse TSHR agonist [351].

In 2019, the compound S37a with seven centers of chirality was developed, which one of the stereoisomers at micromolar concentrations suppressed the AC activity and the cAMP accumulation in HEK293 cells with expressed TSHR when they were treated with TSH and stimulatory monoclonal antibodies TSAb M22 and KSAb1, oligoclonal stimulatory antibodies TSAb characteristic of patients with Graves’ disease, as well as a small C2 agonist [323]. S37a, when administered orally to mice, had a high bioavailability (53%) and was not toxic [323], which indicates the prospects for its clinical trials, especially since there are currently no effective and safe approaches for the treatment of Graves’ disease and Graves’ ophthalmopathy [352].

In addition to allosteric sites localized in the ECLs and TMD, such sites are also located in the ICLs of TSHR as evidenced by our results on the stimulating effect on the TSHR activity of pepducin, which corresponds to region 612–627 of TSHR ICL3 [353,354] (Figure 4). Pepducin 612–627(Palm) modified with palmitate at the C-terminus stimulated the AC activity in rat thyroid membranes, and its action was specific for TSHR and was not detected in membranes where TSHR was absent [353]. When administered intranasally to rats, 612–627(Palm) increased the level of thyroid hormones, and this effect was dose-dependent [354]. The obtained data suggest that 612–627(Palm) is an intracellular allosteric TSHR agonist, and hydrophobic palmitate is critical for its activity providing the transport of pepducin into the cell since the non-palmitoylated analog was inactive [353,354].

#### Autoantibodies to the TSH Receptor

Autoimmune thyroid diseases are found in an average of 5% of the population, the most common among which is diffuse toxic goiter (Graves’ disease), which develops as a result of an autoimmune attack to TSHR [355]. For almost 40 years, the presence of antibodies to TSHR have been used to differentiate Graves’ disease [356], and this approach is constantly being improved [357]. Assessment of the titer and pattern of autoantibodies to TSHR is widely used to differentiate Graves’ disease from toxic multinodular goitre type A [358] and other forms of thyrotoxicosis [359].

Autoantibodies to TSHR have the activity of full agonists, inverse agonists, antagonists, and in some cases do not have their intrinsic activity but allosterically affect TSHR activity stimulated by TSH or stimulatory autoantibodies [360,361,362,363,364]. Stimulatory autoantibodies bind to TSHR, which is in an inactive conformation, activate the receptor, and trigger signaling cascades through G_s_- and G_q/11_-proteins, which leads to increased synthesis and secretion of thyroid hormones [365,366]. At the same time, they can compete with TSH for the binding site in the receptor ectodomain or allosterically affect its conformation. Binding of stimulatory autoantibodies is carried out with the receptor ectodomain but can also include the hinge region connecting the ectodomain with the TMD and the ECLs connecting TMs [362,366,367,368] (Figure 4). Neutral antibodies interact with the receptor in a similar way with the only difference being that for them the main molecular determinants are localized not in the ectodomain but in the hinge region, which includes the sequence 316–366 [366,369]. Monoclonal antibodies to the TSHR hinge region, isolated from the blood serum of animals with experimental autoimmune thyroiditis, stimulated TSHR, increased the level of cAMP in thyrocytes, activated the 3-phosphoinositide pathway, the MAPK cascade, and different isoforms of protein kinase C, and also increased the activity of the pro-inflammatory factor NF-kB [367]. These data indicate a functional similarity between TSH and stimulatory anti-TSHR antibodies and demonstrate that the hormone and stimulatory antibodies interact with highly overlapping sites located in the receptor ectodomain although these sites are still not completely identical [368]. The effect of stimulatory anti-TSHR antibodies is an increase in thyroid hormone production leading to autoimmune hyperthyroidism (Graves’ disease) as well as Graves’ ophthalmopathy, and more than half of the patients with Graves’ disease are positive for such antibodies [370,371].

Blocking autoantibodies inhibits the binding of TSHR to TSH and stimulatory anti-TSHR antibodies preventing activation of the receptor and thereby reducing the production of thyroid hormones by thyrocytes. Blocking antibodies bind to linear and “conformational” epitopes located in the LRR subdomain, which includes a significant part of the TSHR ectodomain [366,368,372]. However, unlike stimulatory antibodies, for which the “center” of binding to the LRR subdomain is shifted to its C-terminus, in the case of blocking anti-TSHR antibodies such a “center” is shifted to its N-terminus and thus only partially overlaps with the binding sites for TSH and stimulatory autoantibodies [366,368]. This is supported by the fact that the Trp^265^ located in the tenth LRR, corresponding to the C-terminal portion of the LRR subdomain, effectively interacts with the light chain of stimulatory autoantibody M22 but is not involved in binding with inhibitory antibody K1-70 [368]. The ability of blocking autoantibodies to prevent TSHR hyperactivation can be used to treat Graves’ disease and Graves’ orbitopathy [341,368,373]. Currently used antithyroid drugs have many side effects, and even after five years of treatment with these drugs 21% of them remain positive for TSHR autoantibodies [374]. However, it should be taken into account that, along with the suppression of thyroid hormone production, some blocking and neutral autoantibodies can affect G_q/11_-dependent signaling pathways, as well as enhance cell proliferation by activating the MAPK cascade [365,366], which indicates their varying degrees of bias in relation to TSHR-dependent signaling pathways.

### 9.2. Luteinizing Hormone Receptor

LHR has a significant similarity with TSHR both in membrane topology and in activation mechanisms since in this case gonadotropin, LH, or human chorionic gonadotropin (hCG) binds with high affinity to the ectodomain, which leads to conformational changes in the TMD and provides a functional interaction of LHR ICLs with G proteins and β-arrestins [375,376]. In gonadotropin-stimulated LHR, the cavity of the transmembrane tunnel in which allosteric sites are located remains ligand-free. It is filled with water molecules that form hydrogen bonds with the polar groups of amino acid residues located in the inner cavity of this site. In LHR, as well as in the FSH receptor, along with the main allosteric site, there is an additional (minor) allosteric site, binding to which modulates the activity of the main site [377,378]. The main allosteric site is formed by TM3, TM4, TM5, and TM6 [378,379,380], while the additional allosteric site is formed by TM1, TM2, TM3, and TM7 [377]. Both sites are deepened to varying degrees into the TMD and overlap resulting in a wide variety of pharmacological profiles for their ligands, including allosteric agonists and antagonists, allosteric modulators (PAM, NAM), and ago-PAM [334,381] (Figure 5).

The need to create allosteric regulators is due to both the low selectivity of LH and hCG to intracellular cascades and the peculiarities of pharmacological preparations of gonadotropins since their urinary forms (hCG) have a significant number of bioactive impurities and are variable in specific activity, while recombinant forms (LH, hCG) significantly differ from natural gonadotropins in the pattern of N-glycosylation and regulatory properties. Along with this, gonadotropins are not very effective in the case of mutant forms of LHR, which have a reduced ligand-binding capacity or impaired translocation to the membrane, even if binding to gonadotropin is preserved [382,383,384]. In this regard, it should be noted that some of the developed small allosteric LHR agonists have the properties of chaperones, which stabilize the structure of LHR and facilitate their translocation to the plasma membrane, which increases their sensitivity to endogenous LH, as was experimentally shown for thieno[2,3-d]-pyrimidine derivative Org42599 [385,386,387].

The first allosteric regulators of LHR with agonist activity were obtained in 2002, the most effective of which were the thieno[2,3-d]-pyrimidine derivatives Org41841 and Org43553, which act on LHR at nanomolar concentrations [327]. They specifically bound to LHR without interfering with the activation of the receptor by gonadotropins and were characterized by a bias towards G_s_-proteins and cAMP-dependent cascades and very weakly stimulated G_q/11_-proteins and phosphoinositide metabolism, as well as β-arrestins [388,389,390]. By studying the binding of Org41841 and Org43553 to LHR with mutations in the TMD, it was found that the Tyr^570^, Phe^585^, and Tyr^643^ are important for the formation of the allosteric site, and this site is structurally similar to that in TSHR [330]. The negatively charged Glu^506^ in TM3 was key for the interaction with thieno[2,3-d]-pyrimidines since its replacement with alanine blocked the binding of Org41841 to mutant LHR. This is due to the formation of a salt bridge between the carboxyl group of glutamic acid and the positively charged amino group of the allosteric ligand [328,330]. In vivo, Org43553 was effective not only when administered intraperitoneally and subcutaneously but also when administered orally due to its good absorption and stability in the gastrointestinal tract [390,391,392,393]. Oral administration of Org 43553 stimulated ovulation in immature mice and adult rats. The resulting eggs were of good quality, had high fertility, and when implanted with a high yield gave viable embryos. Rats did not show signs of ovarian hyperstimulation, which is often observed with gonadotropin treatment, and this was due to a relatively mild stimulation of steroidogenesis and a lower half-life of the drug (two times lower than that of hCG) [390]. In addition, Org 43553 had little effect on the production of vascular endothelial growth factor, the signaling pathways of which are involved in increased vascular permeability and provoke ovarian hyperstimulation syndrome [392]. Oral administration of Org 43553 at the same dose to male rats stimulated testicular steroidogenesis and increased blood testosterone levels [390]. When taken orally by women of reproductive age, Org 43553 (300 mg) caused ovulation in 83% of them without significant adverse effects, including ovarian hyperstimulation syndrome [393].

We have developed a series of thieno[2,3-d]-pyrimidine derivatives with the properties of allosteric agonists and (or) ago-PAMs and allosteric antagonists of LHR [324,325,394,395,396,397,398,399,400,401,402]. Compounds TP03 and TP04, the most active of the full agonists, stimulated the AC activity in testicular membranes isolated from the rat testes, increased testosterone production by the cultured Leydig cells, and also stimulated testicular steroidogenesis and increased testosterone levels when administered intraperitoneally, subcutaneously, and orally to both healthy male rats and animals with androgen deficiency caused by types 1 and 2 diabetes mellitus and aging [395,400,402,403,404]. Using molecular docking, it was found that the efficiency and selectivity of the studied thieno[2,3-d]-pyrimidines correlate with the characteristics of their binding to the transmembrane allosteric site of LHR, and the key parameter of such binding was the intensity of hydrophobic contact while the Coulomb interactions and hydrogen bonds were less significant [400]. When co-administered with low doses of gonadotropin, thieno[2,3-d]-pyrimidines potentiated its effect, and potentiation was more pronounced in rats with type 1 diabetes mellitus [405]. As with hCG, TP03 and TP04 improved spermatogenesis in diabetic and aging rats [400,402,406,407]. When TP03 was administered to metformin-treated diabetic rats, a significant increase in its steroidogenic effect was shown upon short-term (but not long-term) TP03 administration [402,406]. In contrast to hCG, long-term administration of TP03 and TP04 to male rats did not suppress the expression of the *Lhr* gene in the testes, and their steroidogenic effect was preserved and even increased, which indicates the possibility of their long-term use to stimulate testicular steroidogenesis in the conditions of androgen deficiency [400,401]. The stimulating effect of TP03 on progesterone production and ovulation induction was shown by us in immature female rats pretreated with FSH [408], as well as in proestrus adult female rats pretreated with a GnRH antagonist [409]. These results indicate the promise of using TP03 and TP04 for correcting androgen status and improving spermatogenesis in metabolic disorders and aging, as well as for controlled ovulation induction.

Along with thieno[2,3-d]-pyrimidines, 1,3,5-pyrazole and terphenyl derivatives can also function as allosteric LHR ligands with full and inverse agonist activity [378,410,411]. Among terphenyl derivatives, LUF5771 inhibited LHR stimulation by gonadotropins and Org 43553 [378,411]. Compound 1, the most active pyrazole derivative with the activity of a full agonist, stimulated the AC activity and increased testosterone production by Leydig cells both in cell culture and when administered intraperitoneally to male rats [410]. Like Org43553, the pyrazole and terphenyl derivatives did not compete with hCG for binding with LHR, which is in favor of their interaction with the transmembrane allosteric site of the receptor [378,410].

Based on the thieno[2,3-d]-pyrimidine structure, we created an allosteric TP31 antagonist, which, in the in vitro experiments, reduced hCG- and TP03-stimulated AC activity in testicular membranes, and when administered intraperitoneally to male rats, it slightly reduced the baseline testosterone and largely inhibited the stimulatory effect of hCG on testosterone production [397,398] (Figure 5). Since pre-treatment with TP31 suppressed TP03-induced receptor activation in a dose-dependent manner, this indicates a common binding site. Compared to thieno[2,3-d]-pyrimidines with pronounced agonistic activity, TP31 has an additional ethylamine group in the second position of the heterocyclic ring of nicotinic acid, which changes the volume and charge of the variable substituent in 5-amino-4-(3-aminophenyl)-*N*-(*tert*-butyl)-2-(methylsulfanyl)thieno[2,3-d]pyrimidine-6-carboxamide, a framework molecule for the synthesis of LHR allosteric regulators [398]. Small ligands with inverse agonist or antagonist activity can be used to suppress steroidogenic function in the treatment of prostate cancer and in the treatment of gonadotropin-dependent tumors.

The importance of intracellular allosteric sites in the regulation of LHR activity is supported by our data on the effect of peptides derived from ICL3 of this receptor [412,413,414]. It should be mentioned that ICL3, both in LHR and in many other GPCRs, includes the key determinants responsible for functional coupling with G proteins. The ICL3-derived peptide NKDTKIAKK-Nle-A(562–572)-K(Palm)A, corresponding to LHR region 562–572 responsible for interaction with G_s_-proteins, increased the basal AC activity in testicular membranes, inhibiting hCG-induced stimulation of the enzyme [413] (Figure 5). When administered intratesticularly to male rats, NKDTKIAKK-Nle-A(562–572)-K(Palm)A increased testosterone levels exerting a stimulating effect on testicular steroidogenesis, but when administered intraperitoneally, its effect was weak, which may indicate its degradation in the bloodstream [413]. Thus, pepducins corresponding to LHR ICL3 may be effective as intracellular allosteric regulators of LHR, but further studies are required to improve their stability and bioavailability.

Since LHR is similar in structure and regulatory mechanisms to TSHR, it cannot be ruled out that it can also be allosterically regulated by autoantibodies against the LHR ectodomain. However, at present, reliable data on the possibility of such regulation, as well as on the association between the presence of autoantibodies to LHR and reproductive dysfunctions, have not been obtained. There is a clinical study in which an attempt was made to trace the relationship between the presence of stimulatory anti-LHR antibodies in the blood of women with polycystic ovary syndrome (PCOS) and the severity of their hyperandrogenemia [415]. However, the data obtained indicate a low occurrence of anti-LHR autoantibodies in both the control and PCOS groups and also demonstrate the absence of significant differences in the occurrence of these antibodies between groups. At the same time, there is evidence that autoantibodies to the FSH receptor may be involved in the pathogenesis of FSH-resistant PCOS [416]. As a result, further studies are needed to study LHR autoantibodies, primarily in patients resistant to LH.

## 10. Allosteric Regulators of β-Adrenergic Receptors

As in the structure of most GPCRs, allosteric sites in β-AR are located in all main subdomains, which is confirmed by the influence of autoantibodies to ECLs, synthetic ICL-derived pepducins, as well as the endogenous and synthetic small compounds capable of interacting with sites located in different loci of the TMD and its interfaces with ICLs. The multiplicity of these sites was demonstrated using the ligand competitive saturation (SILCS) computational method, which made it possible not only to predict the possible localization and functional activity of allosteric sites in the β_2_-AR but also to suggest potential candidates as regulators of these sites [417]. Below, we will briefly discuss some synthetic small molecular weight allosteric regulators since they, like small endogenous regulators (cholesterol, metal ions, etc.), are described in detail in a number of reviews and experimental works [417,418,419,420,421,422], and we will pay more attention to autoantibodies and pepducins.

In 2017, based on the DNA-encoded small-molecule library, compound 15 (Cmpd-15) with NAM activity for β_2_-AR was developed [423]. Cmpd-15 is able of entering the cell and binding specifically to the intracellular allosteric site formed by the cytoplasmic ends of TM1, TM2, TM3, and TM7, as well as the ICL1 and the H8 helix of β_2_-AR [68]. The result of this is the stabilization of the receptor in an inactive state, a decrease in the affinity of β_2_-AR for orthosteric agonists, and inhibition of its binding to G proteins and β-arrestins [68,421]. As shown for the polyethylene glycol-carboxylic acid derivative Cmpd-15PA, which has the same pharmacological characteristics as Cmpd-15, the polar groups of this derivative bind to the allosteric site through interactions with polar amino acid residues (Thr274^6.36^, Ser329^8.47^, Asp331^8.49^, Asn69^2.40^, and Arg63^ICL1^) while the cyclohexyl and phenyl rings of Cmpd-15PA interacted with the hydrophobic package formed by the residues Val54^1.53^ and Ile58^1.57^ at the end of TM1, Leu64^ICL1^ (ICL1), Ile72^2.43^ (TM2), Leu275^6.37^ (TM6), as well as Tyr326^7.53^ from the conserved NPxxY sequence located in TM7 and Phe332^8.50^ located in the H8 helix [68]. Subsequently, analogs of Cmpd-15 with NAM activity for β_2_-AR were synthesized, and the study of their structure and molecular docking confirmed the decisive importance of their interaction with hydrophobic residues that form the allosteric site of β_2_-AR [424].

In 2018, compound 6 (Cmpd-6) with PAM activity for β_2_-AR was developed, which also interacted with an allosteric site located in the region of the intracellular vestibule of the transmembrane tunnel (ICL2, cytoplasmic ends of TM3 and TM4) stabilizing the active conformation of the receptor [423,425]. This compound interfered with the effects of the β-AR blocker carvedilol. Cmpd-6, on the one hand, increased the affinity of carvedilol for β_2_-AR and thereby enhanced its inhibitory effect on β_2_-agonist-induced stimulation of G_s_-proteins and cAMP signaling and, on the other hand, enhanced the stimulatory effects of carvedilol on ERK1/2 activity, endocytosis of β_2_-Ars, and their trafficking into lysosomes [426], which, as was shown later, is also due to the activation of G_s_-proteins [427].

Allosteric sites located on the cytoplasmic side of the β-AR TMD are targets not only for low-molecular-weight compounds but also for pepducins corresponding to the ICLs and ICL/TM interfaces of β-AR. In 2014, the group of Jeffrey Benovic synthesized pepducins corresponding to different regions of ICL1, ICL2, and ICL3 of β_2_-AR [428]. As a result, three groups of pepducins were identified that differ in the mechanisms of action on β_2_-AR, G_s_-proteins, and β-arrestins. The first group included pepducins, which activated G_s_-proteins in a receptor-independent manner. Pepducins of the second group had a biased activity towards the β-arrestin pathways causing activation of MAPK, desensitization of β_2_-AR, and, as a result, reducing or blocking the cell response to β_2_-agonists. Pepducins of the third group receptor-dependently stimulated G_s_-proteins and cAMP-signaling pathways stabilizing the active conformation of β_2_-AR, and these effects were not accompanied by site-specific phosphorylation by GRK-kinases, which prevented β-arrestin-mediated internalization of β_2_-ARs and maintained or even increased their sensitivity to β_2_-agonists [428].

During congestive heart failure, the level of catecholamines increases and their stimulating effect on β-AR increases, which in cardiomyocytes leads to chronic activation of G_s_-dependent β_1_-AR and G_i/o_-dependent β_2_-AR. As a result, apoptotic processes leading to the death of cardiomyocytes are enhanced, the expression of β_1_-AR is reduced, and the inotropic reserve of the myocardium is weakened. Blockers of the β-AR orthosteric site, as a rule, have low selectivity for transducer proteins, which negatively affects the activity of G protein-independent β-arrestin pathways responsible for transient β-AR desensitization and cardiomyocyte survival. Pepducins, derivatives of the ICL of β-ARs, can become a suitable alternative to such blockers. In cultured cardiomyocytes, pepducin ICL1–9, corresponding to β_2_-AR ICL1, stimulated β_2_-AR phosphorylation, induced β-arrestin recruitment and subsequent β_2_-AR internalization, and activated β-arrestin signaling cascades functioning as β-arrestin-biased allosteric agonist [429]. The study of the mechanisms of action of ICL1–9 showed that, along with the activation of ERK1/2, it activates the epidermal growth factor receptor, which indicates its powerful growth-stimulating and regenerative potential [429]. Intramyocardial injection of pepducin ICL1–9 into mice reduced the size of myocardial infarction induced by ischemia/reperfusion, reduced the death of cardiomyocytes, and improved heart function in acute and prolonged remodeling of myocardial fibrosis indicating its pronounced cardioprotective effect in acute ischemic myocardial injury [430]. The antiapoptotic effect of ICL1–9 was dependent on the functional state of β_2_-AR, as well as on the presence of β-arrestins, since this effect was not detected in mutant mice lacking β_2_-AR and β-arrestins. A key role in the anti-apoptotic effect of ICL1–9 belongs to the signaling pathway, including RhoA, ROCK, and protein kinase D, which can be regulated both through β-arrestins and G_i_-proteins, which can functionally couple with β_2_-AR [430,431]. Thus, pepducins that function as ligands for intracellular allosteric sites of β_2_-AR and have a wide range of pharmacological activity are good candidates for the development of new drugs for the treatment of cardiovascular diseases [234,429,430,431].

Currently, a large number of autoantibodies to the ECLs of β_1_- and β_2_-AR are known, which are closely associated with cardiovascular diseases [432,433,434]. The most extensive and clinically important group are autoantibodies to β_1_-AR ECL2, which the presence of certain types is the root cause of dilated and ischemic cardiomyopathy [434,435,436,437,438]. Antibodies to ECL2, depending on the epitope for which they are produced, upon binding to β_1_-AR induce conformational changes that stabilize both the active and inactive conformation of the receptor. In the case of stabilization of the active conformation, an increase in the affinity of β_1_-AR for the agonist and an increase in cAMP production are observed, which leads to cardiovascular pathology [436,439]. Among the pattern of autoantibodies to β_1_-AR isolated from the blood of patients with dilated cardiomyopathy, antibodies designated as “P4” predominate, which suppress the internalization of receptors, cause a β_1_-AR-mediated increase in intracellular cAMP levels in the absence of an agonist, and inhibit the stimulating effect of isoproterenol on AC activity [439]. According to a number of pharmacological characteristics, these antibodies can be assigned to the ago-NAM group (Table 1) with activity biased towards cAMP-dependent signaling pathways. The fact that the decrease in β_1_-AR internalization caused by autoantibodies does not lead to an increase in agonist-induced AC activity can be explained by the fact that the recycling of desensitized receptors is impaired. A certain contribution may be made by the negative influence of autoantibodies on the binding characteristics and accessibility of the orthosteric site. It is important that autoantibodies to β_1_-AR were detected in a significant proportion of healthy subjects without any signs of cardiovascular pathology, but these autoantibodies, unlike those in patients with dilated cardiomyopathy, did not have a stimulating effect on β_1_-AR and in some cases even had weak antagonistic activity [437].

To identify the ECL epitopes that are most significant for β-AR activity, autoantibodies were studied by immunoprecipitation with peptides and their constructs derived from different ECL regions of β1- and β2-AR, as well as by immunization of experimental animals with peptides structurally corresponding to these regions [433,440]. When analyzing the immunoprecipitation of autoantibodies to β_1_-AR in patients with different forms of cardiomyopathy, it was found that immunogenic epitopes were localized in ECL2 (195–225) of β_1_-AR. In cardiomyopathy in patients with Chagas disease, the epitope included the region 201–205, and in postpartum cardiomyopathy, the epitope corresponded to the region 200–210. In dilated cardiomyopathy, the epitopes were located in overlapping regions 183–208, 197–202, 206–212, and 213–218, including TM4 segments adjacent to ECL2 [440,441,442]. More recently, to establish a more precise localization of the epitope responsible for the production of cardiomyopathy-inducing autoantibodies, a cyclic peptide corresponding to β_1_-AR ECL2 was used, which made it possible to identify the key molecular determinants of this epitope, including the NDPK^211–214^ segment and the C209–C215 cysteine bridge. Segment 211–214 was located at the C-terminal end of the backward-oriented α-helix constituting ECL2 in β1-AR [433]. In this case, the trigger of allosteric activation of β_1_-AR is most likely the Asp^212^, the negative charge of which is neutralized upon interaction with the autoantibody. This assumption is based on the fact that substitution of Asp^212^ for asparagine leads to paradoxical activation of the mutant receptor by antagonists broxaterol and terbutaline [443]. In the experiments with rodents, a cyclic peptide corresponding to β_1_-AR ECL2 prevented the development of chronic heart failure caused by autoantibodies to β_1_-AR [433,444]. A cardioprotective effect on myocardial injury in mice was demonstrated for another cyclic peptide, RD808, also corresponding to ECL2 of β1-AR [445].

Monovalent Fab-fragments of antibodies that were produced against the ECL2 region 173–180 of β_2_-AR bound to β_2_-AR and inhibited the stimulatory effects of β_2_-agonists on AC activity in cultured cardiomyocytes. At the same time, bivalent antibodies stimulated β_2_-AR-mediated signaling pathways and caused contraction of cardiomyocytes, and these effects of bivalent antibodies were due to their two-center binding to receptor molecules promoting β_2_-AR dimerization [446]. Subsequently, it was found that the stimulatory effects of these autoantibodies are specific for cAMP-dependent cascades mediated through G_s_-proteins, which indicates their bias towards β_2_-AR-coupled transducer proteins [447]. These data indicate that the pharmacological profile of autoantibodies to β_2_-AR depends on the number and localization of binding sites in the ECL of the receptors.

## 11. Conclusions

Allosteric regulation plays a decisive role in the functioning of all known representatives of GPCRs since allosteric mechanisms are involved in the regulation of their basal activity and the main stages of GPCR-mediated signal transduction, including orthosteric ligand binding, and are also responsible for receptor desensitization, endocytosis of ligand-receptor complexes (including intracellular signaling), and their recycling or degradation. Since ligands of allosteric sites affect various functions of the receptor, including biased agonism, and are characterized by varying degrees of selectivity for GPCRs, this predetermines their diversity in chemical structure, pharmacological profile, and localization and configuration of the sites with which they interact. Low-specific allosteric regulators include simple ions (protons, Na^+^, Zn^2+^, Mg^2+^, Mn^2+^, and some others), which penetrate into the inner cavity of the transmembrane tunnel of most GPCRs and, being in complex with the amino acid residues that form the allosteric site, modulate their functional activity. This determines the influence of the ionic composition and pH in the extra- and intracellular environments on the activity of GPCRs and may mediate the effect of the electrochemical properties of the membrane on GPCR signaling. Ion chelators, including peptides and proteins, affect the concentration and ratio of ions available for interaction with allosteric sites, and thus are able to regulate the GPCRs. This greatly expands the range of potential allosteric regulators.

An important group of allosteric modulators is membrane lipids (cholesterol, phospholipids, gangliosides, and others) that interact with allosteric sites of GPCRs located on the outer membrane-contacting surface of the TMD, as well as in interfaces formed by TMs and hydrophilic loops. These interactions are involved in the regulation of receptor activity since they affect the conformational mobility and configuration of the transmembrane tunnel and are critical for the ability of receptors to form the complexes with other components of GPCR signaling. The influence of the physicochemical properties and composition of membranes on the activity of GPCRs is carried out largely through allosteric lipid–receptor interactions, and this occurs not only in the plasma membrane but also in the membranes of endosomes, the Golgi apparatus, and the nucleus, where intracellular GPCR signaling occurs. It is not entirely clear to what extent the effects of lipids are specific to a particular receptor. Some lipids, such as cholesterol, target various groups of GPCRs while other lipids, such as PI(4,5)P_2_, may be receptor selective and biased for intracellular signaling.

Endogenous allosteric GPCR regulators include heterotrimeric G proteins and β-arrestins, which are part of pre-activation complexes and mediate signal transduction after receptor activation by an orthosteric or allosteric agonist. Unlike small endogenous regulators, the interaction of transducer proteins with cytoplasmic subdomains and the intracellular vestibule of the receptor’s transmembrane tunnel is usually multicentric and often involves molecular determinants that form various intracellular allosteric sites. In addition, in the functionally active state, some GPCRs are capable of forming homo- and heterodimeric complexes in which protomers interact allosterically, and this is another allosteric mechanism that is responsible for the stability and mutual transitions of active and inactive receptor conformations, the ability of the receptor to be activated by an orthosteric agonist, and also for biased agonism. The formation of homo- and heterodimeric GPCR complexes involves multiple allosteric sites located in hydrophilic loops and on the outer TMD surfaces of GPCRs. Each protomer in the complex is subject to multiple allosteric influences. As a result, in an oligomeric complex that includes protomers from different types of receptors, allosteric regulation becomes much more complicated.

Highly specific allosteric regulators of GPCRs are autoantibodies developed against antigenic determinants located in receptor ECLs. They can stimulate or, conversely, inhibit GPCRs, as well as modulate their activity, including biased regulation of intracellular cascades. Considering the fact that the number of identified and studied autoantibodies to GPCRs is steadily increasing, there is no doubt that for many receptors there is a wide range of such autoantibodies with a different profile of functional activity, and these autoantibodies can be the root causes of many diseases. At the same time, autoantibodies to GPCRs are found in healthy subjects although they differ from autoantibodies in patients with autoimmune diseases. This indicates that, even under normal conditions, autoantibodies to GPCRs may be involved in the allosteric regulation of GPCR signaling.

The complexity of allosteric effects caused by numerous regulatory molecules differing in structure, availability, and mechanisms of action predetermines the multiplicity and different topology of allosteric sites in GPCRs. Allosteric regulators can specifically interact with the sites located in the ECLs, ICLs, and the internal cavity of the transmembrane tunnel, as well as on the lateral, membrane-contacting surface of the TMD. Along with this, allosteric effects can be mediated by the interaction of various molecules (“indirect” allosteric regulators) with signal proteins that form functionally active complexes with GPCRs (G proteins, β-arrestins, RAMPs, protomers of other receptors, etc.). In such cases, one can speak of stepwise, both temporal and spatial, allosteric regulation of GPCR-containing signaling complexes. As a result, the search and development of artificial (synthetic) allosteric ligands should take into account the topological diversity and functional complexity of the endogenous allosteric regulation of GPCRs. It is important that artificial allosteric ligands are balanced and harmoniously included in the mechanisms of endogenous allosteric regulation supplementing and modifying it in the necessary way.

In recent years, for many GPCRs, impressive progress has been made in the development of a large number of allosteric regulators with various pharmacological activities, and many of them are not only selective for a certain type of receptor but also characterized by a biased activity towards intracellular cascades, which is very valuable and in high demand in medicine. Using molecular docking and genetic approaches, new populations of allosteric sites in GPCRs have been identified, which creates prerequisites for the search for their endogenous regulators, as well as for the creation of their artificial ligands. At the same time, it should be noted that in the process of translating the results of studies on the efficacy and safety of allosteric regulators obtained in vitro and in animal models, many intractable problems appear in the clinic, and this is one of the reasons for the very limited arsenal of pharmacological drugs based on allosteric GPCR regulators. More intensive studies are needed on the allosteric mechanisms of the regulation of GPCRs in real biological systems at the organism level, where receptors are under the influence of many factors. At the same time, despite the difficulties and limitations, there is no doubt that given the high specificity of action, including biased agonism, and also taking into account moderate activity, which excludes blocking or hyperactivation of receptors as is often observed for orthosteric ligands, allosteric GPCR-regulators will occupy a significant niche in the pharmacological industry in the coming years.

## Figures and Tables

**Figure 1 ijms-24-06187-f001:**
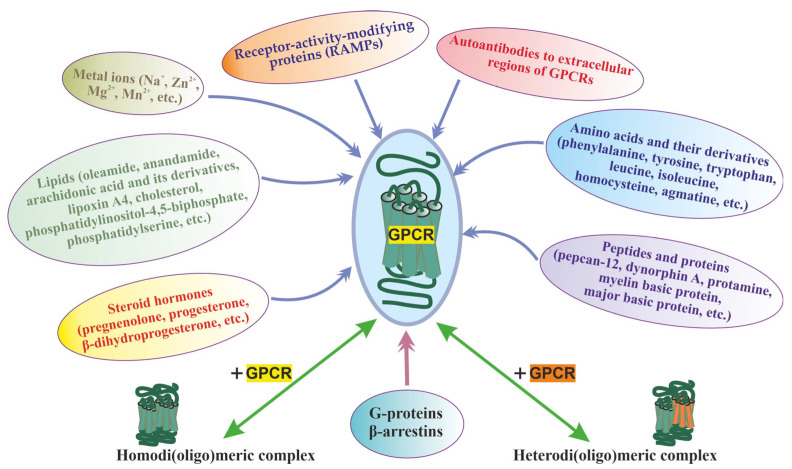
The main group of allosteric regulators of GPCRs, including the homo- and heterodimerization, along with the main classes of allosteric regulators (ions, lipids, amino acids, peptides, proteins, steroid hormones, autoantibodies to extracellular regions of GPCRs), the heterotrimeric G proteins, β-arrestins, and RAMPs are shown, which form complexes with GPCRs during signal transduction acting allosterically on their functional activity. The allosteric effect of homo- and heterodi(oligo)merization of the receptors has also been shown, as demonstrated for class C GPCRs and some representatives of the class.

**Figure 2 ijms-24-06187-f002:**
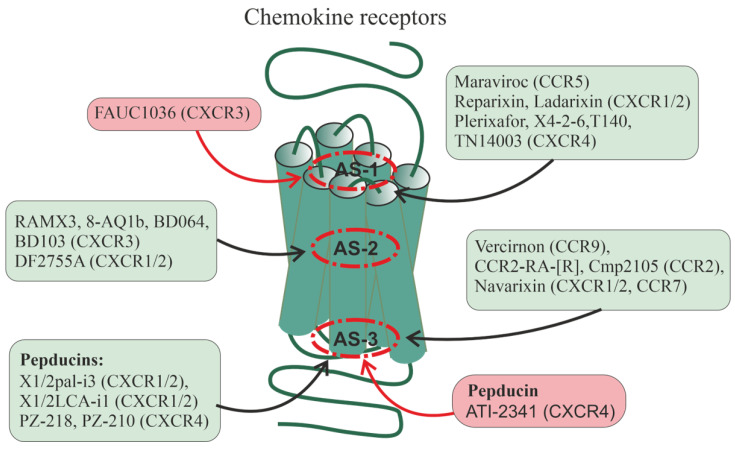
Allosteric regulation of chemokine receptors. There are data on the allosteric regulation of a significant number of chemokine receptors, including CXCR1, CXCR2, CXCR3, CXCR4, CCR2, CCR5, CCR7, and CCR9. Allosteric sites in them can be located at different loci of the TMD: in the upper (outwardly oriented) portion of the transmembrane tunnel (interfaces including the membrane-proximal segments of ECLs and extracellular segments of TMs, as well as the outer vestibule of the transmembrane channel) (designated as AS-1), in the central part of the 7TM bundle (AS-2), and in the lower (oriented to the cytoplasm) portion of the transmembrane tunnel (interfaces, including the membrane-proximal segments of ICLs and the cytoplasmic endings of TMs) (AS-3). In each of these loci, there are several cavities in which allosteric sites can be located, both overlapping and spatially separated. Along with this, allosteric sites can be located in hydrophilic loops being targets for autoantibodies and synthetic pepducins. Endogenous allosteric regulators of the chemokine receptors are sodium (NAM) and zinc (PAM) ions, as well as cholesterol and membrane phospholipids (mainly with PAM activity). Synthetic small allosteric regulators interact with allosteric sites located at all three loci, AS-1, AS-2, and AS-3, while pepducins, lipidated derivatives of peptides corresponding to ICLs, are transported across the plasma membrane and interact with intracellular sites. Figure 2 shows the following small regulators interacting with the AS-1 locus: maraviroc (CCR5-selective NAM), reparixin, ladarixin and their analogs (CXCR1/CXCR2-selective NAMs and/or non-competitive allosteric antagonists), plerixafor (allosteric CXCR4-inhibitor), FAUC1036 (biased allosteric CXCR3-agonist), T140 and its analog TN14003 (biased allosteric CXCR4-antagonists), and the peptide X4-2-6 (TM2/ECL1 of CXCR4; allosteric CXCR4-antagonist). The following ligands bind specifically to the AS-2 locus: RAMX3 and its biased analogs 8-azaquinazolinone (8-AQ) derivative 1b, BD064, and BD103 (CXCR3-selective NAMs), DF2755A (noncompetitive allosteric CXCR1/CXCR2-antagonist). The following ligands bind specifically to the AS-3 locus: vercirnon and its analogs (allosteric CCR9-antagonists), navarixin (allosteric antagonist of CXCR1, CXCR2 and CCR7), CCR2-RA-[R] and Cmp2105 (allosteric CCR2-antagonists and/or NAMs). Pepducins, such as *N*-palmitoylated peptides X1/2pal-i3 (ICL3 of CXCR1/CXCR2; CXCR1/CXCR2-selective NAM), PZ-218 (ICL1 of CXCR4; CXCR4-selective NAM), and PZ-210 (ICL3 of CXCR4; CXCR4-selective NAM), as well as lithocholic acid-modified X1/2LCA-i1 (ICL1 CXCR1/CXCR2; allosteric CXCR1/CXCR2-inhibitor and/or NAM) and ATI-2341 (ICL1 of CXCR4; CXCR4-selective PAM) interact with intracellular sites involved in functional coupling with G proteins and β-arrestins. Antibodies against extracellular regions of chemokine receptors are able to interact with ECLs and the extracellular *N*-terminal domain, but their effects remain poorly understood. Allosteric regulators that increase receptor activity (full agonists) are placed in red squares, while allosteric regulators that decrease receptor activity (antagonists and NAMs) are placed in green squares.

**Figure 3 ijms-24-06187-f003:**
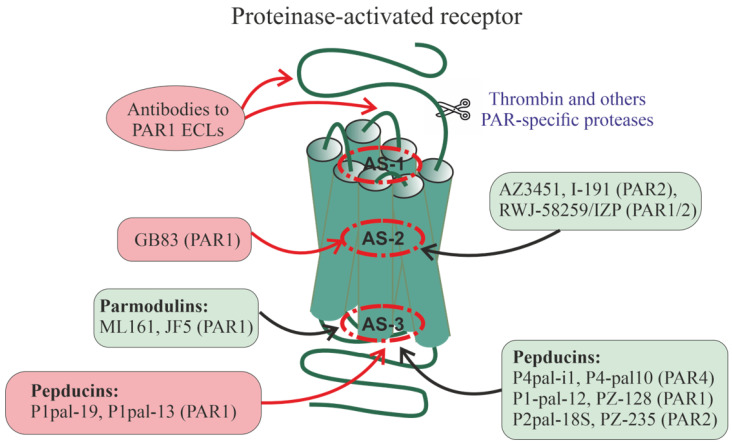
Allosteric regulation of proteinase-activated receptors. There is now evidence of allosteric regulation and localization of allosteric sites for PAR1, PAR2, and PAR4. As in the case of chemokine receptors (see Figure 2), allosteric sites in PARs can be located at different TMD loci (AS-1, AS-2, and AS-3) in the ECLs where they are available for interaction with autoantibodies, as well as in the ICLs where they are targets for pepducins. The small compounds, such as GB83 (PAR1-selective PAM), AZ3451 (allosteric PAR2-inhibitor), I-191 (PAR2-selective NAM), and cross-linked heterobivalent allosteric PAR1/PAR2 inhibitor, RWJ-58259/imidazopyridazine (IZP) derivative), interact with allosteric sites located in the upper and central parts of the TM7 bundle. The low-molecular-weight parmodulins, including the PAR1-allosteric inhibitors ML161 (PM2) and JF5, as well as pepducins, interact with intracellular sites, including the AS-3 locus. The following pepducins have been obtained and have specific activity for PAR1, PAR2, and PAR4: Palm-SGRRYGHALR (P4pal-10) (ICL3 of PAR4, allosteric PAR4-antagonist and NAM and allosteric antagonist for the non-cognate receptors PAR1, FPR2, and FFAR2), Palm-ATGAPRLPST (P4pal-i1) (ICL1 of PAR4, allosteric PAR4-inhibitor), Palm-RCLSSSAVANRS (P1pal-12), Palm-RSLSSSAVANRS (P1pal-12S) and its analog P1pal-7 (PZ-128) (ICL3 of PAR1, allosteric PAR1-antagonists), Pal-RCLSSSAVANRSKKSRALF (P1pal-19), Palm-AVANRSKKSRALF (P1pal-13) (ICL3 of PAR1, full allosteric PAR1-agonists), and Palm-RSSAMDENSEKKRKSAIK^270−287^ (P2pal-18S) and its analog PZ-235 (ICL3 of PAR2, allosteric PAR1/2-antagonists). Data on autoantibodies to PARs are fragmentary, but antibodies to extracellular regions of PAR1 in patients with systemic sclerosis have been shown to function as PAR1-specific allosteric full agonists and/or PAMs. Many of the effects of allosteric regulators are due to their effect on the stability of heterodimeric PAR1/PAR2 complexes, as shown for the heterobivalent PAR1/PAR2-inhibitor and pepducins P1pal-12S and P2pal-18S. Allosteric regulators that increase receptor activity (full agonists and PAMs) are placed in red squares, while allosteric regulators that decrease receptor activity (antagonists and NAMs) are placed in green squares.

**Figure 4 ijms-24-06187-f004:**
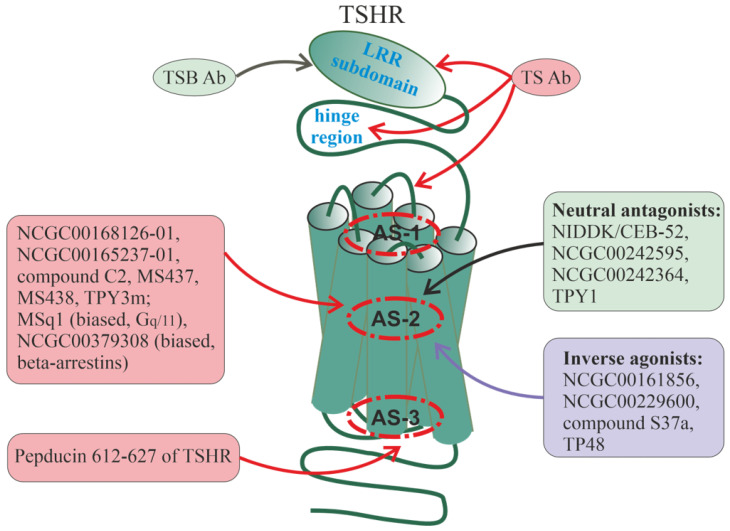
Allosteric regulators of thyroid-stimulating hormone receptor. The TSHR belongs to a family of class A GPCRs having a large extracellular domain in which an orthosteric site is located and which is a target for autoantibodies. Most of the small regulators of TSHR bind to allosteric sites located in the cavity of the AS-2 locus or slightly higher in the lower portions of the AS-1 locus. The following low-molecular-weight compounds specifically interact with allosteric sites in the AS-2 locus: compound C2, NCGC00168126-01 and its analog NCGC00165237-01 (full allosteric TSHR-agonists), MS437, MS438 (biased allosteric TSHR-agonists, did not affect G_i_-proteins), TPY3m (biased allosteric TSHR-agonist, mainly activates G_s_-proteins), MSq1 (biased allosteric TSHR-agonist, mainly activates G_q/11_-proteins), NCGC00379308 (biased PAM, predominantly activates β-arrestins), TPY1, NIDDK/CEB-52 and its analogs NCGC00242595 and NCGC00242364 (allosteric TSHR-antagonists), NCGC00161856, NCGC00229600, compound S37a, and TP48 (allosteric inverse TSHR-agonists). Pepducin 612–627(Palm) (ICL3 of TSHR) interacts with the intracellular allosteric site and functions as an allosteric TSHR-agonist. Stimulating TSHR autoantibodies (TSAb) interact with the LRR subdomain, hinge region, and ECLs. Blocking TSHR autoantibodies (TSBAb) mainly interact with the N-terminal and central portions of the LRR subdomain. Allosteric regulators that increase receptor activity (full agonists and PAMs) are placed in red squares, allosteric regulators with neutral antagonist activity are placed in green squares, while allosteric regulators with inverse agonist activity are placed in blue squares.

**Figure 5 ijms-24-06187-f005:**
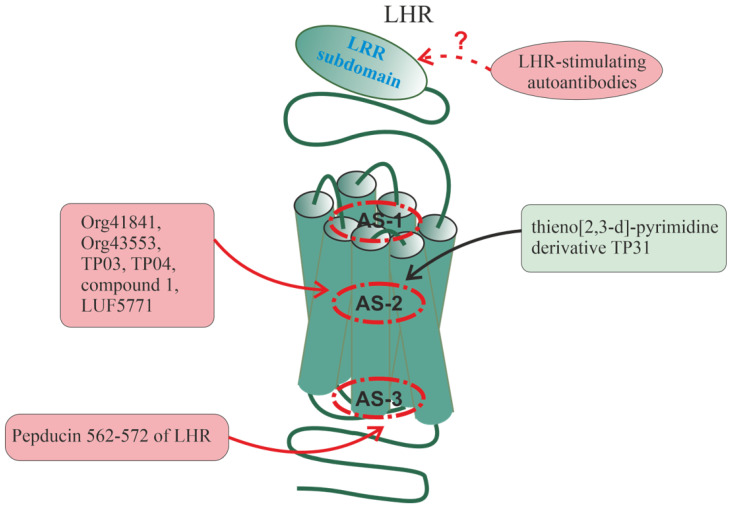
Allosteric regulators of the luteinizing hormone receptor. Like TSHR, LHR has a large extracellular domain in which an orthosteric site is located, and the small LHR regulators bind to allosteric sites located in the cavity of the AS-2 locus. The following low-molecular-weight compounds specifically interact with allosteric sites in the AS-2 locus: thieno[2,3-d]-pyrimidine derivatives Org41841, Org43553, TP03, TP04 and their analogs (allosteric LHR-agonists or ago-PAMs, predominantly activate G_s_-proteins), the derivatives of 1,3,5-pyrazole (compound 1) and terphenyl (LUF5771) (allosteric LHR-agonists), and thieno[2,3-d]-pyrimidine derivative TP31 (allosteric LHR-antagonists). Pepducin NKDTKIAKK-Nle-A(562–572)-K(Palm)A (ICL3 of LHR) interacts with the intracellular allosteric site and functions as an allosteric LHR agonist. Autoantibodies to the extracellular regions of LHR have been suggested but have not been characterized to date. Allosteric regulators that increase receptor activity (full agonists and ago-PAMs) are placed in red squares, while allosteric regulators with neutral antagonist activity are placed in green squares.

**Table 1 ijms-24-06187-t001:** The classification of allosteric regulators of GPCRs.

Allosteric Ligand	Pharmacological Characteristics	α	β	τ
Positive allosteric modulator (PAM)	It increases the affinity and/or efficacy of an orthosteric agonist but has no intrinsic activity.	>1	>1	1 *
Negative allosteric modulator (NAM)	It reduces the affinity and/or efficacy of an orthosteric agonist but has no intrinsic activity.	<1	<1	1
Silent allosteric modulator (SAM)	It does not affect the affinity and efficacy of an orthosteric agonist but is able to change some other characteristics of its interaction with the GPCR (selectivity for different types of G proteins and β-arrestins; specific activation of intracellular target proteins; etc.).	1	1	1
Full allosteric agonist	It stimulates the GPCR in the absence of an orthosteric agonist and does not affect the affinity and efficacy of an orthosteric agonist.	1	1	>1
Full allosteric agonist/PAM (ago-PAM)	It functions as a full agonist and at the same time enhances the affinity and/or efficacy of an orthosteric agonist.	>1	>1	>1
Full allosteric agonist/NAM (ago-NAM)	It functions as a full agonist and at the same time reduces the affinity and/or efficacy of an orthosteric agonist.	<1	<1	>1
Inverse allosteric agonist	It reduces the intrinsic activity of the GPCR (basal or constitutively active) in the absence of an orthosteric agonist.	1	1	<1
Neutral allosteric antagonist	It prevents the activation of the GPCR by an orthosteric agonist, including due to stabilization of its inactive state.	1	1	<1
Neutral allosteric antagonist/PAM	It reduces the effectiveness of an orthosteric agonist, acting as an antagonist, but at the same time increases the affinity of an orthosteric agonist to the receptor functioning as a PAM.	>1	<1	1

Notes: α, the factor of binding cooperativity between the orthosteric agonist and allosteric modulator; β, the operational factor of cooperativity for quantitative evaluation of the effects of allosteric modulator on operational efficacy of orthosteric agonist (receptor activation); τ, operational efficacy for the complex of GPCR with allosteric ligand. Values of binding cooperativity α and operational cooperativity β greater than 1 denote positive cooperativity, and corresponding values below 1 denote negative cooperativity. * The value “one” for operational efficacy τ is normalized since the absolute τ value may vary depending on the used calculated parameters.

**Table 2 ijms-24-06187-t002:** Localization of allosteric sites in the GPCR and examples of ligands that are able to specifically bind to these sites.

	The Structural (Sub)Domain	Function of The (Sub)Domain	Types of Endogenous and Artificial Allosteric Regulators
I	The extracellular loops, including a large ectodomain in some GPCRs, and the external entrance (vestibule) to the transmembrane tunnel.	The recognition of the orthosteric ligands, the participation in receptor di- and oligomerization (especially for class C GPCRs), the targets for N-glycosylation, and, in the GPCRs of glycoprotein and peptide hormones, the localization of high-affinity othosteric site.	Peptides; low-molecular compounds; auto-GPCR antibodies; GPCR protomers that form homo- or heterodi(oligo)meric complexes; RAMPs.
II	In the internal cavity of the transmembrane domain, including the upper (below the external entrance), central, and lower (above the internal entrance) parts of the seven-helix transmembrane bundle.	The localization of high-affinity othosteric site in the GPCRs that are activated by small ligands, the main molecular determinants responsible for conformational changes in the process of receptor activation or its transition to an inactive state.	Low molecular weight compounds; simple ions.
III	The outer-lateral surface of the transmembrane domain.	Interaction with the lipid phase of membranes, participation in complex formation, including di- and oligomerization of GPCRs, and maintenance of the conformation of the seven-helix transmembrane bundle suitable for effective signal transduction.	Membrane lipids, including cholesterol and phosphoinositids, and, possibly, other highly hydrophobic, membrane-associated compounds; transmembrane helix of RAMPs.
IV	The intracellular loops and in their interfaces with transmembrane regions.	The specific interaction with different types of heterotrimeric G proteins and β-arrestins, the targets for GRK-mediated phosphorylation, and the formation of multicomponent signaling complexes.	Heterotrimeric G proteins; β-arrestins; cell-penetrating GPCR-derived peptides (pepducins); low-molecular-weight compounds (parmodulins, etc.).

## Data Availability

Not applicable.

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
