# Peer review of "Allosteric Regulation of G-Protein-Coupled Receptors: From Diversity of Molecular Mechanisms to Multiple Allosteric Sites and Their Ligands"

_ijms, 2023, doi:10.3390/ijms24076187_

Round 1
Reviewer 1 Report
The manuscript by Shpakov AO is focused on the fields of GPCR singling, in particular, the mechanism of allosteric regulation of GPCRs. Examples of several GPCRs which are well-documented mechanisms of allosteric modulation are soundly demonstrated.
This review is excellently described and summarized. Also, the reviewer hopes the review contributes to the readers of IJMS for a better understanding of up-to-date allosteric GPCR regulation theories.
Author Response
RESPONSE TO REVIEWERS
“Allosteric regulation of G-protein-coupled receptors: from diversity of molecular mechanisms to multiple allosteric sites and their ligands” (Alexander O. Shpakov)
COMMON SECTION OF RESPONSE TO REVIEWERS
I am very grateful to the Reviewers for a detailed analysis of the review article “Allosteric regulation of G-protein-coupled receptors: from diversity of molecular mechanisms to multiple allosteric sites and their ligands” and for the comments made. I sincerely hope that the explanation, additions and changes I made based on their comments and remarks have improved the review article.
In accordance with the requirements of the reviewers, I significantly revised the review article and made the following main changes and additions (for more details, the changes and additions are presented in the extended Response to comments of the Reviewers #1, #2, #3 and #4).
- Some sections, most notably the Introduction, have been substantially reorganized and parts of them rewritten. To improve the clarity of the content of sections 2-4, a figure (Figure 1 in Section 4) and two tables (Table 1 in Section 2, and Table 2 in Section 3) are introduced into the text of the manuscript. Due to changes made to the manuscript, the list of cited publications was revised.
- 2. In accordance with the recommendations of the reviewers, the changes were made to the parts of the manuscript discussing various aspects of beta-arrestin signaling in the GPCR-systems. An alternative point of view is presented that the signaling functions of beta-arrestins, at least for a number of GPCRs, require the presence of different types of G-proteins. The information about the targets of carvedilol, a beta2-AR blocker, has been clarified. The information on the localization of beta-arrestin-binding sites in the cytoplasmic domains of GPCRs has been clarified, and redundant and contradictory information on the beta-arrestin conformations has been deleted. As recommended by Reviewer #2, clarifications have been made regarding the classification and designation of beta-arrestins, and corresponding corrections are made throughout the text.
- 3. In accordance with the requirements of the Reviewers, the information on the effect of di(oligo)merization on the functional activity of GPCR has been changed and refined. It was emphasized that this effect is most typical for class C GPCRs, while the role of di(oligo)merization for the other GPCRs is not fully established. At the same time, data are presented demonstrating that heterodimerization of some class A GPCRs can affect binding characteristics, agonist-induced stimulation, and signaling bias. These changes and additions are made to different parts of the manuscript, most notably to the Section 5 and the Conclusion.
- In accordance with the comment of Reviewer #4, the English language has been improved and some errors and unfortunate phrases have been corrected.
All changes and additions to the text and References section are highlighted in yellow, and the changed reference numbers in the manuscript are colored green.
In conclusion, I once again thank the Reviewers for their comments, which allowed us to clarify many aspects of the review article, and I hope that the changes made will be accepted and improve the review article.
With best regards, Alexander Shpakov
Reviewer: 1
COMMON COMMENTS
The manuscript by Shpakov AO is focused on the fields of GPCR singling, in particular, the mechanism of allosteric regulation of GPCRs. Examples of several GPCRs which are well-documented mechanisms of allosteric modulation are soundly demonstrated.
This review is excellently described and summarized. Also, the reviewer hopes the review contributes to the readers of IJMS for a better understanding of up-to-date allosteric GPCR regulation theories.
Response: I am grateful to the reviewer for the high appreciation of my work.

Reviewer 2 Report
The goal of the Review Article "Allosteric regulation of G-protein-coupled receptors: from diversity of molecular mechanisms to multiple allosteric sites and their ligands by Alexander O. Shpakov is to describe the allosteric regulation of various classes of GPCRs, the variety of allosteric sites, their location and function. Moreover, the author describes multiple regulators and modulators interacting with these sites. The author describes mainly the allosteric regulation of chemokine receptors, proteinase-activated receptors, thyroid-stimulating and luteinizing hormone receptors, and beta-adrenergic receptors. The idea of a Review is interesting, and the general schedule of the article content is very promising. My main criticism of this Article is related to the manuscript length, proportions among specific chapters and reference content.
Specific comment:
1. The introduction content till line 184 is unrelated to the goal of the Review stated in lines 185-190. I suggest authors significantly shorten section #1 to the minimum necessary to introduce readers to the article content. Please consider also moving the historical descriptions of GPCRs to a separate section with the appropriate title.
2. If the author cannot remove/shorten mentioned text, I would recommend rephrasing/correcting some unclear sentences in the Introduction section:
a) Line 35- Can you specify when in evolution GPCRs systems occurred?
b) Line 62- 66 - the conformations of beta-arrestins "tail" and "core" are not clear from the manuscript, while authors introduce their names to explain differences in their interaction with GPCR. Please rephrase or describe deeper conformations, tail and core.
c) Lines 69-71: "different types of arrestin, b1 and b2,..." is not precise. How many "different" types of arrestins are known? Only "beta1 and beta2? Explain classification and add literature with details for interested readers.
d) Please add literature to the sentence in lines 110 - 113 about the primary donor of beta gamma from Gi/o activation.
e) Line 116, remove redundancy from the phrase: "see more details, see ..."
3. Instead of clearly classifying allosteric GPCR regulators, as the author suggests in the title of the chapter 2, the reader can find lengthy text as the author's summary of other cited reviews: [50, 72-80]. I recommend significantly shortening this chapter to the most critical information to understand the rest of the Review Article. The Table with precise classification and related literature in a separate column would help understand current classification.
4. In chapter 3, the information in lines 259-269 is repeated from the introduction. Please remove redundancy. Moreover, I have the same criticism as I described in point 3. Please consider reorganizing the content to help readers understand this chapter. The Table with localization, the number of allosteric sites in GPCRs and information about the methodology which served to discover these places together with appropriate references would help.
5. I suggest the author reorganize Chapters 4-5 to let readers easily understand chapter 4 - Diversity of Endogenous Allosteric Regulators of GPCRs, and in chapter 5 - GPCR-complexes and allosteric regulation. I suggest shortening to minimum information not related directly to the content of each chapter. Thematic subsections or Tables with described data would be beneficial.
6. I also have the general comment for review articles cited in the manuscript. Typically, Review Articles collect and confront conclusions from published original research to summarize a particular study area. Sparing citation of review articles to provide information on where out-of-scope knowledge may be acquired is acceptable. However, in the current Article, the authors cite many review articles (163, which is 36.5% of all citations). Please, use appropriate references in the corrected manuscript version to convince readers that the Article is a reliable summary of source research on the topic in question.
Author Response
RESPONSE TO REVIEWERS
“Allosteric regulation of G-protein-coupled receptors: from diversity of molecular mechanisms to multiple allosteric sites and their ligands” (Alexander O. Shpakov)
COMMON SECTION OF RESPONSE TO REVIEWERS
I am very grateful to the Reviewers for a detailed analysis of the review article “Allosteric regulation of G-protein-coupled receptors: from diversity of molecular mechanisms to multiple allosteric sites and their ligands” and for the comments made. I sincerely hope that the explanation, additions and changes I made based on their comments and remarks have improved the review article.
In accordance with the requirements of the reviewers, I significantly revised the review article and made the following main changes and additions (for more details, the changes and additions are presented in the extended Response to comments of the Reviewers #1, #2, #3 and #4).
- Some sections, most notably the Introduction, have been substantially reorganized and parts of them rewritten. To improve the clarity of the content of sections 2-4, a figure (Figure 1 in Section 4) and two tables (Table 1 in Section 2, and Table 2 in Section 3) are introduced into the text of the manuscript. Due to changes made to the manuscript, the list of cited publications was revised.
- 2. In accordance with the recommendations of the reviewers, the changes were made to the parts of the manuscript discussing various aspects of beta-arrestin signaling in the GPCR-systems. An alternative point of view is presented that the signaling functions of beta-arrestins, at least for a number of GPCRs, require the presence of different types of G-proteins. The information about the targets of carvedilol, a beta2-AR blocker, has been clarified. The information on the localization of beta-arrestin-binding sites in the cytoplasmic domains of GPCRs has been clarified, and redundant and contradictory information on the beta-arrestin conformations has been deleted. As recommended by Reviewer #2, clarifications have been made regarding the classification and designation of beta-arrestins, and corresponding corrections are made throughout the text.
- 3. In accordance with the requirements of the Reviewers, the information on the effect of di(oligo)merization on the functional activity of GPCR has been changed and refined. It was emphasized that this effect is most typical for class C GPCRs, while the role of di(oligo)merization for the other GPCRs is not fully established. At the same time, data are presented demonstrating that heterodimerization of some class A GPCRs can affect binding characteristics, agonist-induced stimulation, and signaling bias. These changes and additions are made to different parts of the manuscript, most notably to the Section 5 and the Conclusion.
- In accordance with the comment of Reviewer #4, the English language has been improved and some errors and unfortunate phrases have been corrected.
All changes and additions to the text and References section are highlighted in yellow, and the changed reference numbers in the manuscript are colored green.
In conclusion, I once again thank the Reviewers for their comments, which allowed us to clarify many aspects of the review article, and I hope that the changes made will be accepted and improve the review article.
With best regards, Alexander Shpakov
Reviewer: 2
COMMON COMMENTS
The goal of the Review Article "Allosteric regulation of G-protein-coupled receptors: from diversity of molecular mechanisms to multiple allosteric sites and their ligands by Alexander O. Shpakov is to describe the allosteric regulation of various classes of GPCRs, the variety of allosteric sites, their location and function. Moreover, the author describes multiple regulators and modulators interacting with these sites. The author describes mainly the allosteric regulation of chemokine receptors, proteinase-activated receptors, thyroid-stimulating and luteinizing hormone receptors, and beta-adrenergic receptors. The idea of a Review is interesting, and the general schedule of the article content is very promising. My main criticism of this Article is related to the manuscript length, proportions among specific chapters and reference content.
Response: Thank you very much for the positive evaluation of my work. In line with critical comments, efforts have been made to improve the organization and content of the review (see below). At the same time, I have kept the length of the manuscript, since all its parts contribute to the general idea of complex allosteric regulation of GPCRs. The presence of various allosteric mechanisms, the multiplicity of allosteric sites, and the diversity of endogenous allosteric regulators indicate the great importance of allosteric regulation for the functioning of GPCRs and their signaling pathways. It can be assumed that the signal induced by an orthosteric agonist is a kind of “button” for triggering signal transduction, and allosteric regulators play a decisive role in the implementation of this effect and in the final response of the target cell. In this regard, a comprehensive consideration of the problem is very important, which is done in the "general" sections. In accordance with your comments (see below), to improve the perception of sections 2-4, a Figure (Section 4) and two tables (sections 2 and 3) have been added to them, the Introduction (section 1) and Conclusion (section 11) have been significantly changed.
- The introduction content till line 184 is unrelated to the goal of the Review stated in lines 185-190. I suggest authors significantly shorten section #1 to the minimum necessary to introduce readers to the article content. Please consider also moving the historical descriptions of GPCRs to a separate section with the appropriate title.
Response: Thank you very much for your comment. The Introduction has been significantly changed, including in accordance with the questions presented in comment 2, as well as in accordance with the comments of Reviewers #3 and #4 (see response to comment 2, and the Common Section of Response to Reviewers). It seems to me that in the new version, the Introduction logically precedes information on allosteric regulation. After the changes made, the historical aspect moved to the background, so that in the modified version there is no need to separate it into a separate section.
- If the author cannot remove/shorten mentioned text, I would recommend rephrasing/correcting some unclear sentences in the Introduction section:
2-a) Line 35- Can you specify when in evolution GPCRs systems occurred?
Response: Thank you very much for your comment. In the first paragraph, I have included information about the evolution of GPCRs, as well as their representation in various taxa. The revised and supplemented paragraph is presented below. In addition, I have pointed out hybrid structures that are functional and structural homologues of classical sensory kinases in bacteria. These structures are interesting in that in the case of a combination of the GPCR and phosphatidylinositol phosphate kinases, after the binding of the receptor to the ligand, the kinase is activated and catalyzes the synthesis of phosphoinositides (PI(4,5)P2), the important membrane components and universal signaling molecules. These phosphoinositides are also allosteric regulators of GPCRs. This information is added to the Section 4 where the role of lipids as allosteric GPCR regulators is discussed.
In the Introduction.
“G protein-coupled receptors (GPCRs), located in the plasma membrane, are the largest superfamily of receptor (sensory) proteins in multicellular eukaryotes. GPCRs have been found in fungi [1-3], plants [4], and in all studied invertebrates and vertebrates [5-9], including trypanosomes [10] and ciliates [11]. At the same time, the yeast Saccharomyces cerevisiae has only three genes encoding GPCRs [12], the slime mold Dictyostelium discoideum has 55 such genes [13], while in the mammalian genome there are more than 800 genes for GPCRs [14]. Prototypes of the structural domains of both GPCRs and the adapter and regulatory proteins that interact with them appeared at the earliest stages of evolution, already at the level of prokaryotes and unicellular eukaryotes [2, 9, 15]. During the early evolution of GPCR, different structural models of these receptors existed, including hybrid constructs that consisted of an N-terminal GPCR-like molecule and a C-terminal catalytic phosphatidylinositol phosphate kinase, which were identified in some representatives of lower eukaryotes [11, 16, 17].”
In the Section 4.
“Along with cholesterol, other lipids such as phosphatidylserines [89] and phosphoinositides, including PI(4,5)P2 [90], are also involved in the allosteric regulation of GPCRs. As noted above, some evolutionarily ancient forms of GPCRs were hybrids of GPCR and phosphatidylinositol phosphate kinase, catalyzing the synthesis of PI(4,5)P2 [11, 16, 17]. As a result, it can be assumed that this phosphoinositide could already then function as an allosteric modulator of GPCRs.”
2-b) Line 62- 66 - the conformations of beta-arrestins "tail" and "core" are not clear from the manuscript, while authors introduce their names to explain differences in their interaction with GPCR. Please rephrase or describe deeper conformations, tail and core.
Response: Thank you very much for your comment. In accordance with the recommendations of other reviewers, the discussion of the conformational features of bera-arrestins was excluded from the manuscript, as controversial (in terms of the effect on signaling) and redundant.
2-c) Lines 69-71: "different types of arrestin, b1 and b2,..." is not precise. How many "different" types of arrestins are known? Only "beta1 and beta2? Explain classification and add literature with details for interested readers.
Response: Thank you very much for your remark and for the suggestion to provide data on the classification of beta-arrestins. Four types of beta-arrestins are currently known. These are two retinal beta-arrestins, such as visual arrestin (arrestin1) and cone arrestin (arrestin4), and two more widespread nonvisual arrestins, β-arrestin1 (arrestin2) and β-arrestin2 (arrestin3). In accordance with the ambiguity of designations of arrestins β-arrestin1 (arrestin2) and β-arrestin2 (arrestin3), there are certain difficulties and inconsistencies in their mention. In this regard, both in the fragment of the text under discussion and in other parts of the manuscript, we designated beta-arrestins according to the generally accepted system as β-arrestin1 and β-arrestin2. Necessary additions and clarifications have been made to the text. The modified text is shown below.
“β-Arrestins are an evolutionarily ancient family of structurally and functionally related scaffolding proteins, which in vertebrates includes two retinal β-arrestins and two non-visual β-arrestins, β-arrestin1 (arrestin2) and β-arrestin2 (arrestin3), the latter being widely distributed in various tissues and are of great importance for GPCR-mediated signaling transduction [25, 26].”
2-d) Please add literature to the sentence in lines 110 - 113 about the primary donor of beta gamma from Gi/o activation.
Response: Thank you very much for your remark.
Different types of heterotrimeric G-proteins can be donors of the beta-gamma-dimer, but, according to the works of 1980-2000, pertussis toxin-sensitive Gi/o-proteins are of the greatest importance for the generation of free beta-gamma-dimer. There are several studies showing this, which were summarized in a review by Smrcka (2008) (the Section "Pertussis toxin-sensitive signaling by G protein βγ subunits"). Pertussis toxin ADP-ribosyltransferase modifies the C-terminal segment of the alpha subunit, preventing its functional interaction with the receptor and, thereby, interrupting signal transmission through the G-protein to intracellular effectors and preventing the dissociation of the alpha-beta-gamma-heterotrimeric G-protein. Appropriate additions have been made to the text (the reference [Smrcka, 2008]).
2-e) Line 116, remove redundancy from the phrase: "see more details, see ..."
Response: The error has been corrected.
- Instead of clearly classifying allosteric GPCR regulators, as the author suggests in the title of the chapter 2, the reader can find lengthy text as the author's summary of other cited reviews: [50, 72-80]. I recommend significantly shortening this chapter to the most critical information to understand the rest of the Review Article. The Table with precise classification and related literature in a separate column would help understand current classification.
Response: Thank you very much for your comments and suggestion to reorganize the Section 2. In accordance with these comments, I have prepared and introduced into the text a table with a classification and brief characteristics of the main types of allosteric modulators and regulators (Table 1). Along with this, other parts of this section have been significantly changed. The paragraph on the description of the ternary complex and the assessment of the pharmacological profile of allosteric regulators based on alpha, beta and tau factors has been completely revised. The modified fragments of section 2 are given below.
Corresponding to lines 192-216 in the original version:
“According to the ability to influence the basal and orthosteric/allosteric agonist-stimulated activity, the ligands of GPCR allosteric sites can be divided into allosteric modulators that have no intrinsic activity and allosteric regulators that affect GPCR activity in the absence of orthosteric agonists [56]. In the case of allosteric modulators, the ligand, by binding to the allosteric site, changes or retains unchanged the affinity of the orthosteric agonist to GPCR and/or its ability to activate the receptor, which is assessed by its maximum stimulating effect, but has no intrinsic activity (Table 1). Allosteric ligands that have their intrinsic activity can function as full agonists, inverse agonists and neutral antagonists, and their action is independent of orthosteric site occupancy (Table 1). Such independence of the action of allosteric ligands can be realized only when the orthosteric and allosteric sites do not overlap and do not interact through ligand-induced conformational rearrangements [56]. When an allosteric ligand acts as a full agonist and affects the affinity and/or potency of an orthosteric agonist, it is classified as ago-PAM or ago-NAM (Table 1). In the case when allosteric ligand reduces the effectiveness of an orthosteric agonist, but increases its affinity to GPCR, it is classified as a PAM-antagonist [75].”
Corresponding to lines 234-248 in the original version:
“Given the variety of regulatory influences of allosteric ligands on the interaction of orthosteric agonists with GPCR, in the recent years a model has been widely used that describes a ternary complex that includes GPCR and receptor-bound allosteric and orthosteric ligands. The formation of a ternary complex is described by equations A+R+B <->AR+B (KA) <-> ARB (KB/α) in the case when, at the first stage, the GPCR forms a complex with the orthosteric agonist A, and by equations A+R+B <-> A+RB (KB) <-> ARB (KA/α), when, at the first stage, the receptor forms a complex with the allosteric ligand B (KA and KB, the equilibrium dissociation constants for the GPCR-orthosteric agonist and GPCR-allosteric ligand complexes, respectively; α, the factor of binding cooperativity between the orthosteric agonist A and allosteric ligand B) [78]. Therefore, the effect of an allosteric ligand on the affinity of an orthosteric agonist is described by the factor α. In turn, its effect on the efficacy (maximum regulatory effect) of an orthosteric agonist is described by the factor β. When the allosteric ligand increases the affinity and efficacy of orthosteric agonist, the factors α and β are above 1 (PAM) (Table 1). If the influence of the allosteric ligand is the opposite, then these factors have values below 1 (NAM). In the absence of a significant effect, the α and β are equal to 1 (SAM). To assess the intrinsic activity of allosteric ligands, the factor τ is used, which for full agonists has values above 1 (full agonist, ago-PAM, and ago-NAM), and for an inverse agonist or neutral antagonist it has values below 1. For “pure” allosteric modulators (PAM, NAM, SAM, PAM-antagonist), the factor τ is equal to 1 (Table 1). However, for each specific case and for each specific “allosteric ligand–orthosteric ligand” pair, the values of α, β, and τ may vary.”
- In chapter 3, the information in lines 259-269 is repeated from the introduction. Please remove redundancy. Moreover, I have the same criticism as I described in point 3. Please consider reorganizing the content to help readers understand this chapter. The Table with localization, the number of allosteric sites in GPCRs and information about the methodology which served to discover these places together with appropriate references would help.
Response: Thank you very much for your comment. Some of the redundant information from the paragraph (lines 259-269 in the original version) that was mentioned earlier has been removed.
In accordance with the recommendations, Table 2 is prepared. Table 2 describes the possible localization of allosteric sites in four different structural subdomains of the receptor and provides information on the functional importance of these subdomains for the functional activity of the receptors, This table also gives the main classes of allosteric regulators that typically bind to allosteric sites in these subdomains.
The modified fragments of section 3 are given below.
Corresponding to lines 259-269 in the original version:
“Allosteric sites can be localized in all structural domains of the GPCRs (Table 2). Estimation of the number and localization of allosteric sites in each receptor is a very difficult task, although significant progress in this direction has been made due to the development of new approaches for the identification of GPCR allosteric sites [59, 60, 82, 83].”
- I suggest the author reorganize Chapters 4-5 to let readers easily understand chapter 4 - Diversity of Endogenous Allosteric Regulators of GPCRs, and in chapter 5 - GPCR-complexes and allosteric regulation. I suggest shortening to minimum information not related directly to the content of each chapter. Thematic subsections or Tables with described data would be beneficial.
Response: Thank you very much for your comment. Substantial changes have been made to sections 4 and 5, section 4 is divided into two subsections. To improve the perception of the multiplicity of allosteric regulators, as well as the possible role of receptor dimerization in it, an additional figure was prepared and introduced into the text of the manuscript - Figure 1 (in section 4).
- I also have the general comment for review articles cited in the manuscript. Typically, Review Articles collect and confront conclusions from published original research to summarize a particular study area. Sparing citation of review articles to provide information on where out-of-scope knowledge may be acquired is acceptable. However, in the current Article, the authors cite many review articles (163, which is 36.5% of all citations). Please, use appropriate references in the corrected manuscript version to convince readers that the Article is a reliable summary of source research on the topic in question.
Response: Thank you very much for your comment. I have carried out a thorough revision of references, including reviews. As a result, more than 30 references were excluded from the Reference section. At the same time, the additions made to the manuscript (the evolution of GPCRs, the classification of beta-arrestins, the features of dimerization of various classes of GPCRs, etc.) made it necessary to add more than 20 references, as a result of which their total number in the review decreased by 10 references only. All references have been renumbered, and the new numbers are highlighted in green in the text of the manuscript. New references in the References section are colored yellow.

Reviewer 3 Report
This is a very thorough review about allosteric modulators with a focus on chemokine receptors, proteinase-activated receptors, TSH, LH, FSH and beta-adrenergic receptors which are discussed in much detail. For other important GPCRs (e.g., muscarinic acetylcholine receptors) it refers to recent reviews.
My main objection to the review as it is written currently is the way it deals with signalling bias. It has been shown already some years ago that there is probably no exclusively arrestin-mediated signal generation by GPCRs (O’Hayre et al. (2017) Genetic evidence that beta-arrestins are dispensable for the initiation of beta2-adrenergic receptor signaling to ERK. Sci Signal 10(484); Grundmann et al. (2018) Lack of beta-arrestin signaling in the absence of active G proteins. Nat Commun 9(1): 341) and that the role of arrestins is mostly to scaffold the components of MAP kinase cascades. This concerns, for example, the section between lines 46 and 71, and the discussion about carvedilol-mediated signalling through beta2-adrenergic receptors between lines 1479 and 1490. It has recently been shown that carvedilol most likely mediates its actions at the beta2-adrenergic receptor, including ERK phosphorylation, through Gs-proteins rather than arrestins (Benkel T, Zimmermann M, Zeiner J, Bravo S, Merten N, Lim VJY, et al. (2022). How Carvedilol activates b2-adrenoceptors. Nat Commun 13(1): 7109.).
Some further comments:
Lines 75-78: There is a large number of GPCRs that are phosphorylated by GRKs only in the C-terminus but are internalized nevertheless. The AT1 receptor that is used by the author to support his model (Ref. 33) does not carry any phosphorylation sites whatsoever in its first and third ICL, and the single serine in its second ICL would not be sufficient to mediate arrestin binding to the receptor (it has been demonstrated by several authors that one phosphorylation site is not sufficient to achieve arrestin binding). Of course, there are some GPCRs that are phosphorylated by GRKs on intracellular loops (for example the muscarinic acetylcholine receptors) because they have large ICLs containing lots of Ser/Thr residues in the appropriate context and only a short C-terminus.
Lines 77-78: “… although this IS not a general rule.”
Line 141: It cannot be generally said that orthosteric agonists bind their receptors with high affinity. This very much depends on the nature of the agonist and receptor. For example, a number of peptides bind to their cognate receptor with nanomolar affinity (e.g., oxytocin to the oxytocin receptor), but carboxylic acids bind to the FFA receptors with high micromolar or even millimolar affinity (e.g., propanoic acid to the FFA2 receptor).
Line 234-248: A drawing of the model presented in Ref75 would be useful.
Line 449-450: The author is probably talking about “phosphatidylinositols and phosphatidylserines” rather than “phosphoinositols and phosphoserines” (the latter are no lipids).
Line 484-540: This section claims that receptor dimerization exerts very important allosteric effects at GPCRs. However, the examples given by the author (mGluR, GABABR, CaSR and taste receptors) are all class C GPCRs which are known to form obligate stable dimers. While it is clear that some class A and class B GPCRs form dimers, it is entirely unclear whether all GPCRs do this (there is some evidence for at least some GPCRs being entirely monomeric), how stable these dimers (there are a number of reports suggesting that these dimers are rather transient in nature) are and whether dimerization has any effect on ligand binding or effector coupling. The argumentation would at least be strengthened if examples from GPCR classes other than C were included.
Line 980: use “docetaxel” instead of the brand name “Taxotere”.
Line 1107: the TSH receptor is, according to the GPCRdb, a class A receptor, not a class C receptor.
Author Response
RESPONSE TO REVIEWERS
“Allosteric regulation of G-protein-coupled receptors: from diversity of molecular mechanisms to multiple allosteric sites and their ligands” (Alexander O. Shpakov)
COMMON SECTION OF RESPONSE TO REVIEWERS
I am very grateful to the Reviewers for a detailed analysis of the review article “Allosteric regulation of G-protein-coupled receptors: from diversity of molecular mechanisms to multiple allosteric sites and their ligands” and for the comments made. I sincerely hope that the explanation, additions and changes I made based on their comments and remarks have improved the review article.
In accordance with the requirements of the reviewers, I significantly revised the review article and made the following main changes and additions (for more details, the changes and additions are presented in the extended Response to comments of the Reviewers #1, #2, #3 and #4).
- Some sections, most notably the Introduction, have been substantially reorganized and parts of them rewritten. To improve the clarity of the content of sections 2-4, a figure (Figure 1 in Section 4) and two tables (Table 1 in Section 2, and Table 2 in Section 3) are introduced into the text of the manuscript. Due to changes made to the manuscript, the list of cited publications was revised.
- 2. In accordance with the recommendations of the reviewers, the changes were made to the parts of the manuscript discussing various aspects of beta-arrestin signaling in the GPCR-systems. An alternative point of view is presented that the signaling functions of beta-arrestins, at least for a number of GPCRs, require the presence of different types of G-proteins. The information about the targets of carvedilol, a beta2-AR blocker, has been clarified. The information on the localization of beta-arrestin-binding sites in the cytoplasmic domains of GPCRs has been clarified, and redundant and contradictory information on the beta-arrestin conformations has been deleted. As recommended by Reviewer #2, clarifications have been made regarding the classification and designation of beta-arrestins, and corresponding corrections are made throughout the text.
- 3. In accordance with the requirements of the Reviewers, the information on the effect of di(oligo)merization on the functional activity of GPCR has been changed and refined. It was emphasized that this effect is most typical for class C GPCRs, while the role of di(oligo)merization for the other GPCRs is not fully established. At the same time, data are presented demonstrating that heterodimerization of some class A GPCRs can affect binding characteristics, agonist-induced stimulation, and signaling bias. These changes and additions are made to different parts of the manuscript, most notably to the Section 5 and the Conclusion.
- In accordance with the comment of Reviewer #4, the English language has been improved and some errors and unfortunate phrases have been corrected.
All changes and additions to the text and References section are highlighted in yellow, and the changed reference numbers in the manuscript are colored green.
In conclusion, I once again thank the Reviewers for their comments, which allowed us to clarify many aspects of the review article, and I hope that the changes made will be accepted and improve the review article.
With best regards, Alexander Shpakov
Reviewer: 3
COMMON COMMENTS
This is a very thorough review about allosteric modulators with a focus on chemokine receptors, proteinase-activated receptors, TSH, LH, FSH and beta-adrenergic receptors which are discussed in much detail. For other important GPCRs (e.g., muscarinic acetylcholine receptors) it refers to recent reviews.
Response: I am grateful to the reviewer for the high appreciation of my work, including in relation to certain types of GPCRs.
MAJOR REMARKS
- My main objection to the review as it is written currently is the way it deals with signalling bias. It has been shown already some years ago that there is probably no exclusively arrestin-mediated signal generation by GPCRs (O’Hayre et al. (2017) Genetic evidence that beta-arrestins are dispensable for the initiation of beta2-adrenergic receptor signaling to ERK. Sci Signal 10(484); Grundmann et al. (2018) Lack of beta-arrestin signaling in the absence of active G proteins. Nat Commun 9(1): 341) and that the role of arrestins is mostly to scaffold the components of MAP kinase cascades. This concerns, for example, the section between lines 46 and 71, and the discussion about carvedilol-mediated signalling through beta2-adrenergic receptors between lines 1479 and 1490. It has recently been shown that carvedilol most likely mediates its actions at the beta2-adrenergic receptor, including ERK phosphorylation, through Gs-proteins rather than arrestins (Benkel T, Zimmermann M, Zeiner J, Bravo S, Merten N, Lim VJY, et al. (2022). How Carvedilol activates b2-adrenoceptors. Nat Commun 13(1): 7109).
Response:
Comment 1A. Thank you very much for your comments regarding beta-arrestin-mediated signaling. Indeed, at present there are different points of view regarding the independence of beta-arrestins as signal transducers. I analyzed in detail the very interesting works recommended by you, which provide data that beta-arrestin is not capable of carrying out signal transduction in the absence of all classes of G-proteins. This significantly changes the concept of G-protein-independent signaling through beta-arrestins.
In accordance with this, the paragraph concerning beta-arrestin-mediated signaling has been significantly changed, and the additional references (Saulière et al., 2012; Strachan et al., 2014; O'Hayre et al., 2017; Grundmann et al., 2018; No 25-28) have been introduced into it, indicating the need for coordination and functional interaction of G-proteins and beta-arrestins during signal transduction (primarily during MAPK activation). Information about the conformational features of beta arrestins has been removed because it is redundant.
As recommended by Reviewer #2, clarifications have been made regarding the classification and designation of beta-arrestins. Corresponding corrections regarding the universal designations of beta-arrestin isoforms are made throughout the text of the manuscript.
The new version of the paragraph (lines 46-71 in the original version) is shown below.
“Over the past two decades, many paradigms regarding the functioning of GPCRs and the transduction of hormonal signals through them have undergone revision. In the 1990s, it was generally accepted that signal transduction from the hormone-activated GPCR occurs almost exclusively through the heterotrimeric G proteins. However, later evidence was obtained that various adapter and regulatory proteins, primarily β-arrestins, which are able to interact specifically with the hormone-activated receptor, are also involved in signal transduction, thereby regulating and modulating GPCR-mediated intracellular signaling [20-24]. β-Arrestins are an evolutionarily ancient family of structurally and functionally related scaffolding proteins, which in vertebrates includes two retinal β-arrestins and two non-visual β-arrestins, β-arrestin1 (arrestin2) and β-arrestin2 (arrestin3), the latter being widely distributed in various tissues and are of great importance for GPCR-mediated signaling transduction [25, 26]. In early studies of the role of β-arrestins in the GPCR-signaling, it was recognized that these proteins are responsible for the desensitization of GPCRs, facilitating the dissociation of the G-protein α-subunit from the activated receptor and mediating endocytosis of the ligand-receptor complex, which leads to the degradation of this complex in the proteasomes or to recycling of the ligand-free GPCR back to the plasma membrane. At the turn of 1990-2000, the participation of β-arrestins in GPCR-mediated regulation of the mitogen-activated protein kinases (MAPKs) and several other effector proteins and transcription factors was demonstrated [21, 24, 27-33]. This has changed the paradigm of the exclusive role of the heterotrimeric G-proteins as signal transducers in the GPCR-signaling At the same time, the question of whether β-arrestins are able to carry out signal transduction independently of G-proteins remains open to date. In recent years, evidence has been obtained that in the absence of G-proteins, β-arrestins are unable to activate MAPKs, which indicates the need for coordinated participation of G-proteins and β-arrestins in signal transduction and does not support the concept of G-protein-independent β-arrestin signaling [34-37]. It is also shown that β-arrestins are able to modulate the interaction of GPCRs with different types of G-proteins, as is observed in the case of the type 1 parathyroid hormone receptor (PTH1R) coupled to both Gs- and Gq/11-proteins [38].”
Comment 1B. Based on the results of the work of Benkel and co-authors, published at the end of 2022, the necessary changes were made to the discussion of the influence of the allosteric modulator Cmpd-6 on the regulatory effects of the beta2-AR blocker carvedilol. This is an extremely interesting study that significantly changes the focus on the molecular mechanisms of the cardioprotective effect of carvedilol. A corresponding reference to the work of Benkel et al. (2022) (Reference No 427) has been added to the text. The new version of the text (lines 1479-1490 in the original version) is shown below.
“This compound interfered with the effects of the β-AR blocker carvedilol. Cmpd-6, on the one hand, increased the affinity of carvedilol for β2-AR and thereby enhanced its inhibitory effect on β2-agonist-induced stimulation of Gs-proteins and cAMP signaling, and, on the other hand, enhanced the stimulatory effects of carvedilol on ERK1/2 activity, endocytosis of β2-ARs and their trafficking into lysosomes [416], which, as was shown later, is also due to the activation of Gs-proteins [417].”
Some further comments:
- Lines 75-78: There is a large number of GPCRs that are phosphorylated by GRKs only in the C-terminus but are internalized nevertheless. The AT1 receptor that is used by the author to support his model (Ref. 33) does not carry any phosphorylation sites whatsoever in its first and third ICL, and the single serine in its second ICL would not be sufficient to mediate arrestin binding to the receptor (it has been demonstrated by several authors that one phosphorylation site is not sufficient to achieve arrestin binding). Of course, there are some GPCRs that are phosphorylated by GRKs on intracellular loops (for example the muscarinic acetylcholine receptors) because they have large ICLs containing lots of Ser/Thr residues in the appropriate context and only a short C-terminus.
Response: Thank you very much for your comment. I fully agree that a significant part of the sites for phosphorylation that are targets of beta-arrestins are located in different loci of the cytoplasmic C-terminal domain, which in a number of GPCRs has a significant size and many sites that are targets for receptor-specific protein kinases. There are many works that show the leading role of the C-terminal domain of GPCRs in their GRK-mediated phosphorylation and interaction with beta-arrestins. This inaccuracy in the text has been corrected. In the sentence corresponding to lines 72-74, the phrase "and the cytoplasmic C-terminal domain" is added, and in the sentence corresponding to lines 75-78, "ICLs" is replaced by "intracellular regions".
- Lines 77-78: “… although this IS not a general rule.”
Response: Thank you very much for your comment. A correction has been made to the text.
- Line 141: It cannot be generally said that orthosteric agonists bind their receptors with high affinity. This very much depends on the nature of the agonist and receptor. For example, a number of peptides bind to their cognate receptor with nanomolar affinity (e.g., oxytocin to the oxytocin receptor), but carboxylic acids bind to the FFA receptors with high micromolar or even millimolar affinity (e.g., propanoic acid to the FFA2 receptor).
Response: Thank you very much for your comment. The revised version of the paragraph (lines 141-151 in the original manuscript) clarified that the affinity of an orthosteric site for ligands is not always higher than that of allosteric sites of the receptor. A modified version of this paragraph is presented below.
“As is known, the site of the receptor to which its endogenous ligand specifically binds is designated as orthosteric. Depending on the class of receptors, it can be located within the transmembrane domain (TMD) near the extracellular entrance to the transmembrane tunnel, in the extracellular loops (ECLs) or in the large extracellular domain of GPCR. Binding of an orthosteric agonist to a receptor typically results in a significant stimulating effect, inducing receptor activation. In most cases, the affinity of orthosteric ligands for GPCR is higher than that of allosteric ligands, although this is not a general rule. Compared with orthosteric agonists, interaction of the receptor with allosteric ligands generally results in more moderate and selective effects on basal GPCR activity and also modulates receptor activity stimulated by orthosteric agonists. The GPCR contains not one, but several allosteric sites that differ in localization, configuration, and functional activity. These sites also differ in their influence on the conformation and accessibility of the orthosteric site, the efficiency of the interaction with transducer and regulatory proteins and, for some receptor types, on the ability of GPCR to form homo- and heterooligomeric receptor complexes [55-60].”
- Line 234-248: A drawing of the model presented in Ref75 would be useful.
Response: Many thanks for the comment, which allowed to change the paragraph concerning both the formation of the ternary complex and the alpha, beta and tau coefficients describing the effects of allosteric modulators and regulators. Part of the information is placed in Table 1, which presents all the main types of allosteric regulators. A detailed scheme for the formation of a ternary complex (according to Reference 75, in the new numbering Reference 78) with all the necessary notation is presented in the text, which is equivalent to the presentation of the corresponding figure. The modified text is below (corresponds to fragment 234-248 in the original version).
“Given the variety of regulatory influences of allosteric ligands on the interaction of orthosteric agonists with GPCR, in the recent years a model has been widely used that describes a ternary complex that includes GPCR and receptor-bound allosteric and orthosteric ligands. The formation of a ternary complex is described by equations A+R+B <->AR+B (KA) <-> ARB (KB/α) in the case when, at the first stage, the GPCR forms a complex with the orthosteric agonist A, and by equations A+R+B <-> A+RB (KB) <-> ARB (KA/α), when, at the first stage, the receptor forms a complex with the allosteric ligand B (KA and KB, the equilibrium dissociation constants for the GPCR-orthosteric agonist and GPCR-allosteric ligand complexes, respectively; α, the factor of binding cooperativity between the orthosteric agonist A and allosteric ligand B) [78]. Therefore, the effect of an allosteric ligand on the affinity of an orthosteric agonist is described by the factor α. In turn, its effect on the efficacy (maximum regulatory effect) of an orthosteric agonist is described by the factor β. When the allosteric ligand increases the affinity and efficacy of orthosteric agonist, the factors α and β are above 1 (PAM) (Table 1). If the influence of the allosteric ligand is the opposite, then these factors have values below 1 (NAM). In the absence of a significant effect, the α and β are equal to 1 (SAM). To assess the intrinsic activity of allosteric ligands, the factor τ is used, which for full agonists has values above 1 (full agonist, ago-PAM, and ago-NAM), and for an inverse agonist or neutral antagonist it has values below 1. For “pure” allosteric modulators (PAM, NAM, SAM, PAM-antagonist), the factor τ is equal to 1 (Table 1). However, for each specific case and for each specific “allosteric ligand–orthosteric ligand” pair, the values of α, β, and τ may vary.”
- Line 449-450: The author is probably talking about “phosphatidylinositols and phosphatidylserines” rather than “phosphoinositols and phosphoserines” (the latter are no lipids).
Response: Thank you very much for your comment. A correction has been made to the text (corresponding sentence has been changed).
- Line 484-540: This section claims that receptor dimerization exerts very important allosteric effects at GPCRs. However, the examples given by the author (mGluR, GABABR, CaSR and taste receptors) are all class C GPCRs which are known to form obligate stable dimers. While it is clear that some class A and class B GPCRs form dimers, it is entirely unclear whether all GPCRs do this (there is some evidence for at least some GPCRs being entirely monomeric), how stable these dimers (there are a number of reports suggesting that these dimers are rather transient in nature) are and whether dimerization has any effect on ligand binding or effector coupling. The argumentation would at least be strengthened if examples from GPCR classes other than C were included.
Response: Many thanks for the very valuable and important remarks for the improvement of section 5 regarding the role of di- and oligomerization in the regulation of various classes of GPCRs. Indeed, most of the data presented relates to class C GPCRs, which includes metabotropic glutamate receptors, calcium-sensing receptors, gamma-aminobutyric acid receptors, and taste receptors. This is due to the structural and regulatory features of class C GPCRs. However, there is evidence that oligomerization is important for the functioning and regulation of other classes of GPCRs, primarily the receptors of the most represented class A. This is summarized in a review by Milligan et al. (Milligan G, Ward RJ, Marsango S. GPCR homo-oligomerization Curr Opin Cell Biol 2019 Apr;57:40-47 doi: 10.1016/j.ceb.2018.10.007 Epub 2018 Nov 16 PMID: 30453145; PMCID: PMC7083226). Excerpt from the review below:
«The numerically predominant rhodopsin-like, or Class A, GPCRs are generally described as monomers. However, evidence emerging from a broad range of approaches has shown that they can form both dimers and higher-order oligomers with protomers of either the same receptor (homo-dimers/oligomers) or with partners of the same sub family and even with GPCRs which respond to different ligands (hetero-dimers/oligomers). Many Class A GPCRs are able to form dimers and oligomers. Although this may be transient in many situations, at physiological expression levels dimer and oligomers may represent a substantial population. Growing data indicate the extent and kinetics of such quaternary complexes are regulated by ligand binding and this may have marked significance for the action of therapeutic medicines».
Many works analyze the allosteric effects of heterodimerization among class A GPCRs. It seems to me that allosteric effects occur regardless of whether different (hetero(di)oligomerization) or the same (homo(di)oligomerization) protomers form a receptor complex, although their molecular mechanisms differ. Recent studies include the following:
1) Erol I, Cosut B, Durdagi S. Toward Understanding the Impact of Dimerization Interfaces in Angiotensin II Type 1 Receptor. J Chem Inf Model. 2019 Oct 28;59(10):4314-4327. doi: 10.1021/acs.jcim.9b00294. Epub 2019 Sep 13. PMID: 31429557.
2) Patrone M, Cammarota E, Berno V, Tornaghi P, Mazza D, Degano M. Combinatorial allosteric modulation of agonist response in a self-interacting G-protein coupled receptor. Commun Biol. 2020 Jan 15;3(1):27. doi: 10.1038/s42003-020-0752-4. PMID: 31941999; PMCID: PMC6962373. (it is show that S1PR1 oligomers are required for full response to different agonists and ligand-specific association with arrestins, dictating the downstream signalling kinetics)
3) Aso E, Fernández-Dueñas V, López-Cano M, Taura J, Watanabe M, Ferrer I, Luján R, Ciruela F. Adenosine A2A-Cannabinoid CB1 Receptor Heteromers in the Hippocampus: Cannabidiol Blunts Δ9-Tetrahydrocannabinol-Induced Cognitive Impairment. Mol Neurobiol. 2019 Aug;56(8):5382-5391. doi: 10.1007/s12035-018-1456-3. Epub 2019 Jan 4. PMID: 30610611.
4) Viñals X, Moreno E, Lanfumey L, Cordomí A, Pastor A, de La Torre R, Gasperini P, Navarro G, Howell LA, Pardo L, Lluís C, Canela EI, McCormick PJ, Maldonado R, Robledo P. Cognitive Impairment Induced by Delta9-tetrahydrocannabinol Occurs through Heteromers between Cannabinoid CB1 and Serotonin 5-HT2A Receptors. PLoS Biol. 2015 Jul 9;13(7):e1002194. doi: 10.1371/journal.pbio.1002194. PMID: 26158621; PMCID: PMC4497644.
Confirmation and study of (di)oligomerization of class A GPCRs is of great importance for the development of bivalent ligands, including those with an allosteric model of receptor regulation (see Hiller C, Kühhorn J, Gmeiner P. Class A G-protein-coupled receptor (GPCR) dimers and bivalent ligands. J Med Chem. 2013 Sep 12;56(17):6542-59. doi: 10.1021/jm4004335. Epub 2013 Jun 4. PMID: 23678887.)
At the same time, taking into account the insufficient validity of statements about the universality of the effect of oligomerization for GPCRs that do not belong to class C, I have made appropriate changes to the text of the manuscript.
In addition, a paragraph has been added to the Section 5 regarding the dimerization of class A GPCRs, which emphasizes both the limitations and the importance of heterodimerization (as opposed to the preferred homodimerization for class C GPCRs).
“Despite the fact that the data obtained mainly relate to class C GPCRs, di- and oligomerization affects the functional activity of other classes of GPCRs, including the most representative class A [153], although the data in this case are not always so unambiguous. There are a number of recent studies showing that heterodimerization of class A receptors affects their binding characteristics and the efficiency of agonist activation [154-157] and intracellular signaling bias [155], and also modulates physiological responses under the conditions of agonist-induced stimulation of GPCR [158, 159]. Homodimerization, as shown for the type 1 angiotensin II receptor, can also affect the functional activity of the class A GPCRs [160]. Like class C, in most cases the effect of dimerization on the activity of class A GPCRs is due to allosteric mechanisms.”
- Line 980: use “docetaxel” instead of the brand name “Taxotere”.
Response: Thank you very much for your comment. The text has been changed.
- Line 1107: the TSH receptor is, according to the GPCRdb, a class A receptor, not a class C receptor.
Response: Thank you very much for your comment, the error has been corrected (the first line in the comment to Figure 3, line 1107 in the original version). Of course, the TSH receptor belongs to the family of rhodopsin-like receptors (class A, subfamily A7).

Reviewer 4 Report
The author focuses on allosteric regulation of GPCRs. While the review contains a significant amount of useful data regarding several GPCRs, the general part has too many incorrect statements (some, but not all of which are pointed out below). It needs to be comprehensively rewritten. In particular, the authors lumps together as “allosteric regulators” small molecules, membrane lipids, and downstream signaling proteins. In addition, the review is too long.
1. Lines 61-66. The “tail” binding of arrestins has not been demonstrated to any wild type GPCRs.
2. Lines 69-71. The conformation of any b-arrestin bound to PTHR was never elucidated.
3. Line 73. Many GPCRs have phosphorylation sites relevant for b-arrestin binding in the C-terminus, not in the ICLs.
4. Line 79. The term used is “barcode”, not “phosphcode”.
5. Lines 85-92. Why b-arrestins dissociate from internalized GPCRs is not actually known. Different ideas were formulated, none tested experimentally.
6. Line 117. GPCR is not in several conformation at once (a physical impossibility), but in equilibrium among several conformations.
7. Lines 141-145. Agonists that bind where the endogenous agonist binds are called orthosteric. E.g., orthosteric agonists of calss C GPCRs bind to extracellular VFT domain, whereas their allosteric ligands often bind where orthosteric ligands of class A (rhodopsin-like) GPCRs bind.
8. Line 146. Allosteric and orthosteric ligands can have a wide range of affinities. E.g., orthosteric agonist dopamine binds some dopamine receptors with remarkably low affinity.
9. Line 150 and below, especially section 5. While class C GPCRs are dimers, oligomerization of class A GPCRs is controversial, particularly because the methods used to detect putative oligomers have severe caveats, which signaling outcomes can often be explained by cross-talk of signaling pathways downstream of GPCRs.
10. Line 426 and below. Concentration of extracellular sodium does not change in vivo, so GPCRs either have bound sodium ion or don’t.
11. Lines 351-353. GTP-liganded Ga and free Gbg do not interact with GPCRs, and therefore cannot affect their affinity for agonists. Bound empty heterotrimeric G protein and b-arrestins increase GPCR affinity for agonists.
12. Extensive editing is needed, preferably by a native speaker. Including, but not limited to: lines 30-31, “the most representative and widespread” should be “the largest”; besides, this is not true of plants or Dictyostelium; line 51, evidence has no plural; etc. (too many places to point them all out).
Author Response
RESPONSE TO REVIEWERS
“Allosteric regulation of G-protein-coupled receptors: from diversity of molecular mechanisms to multiple allosteric sites and their ligands” (Alexander O. Shpakov)
COMMON SECTION OF RESPONSE TO REVIEWERS
I am very grateful to the Reviewers for a detailed analysis of the review article “Allosteric regulation of G-protein-coupled receptors: from diversity of molecular mechanisms to multiple allosteric sites and their ligands” and for the comments made. I sincerely hope that the explanation, additions and changes I made based on their comments and remarks have improved the review article.
In accordance with the requirements of the reviewers, I significantly revised the review article and made the following main changes and additions (for more details, the changes and additions are presented in the extended Response to comments of the Reviewers #1, #2, #3 and #4).
- Some sections, most notably the Introduction, have been substantially reorganized and parts of them rewritten. To improve the clarity of the content of sections 2-4, a figure (Figure 1 in Section 4) and two tables (Table 1 in Section 2, and Table 2 in Section 3) are introduced into the text of the manuscript. Due to changes made to the manuscript, the list of cited publications was revised.
- 2. In accordance with the recommendations of the reviewers, the changes were made to the parts of the manuscript discussing various aspects of beta-arrestin signaling in the GPCR-systems. An alternative point of view is presented that the signaling functions of beta-arrestins, at least for a number of GPCRs, require the presence of different types of G-proteins. The information about the targets of carvedilol, a beta2-AR blocker, has been clarified. The information on the localization of beta-arrestin-binding sites in the cytoplasmic domains of GPCRs has been clarified, and redundant and contradictory information on the beta-arrestin conformations has been deleted. As recommended by Reviewer #2, clarifications have been made regarding the classification and designation of beta-arrestins, and corresponding corrections are made throughout the text.
- 3. In accordance with the requirements of the Reviewers, the information on the effect of di(oligo)merization on the functional activity of GPCR has been changed and refined. It was emphasized that this effect is most typical for class C GPCRs, while the role of di(oligo)merization for the other GPCRs is not fully established. At the same time, data are presented demonstrating that heterodimerization of some class A GPCRs can affect binding characteristics, agonist-induced stimulation, and signaling bias. These changes and additions are made to different parts of the manuscript, most notably to the Section 5 and the Conclusion.
- In accordance with the comment of Reviewer #4, the English language has been improved and some errors and unfortunate phrases have been corrected.
All changes and additions to the text and References section are highlighted in yellow, and the changed reference numbers in the manuscript are colored green.
In conclusion, I once again thank the Reviewers for their comments, which allowed us to clarify many aspects of the review article, and I hope that the changes made will be accepted and improve the review article.
With best regards, Alexander Shpakov
Reviewer: 4
COMMON COMMENTS
The author focuses on allosteric regulation of GPCRs. While the review contains a significant amount of useful data regarding several GPCRs, the general part has too many incorrect statements (some, but not all of which are pointed out below). It needs to be comprehensively rewritten. In particular, the authors lumps together as “allosteric regulators” small molecules, membrane lipids, and downstream signaling proteins. In addition, the review is too long.
Response: Thank you very much for the critical analysis of my review and the valuable comments made, which were all taken into account by me. Regarding the representation in one series of allosteric regulators of different nature and mechanisms of action, I would like to justify that in the relevant sections, attention is focused on the fact that these regulators function in a different way, interact with allosteric sites of different localization and lead to different effects. At the same time, I completely agree with the reviewer that the proximity of sodium ions and heterotrimeric G-proteins is perceived ambiguously. In this regard, I have made some remarks to the relevant places in the text (for example, Section 4, see below).
«It should be noted that the molecular mechanisms and their targets in receptor molecules differ significantly, but their regulatory effects are based on allosteric effects on the basal and orthosteric agonist-stimulated activity of various receptor classes».
In addition, the new Figure 1 in Section 4 also highlights the significant differences between the various allosteric regulators.
MAJOR REMARKS
- Lines 61-66. The “tail” binding of arrestins has not been demonstrated to any wild type GPCRs.
Response: Thank you very much for your comment. Indeed, there are no convincing data on the detection of the tail conformation of beta-arrestin in the complex with wild-type receptors, which makes the conclusion about the functional selectivity of various beta-arrestin conformations highly debatable. It should be noted that in the new version of the manuscript, the paragraph concerning beta-arrestin signaling has been significantly revised in accordance with the recommendations of the Reviewer #3. It takes into account an alternative point of view, which is that beta-arrestins, at least for a number of receptors, cannot transmit a hormonal signal in the absence of G-proteins. As a result, the problem of G-protein-independent arrestin signaling in the GPСR-systems still remains open and requires further study (this is reflected in the new version of the paragraph corresponding to paragraph 47-71 in the original version). Since information about the various conformations of beta arrestins and supporting references became redundant in the new context, it was removed from the text.
As recommended by Reviewer #2, clarifications have been made regarding the classification and designation of beta-arrestins. Corresponding corrections regarding the universal designations of beta-arrestin isoforms are made throughout the text of the manuscript.
The new version of the paragraph is presented below.
“Over the past two decades, many paradigms regarding the functioning of GPCRs and the transduction of hormonal signals through them have undergone revision. In the 1990s, it was generally accepted that signal transduction from the hormone-activated GPCR occurs almost exclusively through the heterotrimeric G proteins. However, later evidence was obtained that various adapter and regulatory proteins, primarily β-arrestins, which are able to interact specifically with the hormone-activated receptor, are also involved in signal transduction, thereby regulating and modulating GPCR-mediated intracellular signaling [20-24]. β-Arrestins are an evolutionarily ancient family of structurally and functionally related scaffolding proteins, which in vertebrates includes two retinal β-arrestins and two non-visual β-arrestins, β-arrestin1 (arrestin2) and β-arrestin2 (arrestin3), the latter being widely distributed in various tissues and are of great importance for GPCR-mediated signaling transduction [25, 26]. In early studies of the role of β-arrestins in the GPCR-signaling, it was recognized that these proteins are responsible for the desensitization of GPCRs, facilitating the dissociation of the G-protein α-subunit from the activated receptor and mediating endocytosis of the ligand-receptor complex, which leads to the degradation of this complex in the proteasomes or to recycling of the ligand-free GPCR back to the plasma membrane. At the turn of 1990-2000, the participation of β-arrestins in GPCR-mediated regulation of the mitogen-activated protein kinases (MAPKs) and several other effector proteins and transcription factors was demonstrated [21, 24, 27-33]. This has changed the paradigm of the exclusive role of the heterotrimeric G-proteins as signal transducers in the GPCR-signaling At the same time, the question of whether β-arrestins are able to carry out signal transduction independently of G-proteins remains open to date. In recent years, evidence has been obtained that in the absence of G-proteins, β-arrestins are unable to activate MAPKs, which indicates the need for coordinated participation of G-proteins and β-arrestins in signal transduction and does not support the concept of G-protein-independent β-arrestin signaling [34-37]. It is also shown that β-arrestins are able to modulate the interaction of GPCRs with different types of G-proteins, as is observed in the case of the type 1 parathyroid hormone receptor (PTH1R) coupled to both Gs- and Gq/11-proteins [38].”
- Lines 69-71. The conformation of any b-arrestin bound to PTHR was never elucidated.
Response: Thank you very much for your comment. Indeed, the paper (Haider et al., 2022) suggests that different preferred conformations of types 1 and 2 beta-arrestins may mediate their role in signal transduction, which is logical but not experimentally confirmed. Since the discussion of the conformational features of beta-arrestins is excluded from the new version of paragraph 47-71 (see Response to Comment 1), the question of the conformations of beta1- and beta2-arrestins when binding to the parathyroid hormone receptor is also not discussed in it. At the same time, the influence of the conformational features of different types of beta-arrestins on their signaling functions is of great interest as a large independent problem.
- Line 73. Many GPCRs have phosphorylation sites relevant for b-arrestin binding in the C-terminus, not in the ICLs.
Response: Thank you very much for your comment. I fully agree that a significant part of the sites for phosphorylation that are targets of beta-arrestins are located in different loci of the cytoplasmic C-terminal domain, which in a number of GPCRs has a significant size and many sites that are targets for receptor-specific protein kinases. There are many works that show the leading role of the C-terminal domain of GPCRs in their GRK-mediated phosphorylation and interaction with beta-arrestins. This inaccuracy in the text has been corrected. In the sentence corresponding to lines 72-74, the phrase "and the cytoplasmic C-terminal domain" is added, and in the sentence corresponding to lines 75-78, "ICLs" is replaced by "intracellular regions".
- Line 79. The term used is “barcode”, not “phosphcode”.
Response: Thank you very much for your comment. The text has been changed.
- Lines 85-92. Why b-arrestins dissociate from internalized GPCRs is not actually known. Different ideas were formulated, none tested experimentally.
Response: Thank you very much for your comment. The corresponding fragment (corresponding to lines 84-92 in the original manuscript) has been changed. It emphasizes that there are various factors that influence the dynamics of the formation of the receptor-beta-arrestin complex, and they have not been fully investigated. In this fragment, emphasis is placed on the content of phosphoninositides in the plasma membrane, as one of such factors. At the same time, it was noted that only for a part of the receptors, the dependence of the stability of the receptor-beta-arrestin complex on the content of phosphoinositides was shown. The modified text is below.
The corresponding fragment has been changed. It emphasizes that there are various factors affecting the dynamics of the formation of the receptor-beta-arrestin complex, and they have not been fully investigated, and also focuses on the content of phosphoninositides in the plasma membrane, as one of such factors. At the same time, it was noted that only for a part of the receptors, the dependence of the stability of the receptor-beta-arrestin complex on the content of phosphoinositides was shown. The modified text (corresponding to lines 84-92 in the original manuscript) is below.
“The factors that influence the recruitment of β-arrestins, the dynamics of the formation of their complex with GPCR and the further dissociation of β-arrestins from the internalized receptor complex are currently being intensively studied. It is assumed that for some types of GPCRs, the lipid composition of the plasma membrane, including the content of phosphoinositides, is important for the implementation of these processes. For these GPCRs, phosphoinositides, including phosphatidylinositol-4,5-bisphosphate (PI(4,5)P2), are required to form the functionally active GPCR–β-arrestin complexes [44]. A decrease in their content in the membrane during endocytosis leads to dissociation of the GPCR–β-arrestin complexes and recycling of the free receptor into the plasma membrane. On the other hand, after GRK phosphorylation, some types of GPCRs do not require phosphoinositides for the formation of such complexes, which remain stable during endocytosis and continue to perform signaling functions in endosomes, where phosphoinositides content are significantly reduced [44].”
- Line 117. GPCR is not in several conformation at once (a physical impossibility), but in equilibrium among several conformations.
Response: Thank you very much for your comment. The phrase (lines 116-118 in the original manuscript) has been changed to emphasize that it is about an equilibrium between different active conformations (the new version is given below).
This is due to the fact that GPCR is able to exist in several conformations that are in dynamic equilibrium and are quite close in energy characteristics, but each of which mediates the activation of a certain type of G-protein or β-arrestin.
- Lines 141-145. Agonists that bind where the endogenous agonist binds are called orthosteric. E.g., orthosteric agonists of calss C GPCRs bind to extracellular VFT domain, whereas their allosteric ligands often bind where orthosteric ligands of class A (rhodopsin-like) GPCRs bind.
- Line 146. Allosteric and orthosteric ligands can have a wide range of affinities. E.g., orthosteric agonist dopamine binds some dopamine receptors with remarkably low affinity.
Response: I am grateful for the valuable comments, which allowed for a significant revision of the paragraph concerning the description of the orthosteric site and its comparison with allosteric sites (lines 141-151 in the original manuscript). All comments of the reviewer have been taken into account and incorporated into the revised text. A modified version of this paragraph is presented below.
“As is known, the site of the receptor to which its endogenous ligand specifically binds is designated as orthosteric. Depending on the class of receptors, it can be located within the transmembrane domain (TMD) near the extracellular entrance to the transmembrane tunnel, in the extracellular loops (ECLs) or in the large extracellular domain of GPCR. Binding of an orthosteric agonist to a receptor typically results in a significant stimulating effect, inducing receptor activation. In most cases, the affinity of orthosteric ligands for GPCR is higher than that of allosteric ligands, although this is not a general rule. Compared with orthosteric agonists, interaction of the receptor with allosteric ligands generally results in more moderate and selective effects on basal GPCR activity and also modulates receptor activity stimulated by orthosteric agonists. The GPCR contains not one, but several allosteric sites that differ in localization, configuration, and functional activity. These sites also differ in their influence on the conformation and accessibility of the orthosteric site, the efficiency of the interaction with transducer and regulatory proteins and, for some receptor types, on the ability of GPCR to form homo- and heterooligomeric receptor complexes [55-60].”
- Line 150 and below, especially section 5. While class C GPCRs are dimers, oligomerization of class A GPCRs is controversial, particularly because the methods used to detect putative oligomers have severe caveats, which signaling outcomes can often be explained by cross-talk of signaling pathways downstream of GPCRs.
Response: Thanks for the very important remarks that the key role of dimerization (including allosteric regulatory mechanisms) has been shown mainly for class C receptors, while for other classes of receptors, including the most represented class A, the data are not always unambiguous. Indeed, most of the data presented relates to class C GPCRs, which includes metabotropic glutamate receptors, calcium-sensing receptors, gamma-aminobutyric acid receptors, and taste receptors. This is due to the structural and regulatory features of class C GPCRs.
Based on this, some of the emphasis regarding di- and oligomerization has been changed in various parts of the manuscript, including lines 150-151 (see Response to the Comments 7 and 8 above).
However, there is evidence that oligomerization is important for the functioning and regulation of other classes of GPCRs, primarily the receptors of the most represented class A. This is summarized in a review by Milligan et al. (Milligan G, Ward RJ , Marsango S. GPCR homo-oligomerization Curr Opin Cell Biol 2019 Apr;57:40-47 doi: 10.1016/j.ceb.2018.10.007 Epub 2018 Nov 16 PMID: 30453145; PMCID: PMC7083226). Excerpt from the review below:
«The numerically predominant rhodopsin-like, or Class A, GPCRs are generally described as monomers. However, evidence emerging from a broad range of approaches has shown that they can form both dimers and higher-order oligomers with protomers of either the same receptor (homo-dimers/oligomers) or with partners of the same sub family and even with GPCRs which respond to different ligands (hetero-dimers/oligomers). Many Class A GPCRs are able to form dimers and oligomers. Although this may be transient in many situations, at physiological expression levels dimer and oligomers may represent a substantial population. Growing data indicate the extent and kinetics of such quaternary complexes are regulated by ligand binding and this may have marked significance for the action of therapeutic medicines».
There are many works that analyze the effects of heterodimerization among class A receptors, including allosteric. It seems to me that allosteric effects occur regardless of whether different (hetero(di)oligomerization) or the same (homo(di)oligomerization) protomers form a receptor complex, although their molecular mechanisms certainly differ. Recent studies include the following:
1) Erol I, Cosut B, Durdagi S. Toward Understanding the Impact of Dimerization Interfaces in Angiotensin II Type 1 Receptor. J Chem Inf Model. 2019 Oct 28;59(10):4314-4327. doi: 10.1021/acs.jcim.9b00294. Epub 2019 Sep 13. PMID: 31429557.
2) Patrone M, Cammarota E, Berno V, Tornaghi P, Mazza D, Degano M. Combinatorial allosteric modulation of agonist response in a self-interacting G-protein coupled receptor. Commun Biol. 2020 Jan 15;3(1):27. doi: 10.1038/s42003-020-0752-4. PMID: 31941999; PMCID: PMC6962373. (it is show that S1PR1 oligomers are required for full response to different agonists and ligand-specific association with arrestins, dictating the downstream signalling kinetics)
3) Aso E, Fernández-Dueñas V, López-Cano M, Taura J, Watanabe M, Ferrer I, Luján R, Ciruela F. Adenosine A2A-Cannabinoid CB1 Receptor Heteromers in the Hippocampus: Cannabidiol Blunts Δ9-Tetrahydrocannabinol-Induced Cognitive Impairment. Mol Neurobiol. 2019 Aug;56(8):5382-5391. doi: 10.1007/s12035-018-1456-3. Epub 2019 Jan 4. PMID: 30610611.
4) Viñals X, Moreno E, Lanfumey L, Cordomí A, Pastor A, de La Torre R, Gasperini P, Navarro G, Howell LA, Pardo L, Lluís C, Canela EI, McCormick PJ, Maldonado R, Robledo P. Cognitive Impairment Induced by Delta9-tetrahydrocannabinol Occurs through Heteromers between Cannabinoid CB1 and Serotonin 5-HT2A Receptors. PLoS Biol. 2015 Jul 9;13(7):e1002194. doi: 10.1371/journal.pbio.1002194. PMID: 26158621; PMCID: PMC4497644.
Confirmation and study of (di)oligomerization of class A GPCRs is of great importance for the development of bivalent ligands, including those with an allosteric model of receptor regulation (see Hiller C, Kühhorn J, Gmeiner P. Class A G-protein-coupled receptor (GPCR) dimers and bivalent ligands. J Med Chem. 2013 Sep 12;56(17):6542-59. doi: 10.1021/jm4004335. Epub 2013 Jun 4. PMID: 23678887.)
At the same time, taking into account the insufficient validity of statements about the universality of the effect of oligomerization for GPCRs that do not belong to class C, I have made appropriate changes to the text of the manuscript.
In addition, a paragraph has been added to section 5 regarding the dimerization of class A GPCRs, which emphasizes both the limitations and the importance of heterodimerization (as opposed to the preferred homodimerization for class C GPCRs).
“Despite the fact that the data obtained mainly relate to class C GPCRs, di- and oligomerization affects the functional activity of other classes of GPCRs, including the most representative class A [153], although the data in this case are not always so unambiguous. There are a number of recent studies showing that heterodimerization of class A receptors affects their binding characteristics and the efficiency of agonist activation [154-157] and intracellular signaling bias [155], and also modulates physiological responses under the conditions of agonist-induced stimulation of GPCR [158, 159]. Homodimerization, as shown for the type 1 angiotensin II receptor, can also affect the functional activity of the class A GPCRs [160]. Like class C, in most cases the effect of dimerization on the activity of class A GPCRs is due to allosteric mechanisms.”
- Line 426 and below. Concentration of extracellular sodium does not change in vivo, so GPCRs either have bound sodium ion or don’t.
Response: Many thanks for the comment, which prompted us to change the description of sodium ion-mediated allosteric regulation of GPCRs (see below).
The sentence “Sodium ions bind to an allosteric site…” (Lines 426-428) has been changed to emphasize the ability of sodium ions to allosterically affect the activity of certain types of receptors. New version: “Sodium ions are able to bind to the allosteric site of these receptors located in the internal cavity of their TMD. This site, a target for sodium ions, includes several highly conserved amino acid residues, the most important of which is the negatively charged aspartic acid located in TM2”.
The sentence "The effects of sodium ions on GPCR activity are carried out at their physiological concentrations, which indicates the involvement of Na+ in the regulation of GPCR signaling in real biological systems” (Lines 430-432) has been removed based on reviewer's comment. It should be noted that in the original version, this sentence only emphasized that sodium ions are capable of influencing the activity of GPCRs at the concentrations (physiological) that are present in living systems.
In addition, I would like to draw attention to the fact that in the sentence “Such receptor-nonspecific activity…” (Lines 423-425) it was noted that sodium ions are negative allosteric modulators for a large number of receptors, but they are not universal allosteric regulators, which is meaningless from the point of view of physiological regulation.
- Lines 351-353. GTP-liganded Ga and free Gbg do not interact with GPCRs, and therefore cannot affect their affinity for agonists. Bound empty heterotrimeric G protein and b-arrestins increase GPCR affinity for agonists.
Response: Thank you very much for your comment. Early results indicate a negative effect of GTP and non-hydrolysable GTP analogues, which cause the activation and dissociation of heterotrimeric G-proteins, on agonist affinity for the GPCRs. At the same time, I agree with the reviewer that in this case it is not entirely correct to speak about the allosteric effect of the GTP-bound Galpha-subunit and free Gbeta-gamma-dimer on agonist affinity, given the absence of their direct interaction with the receptor.
In accordance with this, the disputed fragment 352-356 has been deleted from the text, and the corresponding references No. 101 and 102 have also been removed (see below).
«The GTP-bound Gα-subunit and Gβγ-dimer allosterically affect the conformation of the orthosteric site and reduce its affinity for agonists, downregulating signal transduction. The first evidence of this was obtained almost half a century ago, in experiments on the effect of GTP and its non-hydrolysable analogs on the binding of agonists to β-AR [101, 102], and later were confirmed for other receptors [73]. At the same time,…»
- Extensive editing is needed, preferably by a native speaker. Including, but not limited to: 1) lines 30-31, “the most representative and widespread” should be “the largest”; besides, this is not true of plants or Dictyostelium; 2) line 51, evidence has no plural; etc. (too many places to point them all out).
Response: The text has been edited, identified errors or unsuccessful phrases have been corrected (everything is marked in yellow).
12-1) The first sentence in the Introduction (lines 30-32) has been modified to take into account the reviewer's comment that GPCRs in plants and fungi are not the dominant receptors and are usually represented by a small number of them. For example, the slime fungus Dictyostelium discoideum has 55 GPCRs genes, the yeast Saccharomyces cerevisiae has three GPCRs genes, while mammals have more than 800 GPCRs genes.
New version: “G protein-coupled receptors (GPCRs), located in the plasma membrane, are the largest superfamily of receptor (sensory) proteins in multicellular eukaryotes.”
Also, in accordance with the recommendations of reviewer #2, some information on the evolution of GPCRs has been added, which expands the data on the presence of these receptors in various taxa of unicellular and multicellular eukaryotes.
12-2) The word "evidence" (line 51) in the revised version of the paragraph on beta arrestins is in the singular.

Round 2
Reviewer 4 Report
While the manuscript was improved in revision, several factually wrong statements remain. In particular:
1. Line 36. The author calims that mammalian genome has 800 GPCRs. In fact, human genome has ~800 GPCRs. In different mammals this number varies widely, from ~600 in anosmatic cetaceans to >3,000 in elephants.
2. Line 61. Neither of the two visual arrestin subtypes was ever called b-arrestin.
3. Line 65. Arrestins (neither visual nor b-arrestin) were ever implicated in facilitating dissociation of G protein a-subunit from the GPCR. Arrestins compete with heterotrimeric G proteins for active GPCRs and win this competition when GPCR is active and phosphorylated at the same time.
4. Line 67. GPCRs are integral membrane proteins. They cannot be degraded by proteasomes: proteasomes work only with soluble proteins. GPCRs, like all membrane proteins, are degraded by specialized lysosomes (often called multi-vesicular bodies, as vesicles containing membrane proteins are inside the lysosome).
5. Lines 221, 1580-1581. PAMs and NAMs are opposites: PAMs facilitate signaling, whereas NAMs suppress it (as correctly described below line 221). There cannot be such a thing as “ago-NAM”.
6. Table 2, item IV, and Fig. 1. RAMPs bind transmembrane helices of GPCRs.
7. Section 5. While the author extensively cites papers suggesting the existence of class A GPCR oligomers, equally numerous papers presenting opposite evidence are not cited and discussed. The studies demonstrating that many reported changes in signaling can be just as easily explained by well-established cross-talk of signaling pathways downstream of GPCRs were not mentioned or cited.
Author Response
RESPONSE TO REVIEWER #4 (the second round)
“Allosteric regulation of G-protein-coupled receptors: from diversity of molecular mechanisms to multiple allosteric sites and their ligands” (Alexander O. Shpakov)
I am very grateful to the Reviewer for a detailed repeated analysis of the review article “Allosteric regulation of G-protein-coupled receptors: from diversity of molecular mechanisms to multiple allosteric sites and their ligands” and for the additional comments made. I sincerely hope that the changes I made based on their comments have improved the review article.
In accordance with the comments made, the necessary changes were made to the text (highlighted in yellow) and clarifications were made, which are presented below in a detailed response to all comments.
With best regards, Alexander Shpakov
COMMON COMMENTS:
While the manuscript was improved in revision, several factually wrong statements remain. In particular:
Comment 1. Line 36. The author claims that mammalian genome has 800 GPCRs. In fact, human genome has ~800 GPCRs. In different mammals this number varies widely, from ~600 in anosmatic cetaceans to >3,000 in elephants.
Response: Thank you very much for your very valuable comment. The necessary correction has been made: the term "mammals" (line 36) has been replaced by "human". At the same time, indeed, the fact of great diversity and quantitative differences of receptors in the mammalian genome is extremely interesting. This is largely due to differences in the number of odorant receptors. These receptors (mostly belonging to the GPCR superfamily) are abundant in some rodents and have a record high abundance in the African elephant (Niimura et al., 2014), while they are significantly less in aquatic mammals. In whales, their number is about 600 (Kishida et al., 2007; Liu et al., 2019). Since this information is not directly related to the topic of the review, it was not included in the manuscript, but is of great interest and will be analyzed and used in the future.
Kishida T, Kubota S, Shirayama Y, Fukami H. The olfactory receptor gene repertoires in secondary-adapted marine vertebrates: evidence for reduction of the functional proportions in cetaceans. Biol Lett. 2007 Aug 22;3(4):428-30. doi: 10.1098/rsbl.2007.0191. PMID: 17535789; PMCID: PMC2390674.
Liu A, He F, Shen L, Liu R, Wang Z, Zhou J. Convergent degeneration of olfactory receptor gene repertoires in marine mammals. BMC Genomics. 2019 Dec 12;20(1):977. doi: 10.1186/s12864-019-6290-0. PMID: 31842731; PMCID: PMC6916060.
Niimura Y, Matsui A, Touhara K. Extreme expansion of the olfactory receptor gene repertoire in African elephants and evolutionary dynamics of orthologous gene groups in 13 placental mammals. Genome Res. 2014 Sep;24(9):1485-96. doi: 10.1101/gr.169532.113. Epub 2014 Jul 22. Erratum in: Genome Res. 2015 Jun;25(6):926. PMID: 25053675; PMCID: PMC4158756.
Comment 2. Line 61. Neither of the two visual arrestin subtypes was ever called b-arrestin.
Response: Thank you very much for your comment. Necessary correction has been made.
Comment 3. Line 65. Arrestins (neither visual nor b-arrestin) were ever implicated in facilitating dissociation of G protein a-subunit from the GPCR. Arrestins compete with heterotrimeric G proteins for active GPCRs and win this competition when GPCR is active and phosphorylated at the same time.
Response: Thank you very much for your comment. Necessary correction has been made, and the corresponding sentence (lines 63-68) has been modified.
Comment 4. Line 67. GPCRs are integral membrane proteins. They cannot be degraded by proteasomes: proteasomes work only with soluble proteins. GPCRs, like all membrane proteins, are degraded by specialized lysosomes (often called multi-vesicular bodies, as vesicles containing membrane proteins are inside the lysosome).
Response: Thank you very much for your very valuable comment. Necessary change has been made, and the corresponding sentence (lines 63-68) has been modified.
The new version of the proposal is presented below (lines 63-68 in the new version of the manuscript)
«In early studies of the role of β-arrestins in the GPCR-signaling, it was recognized that these proteins are responsible for the desensitization of GPCRs and also mediate the endocytosis of ligand-receptor complexes. The study of the further processing of these complexes showed that they are first internalized and transported to early endosomes. Then, the GPCRs are either recycled back to the plasma membrane in an active, ligand-free state, or they are sorted within the endocytic pathway and packaged into intraluminal vesicles, forming multivesicular bodies that fuse with lysosomes, and this leads to complete degradation of the receptors (Dores, Trejo, 2015; Li X. et al., 2019)».
References:
Dores, M.R.; Trejo, J. GPCR sorting at multivesicular endosomes. Methods Cell. Biol. 2015, 130, 319-332, doi: 10.1016/bs.mcb.2015.05.006.
Li, X.; Rosciglione, S.; Laniel, A.; Lavoie, C. Combining RNAi and Immunofluorescence Approaches to Investigate Post-endocytic Sorting of GPCRs into Multivesicular Bodies. Methods Mol. Biol. 2019, 1947, 303-322, doi: 10.1007/978-1-4939-9121-1_17.
Comment 5. Lines 221, 1580-1581. PAMs and NAMs are opposites: PAMs facilitate signaling, whereas NAMs suppress it (as correctly described below line 221). There cannot be such a thing as “ago-NAM”.
Response: Thank you very much for your very important comment. The classification, including ago-NAM, considers various pharmacological scenarios that can be realized with the participation of allosteric regulators. Indeed, PAMs and NAMs are opposite in their modulatory activity on the signal generated by an orthosteric agonist. However, they are not able to influence the activity of the receptor themselves, which distinguishes them from ago-PAMs and ago-NAMs, which are able to activate the receptor in the absence of an orthosteric agonist, and this is a very real case, which was also shown by us in the study of low-molecular-weight allosteric regulators of the LH and TSH receptors.
Unlike ago-PAMs, ago-NAMs stimulate the receptor (agonist), but at the same time negatively affect the efficacy and (or) affinity of the orthosteric agonist, thus their activity can be considered as an agonist plus NAM. This is presented in Table 1 (The classification of allosteric regulators of GPCRs): “It (ago-NAM) functions as a full agonist and at the same time reduces the affinity and/or efficacy of an orthosteric agonist”. This Table takes into account the classification given in the review article (Grundmann et al., 2021), which presents concentration curves (dose-response), including for ago-NAMs.
Grundmann M, Bender E, Schamberger J, Eitner F. Pharmacology of Free Fatty Acid Receptors and Their Allosteric Modulators. Int J Mol Sci. 2021 Feb 10;22(4):1763. doi: 10.3390/ijms22041763. PMID: 33578942; PMCID: PMC7916689.
Lines 1580-1581 describe the effects of autoantibodies to β1-AR, and for one of them (indexed ‘P4’) it is indicated that, by their activity, they can be considered as ago-NAMs (Bornholz et al., 2013). Indeed, ‘P4’ autoantibodies stimulate the activity of β1-AR (an allosteric agonist), but at the same time reduce isoproterenol-mediated cAMP stimulation (NAM). However, since the authors (Bornholz et al., 2013) did not assign ‘P4’ autoantibodies to the ago-NAM group, the sentences (corresponding to lines 1576-1581 in the earlier version) have been changed in the new version (only the possibility of classifying ‘P4’ autoantibodies as ago-NAM is indicated).
“Among the pattern of autoantibodies to β1-AR isolated from the blood of patients with dilated cardiomyopathy, antibodies designated as “P4” predominate, which suppress the internalization of receptors, cause a β1-AR-mediated increase in intracellular cAMP levels in the absence of an agonist, and inhibit the stimulating effect of isoproterenol on AC activity [439]. According to a number of pharmacological characteristics, these antibodies can be assigned to the ago-NAM group (Table 1), with activity biased towards cAMP-dependent signaling pathways”.
Bornholz B, Weidtkamp-Peters S, Schmitmeier S, Seidel CA, Herda LR, Felix SB, Lemoine H, Hescheler J, Nguemo F, Schäfer C, Christensen MO, Mielke C, Boege F. Impact of human autoantibodies on β1-adrenergic receptor conformation, activity, and internalization. Cardiovasc Res. 2013 Mar 1;97(3):472-80. doi: 10.1093/cvr/cvs350. Epub 2012 Dec 3. PMID: 23208588; PMCID: PMC3567785.
Comment 6. Table 2, item IV, and Fig. 1. RAMPs bind transmembrane helices of GPCRs.
Response: Thank you very much for your very valuable comment. The localization of RAMP interaction with receptors is specified both in the Table 2 and in the Figure 1. Corresponding changes have been made to the text. Made a text insert after line 381 (see below).
“The G-proteins and beta-arrestins interact with the cytoplasmic regions of the GPCRs and their interfaces with the TMD. At the same time, RAMPs interact with the TMD of GPCRs and, in the case of class B GPCRs, with the extracellular domain (Pioszak, Hay, 2020; Kotliar et al., 2023), indirectly affecting the interaction of receptors with transducer proteins by changing the conformation of TM6 and ICL2 (Kotliar et al., 2023)”
References:
Kotliar, I.B.; Lorenzen, E.; Schwenk, J.M.; Hay, D.L.; Sakmar, T.P. Elucidating the Interactome of G Protein-Coupled Receptors and Receptor Activity-Modifying Proteins. Pharmacol. Rev. 2023, 75, 1-34, doi: 10.1124/pharmrev.120.000180.
Pioszak, A.A.; Hay, D.L. RAMPs as allosteric modulators of the calcitonin and calcitonin-like class B G protein-coupled receptors. Adv. Pharmacol. 2020, 88, 115-141, doi: 10.1016/bs.apha.2020.01.001.
Comment 7. Section 5. While the author extensively cites papers suggesting the existence of class A GPCR oligomers, equally numerous papers presenting opposite evidence are not cited and discussed. The studies demonstrating that many reported changes in signaling can be just as easily explained by well-established cross-talk of signaling pathways downstream of GPCRs were not mentioned or cited.
Response: Thank you very much for your comment. In full compliance with the reviewer's request, I have added information about the facts that demonstrate the limitations of the concept of the role of heterodi(oligo)merization in the functioning of class A GPCRs. This is reflected in the box that is included at the end of Section 5 (see below).
“Along with this, there are alternative data that signal transduction through class A GPCRs does not require the formation of a di(oligo)heteromeric complex between their protomers (Gurevich, Gurevich, 2018). In this case, the functional interaction between uncomplexed protomers belonging to different GPCR types is due to the influence of one protomer on the internalization, traffic, and GRK-mediated phosphorylation of another protomer (Torvinen et al., 2005; He S.Q. et al., 2011; Tóth et al., 2018). In addition, competition between protomers for binding to β-arrestins can make a certain contribution (Schmidlin et al., 2002). All this must be taken into account when evaluating the contribution of the formation of receptor complexes to the allosteric regulation of the GPCR-signaling. In any case, it is necessary to differentiate the mechanisms of allosteric regulation that take place in GPCR complexes, in monomeric forms of GPCRs, as well as in the case of dynamic monomeric-dimeric equilibrium among GPCR protomers (Gurevich, Gurevich, 2018; Dale et al., 2022)”.
References:
Dale, N.C.; Johnstone, E.K.M.; Pfleger, K.D.G. GPCR heteromers: An overview of their classification, function and physiological relevance. Front. Endocrinol. (Lausanne) 2022, 13, 931573, doi: 10.3389/fendo.2022.931573.
Gurevich, V.V.; Gurevich, E.V. GPCRs and Signal Transducers: Interaction Stoichiometry. Trends Pharmacol. Sci. 2018, 39, 672-684, doi: 10.1016/j.tips.2018.04.002.
He, S.Q.; Zhang, Z.N.; Guan, J.S; Liu, H.R.; Zhao, B.; Wang, H.B.; Li, Q.; Yang, H.; Luo, J.; Li, Z.Y.; Wang, Q.; Lu, Y.J.; Bao, L.; Zhang, X. Facilitation of μ-opioid receptor activity by preventing δ-opioid receptor-mediated codegradation. Neuron 2011, 69, 120-131, doi: 10.1016/j.neuron.2010.12.001.
Schmidlin, F.; Déry, O.; Bunnett, N.W.; Grady, E.F. Heterologous regulation of trafficking and signaling of G protein-coupled receptors: beta-arrestin-dependent interactions between neurokinin receptors. Proc. Natl. Acad. Sci. USA 2002, 99, 3324-3329, doi: 10.1073/pnas.052161299.
Torvinen, M.; Torri, C.; Tombesi, A.; Marcellino, D.; Watson, S.; Lluis, C.; Franco, R.; Fuxe, K.; Agnati, L.F. Trafficking of adenosine A2A and dopamine D2 receptors. J. Mol. Neurosci. 2005, 25, 191-200, doi: 10.1385/JMN:25:2:191.
Tóth, A.D.; Prokop, S.; Gyombolai, P.; Várnai, P.; Balla, A.; Gurevich, V.V.; Hunyady, L.; Turu, G. Heterologous phosphorylation-induced formation of a stability lock permits regulation of inactive receptors by β-arrestins. J. Biol. Chem. 2018, 293, 876-892, doi: 10.1074/jbc.M117.813139.
Additional references (highlighted in yellow):
- Dores, M.R.; Trejo, J. GPCR sorting at multivesicular endosomes. Methods Cell. Biol. 2015, 130, 319-332, doi: 10.1016/bs.mcb.2015.05.006.
- Li, X.; Rosciglione, S.; Laniel, A.; Lavoie, C. Combining RNAi and Immunofluorescence Approaches to Investigate Post-endocytic Sorting of GPCRs into Multivesicular Bodies. Methods Mol. Biol. 2019, 1947, 303-322, doi: 10.1007/978-1-4939-9121-1_17.
- Pioszak, A.A.; Hay, D.L. RAMPs as allosteric modulators of the calcitonin and calcitonin-like class B G protein-coupled receptors. Adv. Pharmacol. 2020, 88, 115-141, doi: 10.1016/bs.apha.2020.01.001.
- Kotliar, I.B.; Lorenzen, E.; Schwenk, J.M.; Hay, D.L.; Sakmar, T.P. Elucidating the Interactome of G Protein-Coupled Receptors and Receptor Activity-Modifying Proteins. Pharmacol. Rev. 2023, 75, 1-34, doi: 10.1124/pharmrev.120.000180.
- Gurevich, V.V.; Gurevich, E.V. GPCRs and Signal Transducers: Interaction Stoichiometry. Trends Pharmacol. Sci. 2018, 39, 672-684, doi: 10.1016/j.tips.2018.04.002.
X6
- Torvinen, M.; Torri, C.; Tombesi, A.; Marcellino, D.; Watson, S.; Lluis, C.; Franco, R.; Fuxe, K.; Agnati, L.F. Trafficking of adenosine A2A and dopamine D2 receptors. J. Mol. Neurosci. 2005, 25, 191-200, doi: 10.1385/JMN:25:2:191.
- He, S.Q.; Zhang, Z.N.; Guan, J.S; Liu, H.R.; Zhao, B.; Wang, H.B.; Li, Q.; Yang, H.; Luo, J.; Li, Z.Y.; Wang, Q.; Lu, Y.J.; Bao, L.; Zhang, X. Facilitation of μ-opioid receptor activity by preventing δ-opioid receptor-mediated codegradation. Neuron 2011, 69, 120-131, doi: 10.1016/j.neuron.2010.12.001.
- Tóth, A.D.; Prokop, S.; Gyombolai, P.; Várnai, P.; Balla, A.; Gurevich, V.V.; Hunyady, L.; Turu, G. Heterologous phosphorylation-induced formation of a stability lock permits regulation of inactive receptors by β-arrestins. J. Biol. Chem. 2018, 293, 876-892, doi: 10.1074/jbc.M117.813139.
- Schmidlin, F.; Déry, O.; Bunnett, N.W.; Grady, E.F. Heterologous regulation of trafficking and signaling of G protein-coupled receptors: beta-arrestin-dependent interactions between neurokinin receptors. Proc. Natl. Acad. Sci. USA 2002, 99, 3324-3329, doi: 10.1073/pnas.052161299.
- Dale, N.C.; Johnstone, E.K.M.; Pfleger, K.D.G. GPCR heteromers: An overview of their classification, function and physiological relevance. Front. Endocrinol. (Lausanne) 2022, 13, 931573, doi: 10.3389/fendo.2022.931573.

Round 3
Reviewer 4 Report
My concerns have been mostly addressed.
The wording in lines 381-385 still does not reflect what every arrestin-GPCR structure showed: at least one arrestin element, the finger loop, invariably interacts with the cavity formed by TM helices of the receptor, which opens on the cytoplasmic side of the receptor upon its activation. In fact, the elements of G proteins and GRKs engage the same inter-helical cavity, as relevant structures show.
Author Response
RESPONSE TO REVIEWER #4 (the third round)
“Allosteric regulation of G-protein-coupled receptors: from diversity of molecular mechanisms to multiple allosteric sites and their ligands” (Alexander O. Shpakov)
I am very grateful to the Reviewer for a thorough analysis of my review article " Allosteric regulation of G-protein-coupled receptors: from diversity of molecular mechanisms to multiple allosteric sites and their ligands" and for an additional comment. I sincerely hope that the changes I've made based on this comment have cleared up the concerns.
With best regards, Alexander Shpakov
COMMON COMMENTS:
My concerns have been mostly addressed.
The wording in lines 381-385 still does not reflect what every arrestin-GPCR structure showed: at least one arrestin element, the finger loop, invariably interacts with the cavity formed by TM helices of the receptor, which opens on the cytoplasmic side of the receptor upon its activation. In fact, the elements of G proteins and GRKs engage the same inter-helical cavity, as relevant structures show.
Response:
Thank you very much for your very valuable comment.
The description of the interaction of beta-arrestins and G-proteins with GPCRs in the corresponding text fragment (the lines 381-385) was given by me in a general form, demonstrating that their main targets are the interfaces between cytoplasmic loops and TMD hydrophobic helices and some other regions of the cytoplasmic loops of the GPCRs.
In accordance with the Reviewer's comment, I made an important clarification and indicated that the beta-arrestin finger loop, during interaction with the ligand-activated receptor, is located in the same cavity as the α5 helix of the G-protein alpha-subunit, and this cavity is formed by TM6 and H8 (the cytoplasmic vestibule to transmembrane tunnel). Indeed, beta-arrestin, being recruited by the GRK-phosphorylated GPCR, competitively displaces the G-protein from its complex with the receptor and, thereby, terminates the G-protein-mediated signal transduction. Structural details of this process are demonstrated in a number of works (Chaturvedi et al., 2020, PMID: 32169216; Huang et al., 2020, PMID: 31945771; Staus et al., 2020, PMID: 31945772; and others).
The necessary clarification in the text has been made:
Old version: The G-proteins and β-arrestins interact with the cytoplasmic regions of the GPCRs and their interfaces with the TMD.
New version: The G-proteins and β-arrestins interact with different intracellular regions of GPCRs and with the cytoplasmic part of their TMD, and the most important site for such interaction is the cavity formed by TM6 and the hydrophobic helix H8. Both the α5-helix of the α-subunit of the G-protein, which is responsible for the formation of the complex between the ligand-activated GPCR and the G-protein, and the "finger" region of β-arrestin, which is responsible for the formation of the complex between the GRK-phosphorylated receptor and β-arrestin, interact with this cavity. Displacement of the G-protein α-subunit by β-arrestin from this cavity is the main mechanism mediating the termination of G-protein-mediated signaling after GPCR phosphorylation.
